# TABSTRUCT: MEASURING STRUCTURAL FIDELITY OF TABULAR DATA

**Xiangjian Jiang[1], Nikola Simidjievski[2,1], Mateja Jamnik[1]**
[1]Department of Computer Science and Technology, University of Cambridge, UK
[2]Télécom Paris, Institut Polytechnique de Paris, France
`xj265@cam.ac.uk, nikola.simidjievski@telecom-paris.fr, mateja.jamnik@cl.cam.ac.uk`

## ABSTRACT

Evaluating tabular generators remains a challenging problem, as the unique causal structural prior of heterogeneous tabular data does not lend itself to intuitive human inspection. Recent work has introduced structural fidelity as a tabular-specific evaluation dimension to assess whether synthetic data complies with the causal structures of real data. However, existing benchmarks often neglect the interplay between structural fidelity and conventional evaluation dimensions, thus failing to provide a holistic understanding of model performance. Moreover, they are typically limited to toy datasets, as quantifying existing structural fidelity metrics requires access to ground-truth causal structures, which are rarely available for real-world datasets. In this paper, we propose a novel evaluation framework that jointly considers structural fidelity and conventional evaluation dimensions. We introduce a new evaluation metric, *global utility*, which enables the assessment of structural fidelity even in the absence of ground-truth causal structures. In addition, we present *TabStruct*, a comprehensive evaluation benchmark offering large-scale quantitative analysis on 13 tabular generators from nine distinct categories, across 29 datasets. Our results demonstrate that global utility provides a task-independent, domain-agnostic lens for tabular generator performance. We release the TabStruct benchmark suite, including all datasets, evaluation pipelines, and raw results. Code is available at `https://github.com/SilenceX12138/TabStruct`.

## 1 INTRODUCTION

Tabular data generation is a cornerstone of many real-world machine learning tasks (Borisov et al., 2022; Fang et al., 2024), ranging from training data augmentation (Margeloiu et al., 2024; Cui et al., 2024) to missing data imputation (Zhang et al., 2023; Shi et al., 2025). These applications underscore the importance of generative modelling, which necessitates an appropriate understanding of the underlying data structure (Kingma & Welling, 2014; Goodfellow et al., 2014; Bilodeau et al., 2022). For instance, textual data conforms to the distributional hypothesis, and thus the autoregressive models are a natural workhorse for the text generation process (Zhao et al., 2023; Sahlgren, 2008). In contrast to the homogeneous modalities like text, tabular data can pose a different structural prior due to its heterogeneity – the features within a dataset typically have varying types and semantics, with feature sets that can differ across datasets (Grinsztajn et al., 2022; Shi et al., 2025). Recent work (Hollmann et al., 2025) on tabular foundation predictors has empirically demonstrated that the Structural Causal Model (SCM) is an effective structural prior of tabular data. As such, it is important to investigate how effectively existing tabular generative models capture and leverage the causal structures.

Prior work (Hansen et al., 2023; Qian et al., 2024; Du & Li, 2024; Tu et al., 2024; Livieris et al., 2024; Kapar et al., 2025) has attempted to assess tabular data generators by evaluating the synthetic data they produce. However, the prevailing evaluation paradigms still exhibit three primary limitations, which are summarised in Table 1: *(i) Insufficient tabular-specific fidelity assessments.* Current benchmarks largely adopt evaluation dimensions from homogeneous data modalities, such as density estimation (Alaa et al., 2022), machine learning (ML) efficacy (Xu et al., 2019), and privacy preservation (Kotelnikov et al., 2023). While effective in other modalities, they exhibit conceptual limitations when applied to tabular data – they do not explicitly assess the unique structural prior of tabular

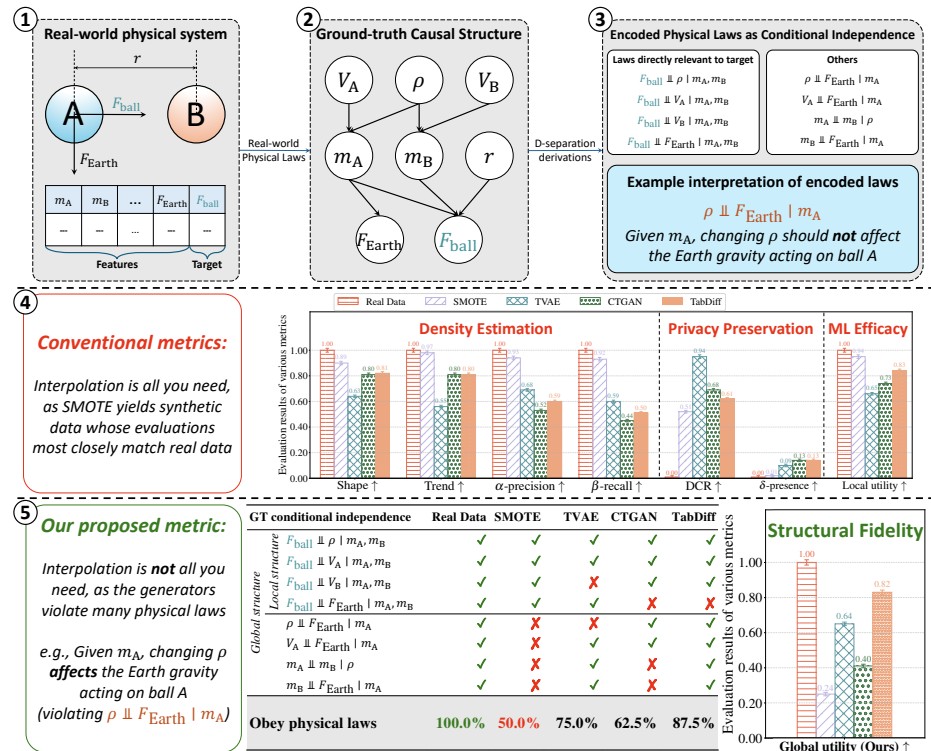

Figure 1: **Illustrative example highlighting the importance of fidelity check for tabular data structure.** ①: A real-world physical system showing the gravitational forces acting on ball A. The system is described by ball density ($\rho$), volume ($V$), masses ($m_A$ & $m_B$), distance ($r$), and gravitational forces ($F_{\text{ball}}$ & $F_{\text{Earth}}$). For simplicity, we assume both balls share identical density. ②: We derive the ground-truth (GT) causal structure of the system based on Newton's law of universal gravitation. ③: We interpret the encoded physical laws of the system as the conditional independence (CI) across variables. ④: We evaluate four generators by conventional metrics. ⑤: We assess the structural fidelity by CI tests and the proposed global utility metric. We note that the *global structure* reflects full conditional independence across all variables, while the local structure includes only those directly relevant to a specific prediction task at hand ($F_{\text{ball}}$). Results demonstrate that conventional metrics are insufficient: for instance, while SMOTE is able to outperform other generators on conventionally used dimensions (e.g., ML efficacy) – the generated synthetic data only preserves local structure and violates most physical laws. For tabular data, where the truthfulness and authenticity of synthetic data is hard to verify, global utility provides an effective mechanism for evaluating the alignment of the synthetic data to the likely ground-truth causal structure.

data. A notable example is that many generators (e.g., SMOTE) can produce synthetic data with similar density estimation as real data, yet still violate underlying causal structures – such as physical laws illustrated in Figure 1(③). Although CauTabBench (Tu et al., 2024) takes a step forward to assess the structural fidelity of synthetic data, it remains confined to toy SCM datasets (i.e., synthetic datasets derived from random SCMs), offering limited insight into real-world tabular data, where the ground-truth SCMs are unavailable. *(ii) Potential evaluation biases.* Many benchmarks (Hansen et al., 2023; Qian et al., 2024) and model studies (Xu et al., 2019; Margeloiu et al., 2024; Zhang et al., 2023) prioritise ML efficacy as the principal dimension for assessing generator performance. For instance, in a classification setting, a generator is often considered effective if its synthetic data allows downstream models to achieve high predictive performance. However, while useful, ML efficacy can be highly sensitive to the choice of prediction task and target (Figure 1(⑤) and Section 3.2). *(iii) Limited evaluation scope.* Existing benchmarks mainly consider only a narrow range of datasets and generative models (Table 1), which restricts their ability to provide a thorough and generalisable comparison of model performance across the broader landscape of tabular generative modelling.

In this paper, we aim to bridge these gaps by introducing a systematic and comprehensive evaluation framework for existing tabular generative models, with a particular focus on the structural prior

Figure 2: **Overview of the proposed evaluation framework.** TabStruct provides a comprehensive evaluation benchmark, including structural fidelity and conventional dimensions, for 13 representative tabular generative models on 29 challenging tabular datasets.

of tabular data. Our proposed framework is characterised by five key concepts: (i) We explicitly incorporate *structural fidelity* of synthetic data as a core evaluation dimension for tabular generative models. Structural fidelity can directly reflect model capability in learning the structure of tabular data, without biasing towards a specific prediction target. In addition, we investigate its interplay with three conventional evaluation dimensions, offering customised guidance for selecting suitable generators across diverse use cases. (ii) We evaluate structural fidelity on expert-validated SCM datasets. To ensure alignment with ground-truth causal structures, we avoid using toy SCMs and instead select SCM datasets with expert-validated causal structures. With ground-truth SCMs, we can quantify structural fidelity through the difference in *conditional independence (CI)* between real and synthetic data as shown in Figure 1(⑤). (iii) We further extend the evaluation of structural fidelity to real-world datasets, where the ground-truth SCMs are unavailable. To this end, we propose a novel evaluation metric, *global utility*, which treats each variable as a prediction target and measures how well it can be predicted using other variables. Importantly, global utility does not require ground-truth causal structures, thus enabling the evaluation of structural fidelity in real-world scenarios. (iv) We conduct an extensive empirical study on the performance of *13 tabular generators spanning nine categories on 29 datasets*, resulting in a total of *over 150,000 evaluations*. The large evaluation scope can ensure holistic and robust benchmarking results. (v) We introduce *TabStruct* (Figure 2), the benchmark suite developed for this work. This open-source library aims to help the research community explore tabular generative modelling within a standardised framework.

Across both SCM and real-world datasets, our primary finding is:

*Structural fidelity, as quantified by the proposed global utility, should be a core dimension when evaluating tabular generative models.*

The benchmark results suggest the prevailing paradigm (i.e., optimising tabular generators primarily for improved density estimation and ML efficacy) is insufficient. In contrast, our proposed global utility provides insights into a crucial yet underexplored perspective – tabular-specific fidelity assessments. Our contributions can be summarised as follows:

- **Conceptual** (Section 3): We propose a unified evaluation framework for tabular generators that integrates structural fidelity with conventional dimensions, and introduce *global utility*, a novel metric that measures structural fidelity without requiring access to ground-truth causal structures.
- **Technical** (Section 3): We release the *TabStruct* benchmark suite, including datasets, generator implementations, evaluation pipelines, and all raw results.
- **Empirical** (Section 4): We conduct a large-scale quantitative study of 13 tabular generators on 29 datasets. The results offer actionable insights into model performance and can inspire the design of more effective tabular generators by attending to the unique structural prior of tabular data.

## 2 RELATED WORK

**Tabular Generator Benchmarks.** An extensive line of benchmarks (Stoian et al., 2025; Hansen et al., 2023; Qian et al., 2024; Du & Li, 2024; Kindji et al., 2024; Sidorenko et al., 2025; Long et al., 2025) has been proposed for tabular data generation, conventionally established around three dimensions: density estimation, privacy preservation, and ML efficacy. Mainstream evaluation metrics typically capture specific aspects of inter-feature interactions. However, they rarely assess whether the underlying causal structures are preserved in the generated tabular data.

Density estimation (Hansen et al., 2023; Alaa et al., 2022; Shi et al., 2025; Zhang et al., 2023) assesses the divergence between real and synthetic data distributions. However, it fails to explicitly

Table 1: **Evaluation scope comparison between TabStruct and prior tabular generative modelling benchmarks.** TabStruct presents a comprehensive evaluation framework for tabular generative models, incorporating a wide range of evaluation dimensions, datasets, and generator categories.

| Benchmark | Conventional dimensions | | | Structural fidelity | | Data | | Generator | |
|---|---|---|---|---|---|---|---|---|---|
| | Density Estimation | Privacy Preservation | ML Efficacy | SCM data | Real-world data | # Datasets | Contamination-free | # Models | # Categories |
| Hansen et al. (2023) | ✔ | ✗ | ✔ | ✗ | ✗ | 11 | ✔ | 5 | 5 |
| Synthcity (Qian et al., 2024) | ✔ | ✔ | ✔ | ✗ | ✗ | 18 | ✗ | 6 | 4 |
| SynMeter (Du & Li, 2024) | ✔ | ✔ | ✔ | ✗ | ✗ | 12 | ✗ | 8 | 4 |
| CauTabBench (Tu et al., 2024) | ✔ | ✗ | ✗ | ✔ | ✗ | 10 | ✔ | 7 | 4 |
| Livieris et al. (2024) | ✔ | ✗ | ✗ | ✗ | ✗ | 2 | ✔ | 5 | 2 |
| SynthEval (Lautrup et al., 2025) | ✗ | ✔ | ✔ | ✗ | ✗ | 1 | ✔ | 5 | 3 |
| Kapar et al. (2025) | ✔ | ✗ | ✔ | ✗ | ✗ | 2 | ✔ | 6 | 4 |
| **TabStruct (Ours)** | ✔ | ✔ | ✔ | ✔ | ✔ | **29** | ✔ | **13** | **9** |

capture inter-feature causal interactions. ML efficacy (Xu et al., 2019; Qian et al., 2024; Seedat et al., 2024; Tiwald et al., 2025) evaluates the performance difference when real data is replaced with synthetic data in downstream tasks, which primarily focuses on $p(y \mid \mathbf{x})$, thus inherently prioritising feature-target relationships over inter-feature interactions. Privacy preservation (Du & Li, 2024; Kotelnikov et al., 2023; Hu et al., 2024; Espinosa & Figueira, 2023), although essential in privacy-sensitive scenarios, is generally task-specific and usually does not necessitate high structural fidelity (Chundawat et al., 2022; Livieris et al., 2024; McLachlan et al., 2018). Recent efforts such as Synthcity (Qian et al., 2024) and SynMeter (Du & Li, 2024) have aimed to standardise the evaluation of tabular data generators by incorporating the three conventional dimensions. Nonetheless, they omit explicit assessment of tabular data structure. To the best of our knowledge, CauTabBench (Tu et al., 2024) is the only other benchmark to explicitly evaluate structural fidelity, but it is limited to toy SCM datasets, as existing metrics (Chen et al., 2023a; Spirtes et al., 2001) typically assume access to the ground-truth SCMs – a condition that is seldom satisfied and arguably infeasible for most real-world datasets (Kaddour et al., 2022; Glymour et al., 2019; Zhou et al., 2024). In addition, some prior studies (Pang et al., 2024; Solatorio & Dupriez, 2023) have attempted to examine relationships across multiple tables within a relational database. However, such approaches remain limited in their ability to reflect inter-feature causal interactions within a single table. We further provide a detailed summary of prior studies on tabular data generation in Appendix A. To bridge these gaps, we introduce TabStruct, a unified evaluation framework, along with global utility, an SCM-free metric that quantifies the preservation of causal structures in tabular data.

## 3 METHODS

### 3.1 PROBLEM SETUP

**Dataset and tabular generator.** Let $\mathcal{D}_{\text{full}} := \{(\mathbf{x}^{(i)}, y^{(i)})\}_{i=1}^N \sim p(\mathbf{x}, y)$ represent a labelled tabular dataset with $\mathbf{x}^{(i)} \in \mathbb{R}^D$. We refer to the $d$-th feature (i.e., a column/variable) as $\boldsymbol{x}_d$, and the $d$-th feature of the $i$-th sample (i.e., a cell) as $x_d^{(i)}$. For notational simplicity, we define $\boldsymbol{x}_{D+1} := \{y^{(i)}\}_{i=1}^N$, so that the full collection of variables, including both features and target, can be written as $\mathcal{X} := \{\boldsymbol{x}_1, \ldots, \boldsymbol{x}_D, \boldsymbol{x}_{D+1}\}$. We denote the training split of $\mathcal{D}_{\text{full}}$ as the reference dataset ($\mathcal{D}_{\text{ref}}$), and test data as $\mathcal{D}_{\text{test}}$. A tabular generator is trained on $\mathcal{D}_{\text{ref}}$ and aims to generate synthetic data $\mathcal{D}_{\text{syn}} \sim p(\tilde{\mathbf{x}}, \tilde{y})$ close to $p(\mathbf{x}, y)$. We evaluate the quality of $\mathcal{D}_{\text{ref}}$ wrt. all the metrics, thus providing a benchmark performance against which $\mathcal{D}_{\text{syn}}$ is compared. We refer to any dataset being assessed as "evaluation dataset $\mathcal{D}$", thus, both $\mathcal{D}_{\text{ref}}$ and $\mathcal{D}_{\text{syn}}$ may serve as evaluation datasets.

**Structural causal models (SCM).** Under the assumptions of causal sufficiency, the Markov property, and faithfulness, an SCM is defined by the quadruple $M := \langle \mathcal{X}, \mathcal{G}, \mathcal{F}, \mathcal{E} \rangle$. $\mathcal{G}$ is the causal graph that encodes the causal relationships among the variables. $\mathcal{E} := \{\boldsymbol{\epsilon}_j\}_{j=1}^{D+1}$ denotes the exogenous noise, and $\mathcal{F} := \{f_j\}_{j=1}^{D+1}$ is the set of structural functions. Each variable $\boldsymbol{x}_j$ is determined by a function $f_j$ of its parents and its exogenous noise, that is, $\boldsymbol{x}_j = f_j (\text{pa}(\boldsymbol{x}_j), \boldsymbol{\epsilon}_j)$, where $\text{pa}(\boldsymbol{x}_j) \subseteq \mathcal{X} \setminus \{\boldsymbol{x}_j\}$ denotes the parent set of $\boldsymbol{x}_j$ in the graph $\mathcal{G}$.

**Structural fidelity.** As an empirically effective structural prior for tabular data, SCM provides a formal framework for the underlying generative processes of tabular data (Hollmann et al., 2025; Tu et al., 2024). Therefore, we define the structural fidelity of a tabular generator as the alignment between the SCMs in its synthetic data and the ground-truth causal structures. We further discuss the rationales behind using causal structural prior for tabular data in Appendix C.

## 3.2 Conditional Independence: Quantifying Structural Fidelity with SCM

**Motivation.** We begin by quantifying structural fidelity under the assumption that the ground-truth SCM is available. Following established benchmarks in causal discovery and inference (Spirtes et al., 2001; Kaddour et al., 2022; Tu et al., 2024), we evaluate structural fidelity at the level of the Markov equivalence class. At this level, causal structures are represented as completed partially directed acyclic graphs (CPDAGs). The SCMs of $\mathcal{D}_{\text{ref}}$ and $\mathcal{D}_{\text{syn}}$ are equivalent if they entail the same set of conditional independence (CI) statements (see Figure 1(②&③) for an illustration).

**CI scores at various granularities.** Following prior work (Spirtes et al., 2001; Tu et al., 2024), the full set of CI statements implied by the ground-truth SCM on $\mathcal{D}_{\text{ref}}$ is defined as

$$\mathcal{C}_{\text{global}} \coloneqq \left\{ (\boldsymbol{x}_j \perp\!\!\!\perp \boldsymbol{x}_k \mid S_{j,k}) \mid S_{j,k} \subseteq \mathcal{X} \setminus \{\boldsymbol{x}_j, \boldsymbol{x}_k\} \right\} \cup \left\{ (\boldsymbol{x}_j \not\perp\!\!\!\perp \boldsymbol{x}_k \mid \widehat{S}_{j,k}) \mid \widehat{S}_{j,k} \subsetneq S_{j,k} \right\} \quad (1)$$

where $S_{j,k}$ and $\widehat{S}_{j,k}$ are the d-separation and d-connection sets for $(\boldsymbol{x}_j, \boldsymbol{x}_k)$, respectively. For each CI statement, we assess whether it holds in the evaluation dataset $\mathcal{D}$ (i.e., $\mathcal{D}_{\text{ref}}$ or $\mathcal{D}_{\text{syn}}$) by conducting a CI test at the significance level $\alpha = 0.01$ via

$$\widehat{\mathcal{I}}_\alpha(\boldsymbol{x}_j, \boldsymbol{x}_k \mid S_{j,k}, \widehat{S}_{j,k}; \mathcal{D}) = \begin{cases} 1, & \text{if the CI statement is } \textit{not} \text{ rejected on } \mathcal{D} \text{ at level } \alpha, \\ 0, & \text{otherwise.} \end{cases} \quad (2)$$

To quantify structural fidelity at varying levels of granularity, we define the CI score for any subset of CI statements $\mathcal{C} \subseteq \mathcal{C}_{\text{global}}$ as:

$$\text{CI}(\mathcal{C}, \mathcal{D}) = \frac{1}{|\mathcal{C}|} \sum_{\mathcal{C}} \mathbb{1}\left[ \widehat{\mathcal{I}}_\alpha(\boldsymbol{x}_j, \boldsymbol{x}_k \mid S_{j,k}, \widehat{S}_{j,k}; \mathcal{D}) = 1 \right] \quad (3)$$

where $\text{CI}(\mathcal{C}, \mathcal{D}) \in [0, 1]$ measures the fidelity of selected CI statements in $\mathcal{D}$, and $\mathbb{1}(\cdot)$ denotes the indicator function. A higher CI score indicates that the evaluation dataset more closely aligns with the structure of the ground-truth SCM. Implementation details for the CI scores are in Appendix B.

**Local structure vs. Global structure.** We assess structural fidelity at two levels of granularity: local and global. For local structural fidelity, we define the local CI score, $\text{CI}(\mathcal{C}_{\text{local}}, \mathcal{D})$, by considering only the CI statements that directly involve the prediction target $y$ of a given dataset and predictive task. Specifically, we compute the local CI score using Equation (3) with $\mathcal{C}_{\text{local}} = \big\{ (\boldsymbol{x}_j \perp\!\!\!\perp \boldsymbol{x}_{D+1} \mid S_{j,D+1}) \mid j \in [D] \big\} \cup \big\{ (\boldsymbol{x}_j \not\perp\!\!\!\perp \boldsymbol{x}_{D+1} \mid \widehat{S}_{j,D+1}) \mid j \in [D] \big\}$ (see Figure 1(③) for an illustration). $\mathcal{C}_{\text{local}}$ highlights which features are uninformative for predicting $y$ when conditioned on the corresponding d-separation sets. Therefore, matching the local CI set indicates which features should be ignored when learning $p(y \mid \mathbf{x})$. A higher local CI score suggests the generator faithfully captures the local structure around the target, implying higher utility for downstream predictive tasks (Section 4.2).

For global structural fidelity, we define the global CI score as the CI score computed over the full set of CI statements, that is, $\text{CI}(\mathcal{C}_{\text{global}}, \mathcal{D})$. Global CI provides a comprehensive assessment of the entire causal structure encoded in the dataset, mitigating potential bias towards any particular variable.

**Rationales for CPDAG-level evaluation.** Prior studies (Tu et al., 2024; Spirtes et al., 2001) typically evaluate the causal structure alignment at three different levels: skeleton level, Markov equivalence class level, and causal graph level. At the skeleton level, all causal directions are ignored, resulting in a loss of information about the causal relationships between features. For instance, the causal skeleton is unable to reflect encoded physical laws shown in Figure 1. Therefore, we choose not to evaluate structural fidelity at the skeleton level due to its inability to capture reliable causal relationships across variables. At the causal graph level, structural fidelity is assessed by comparing the directed acyclic graphs (DAGs) of the reference and synthetic datasets, which requires reliable causal discovery methods as basis. However, current causal discovery methods struggle to recover accurate DAGs from tabular data (Zanga et al., 2022; Kaddour et al., 2022; Nastl & Hardt, 2024). Grounding structural fidelity at the DAG level would introduce additional uncertainty on top of results with questionable reliability, making it even harder to draw reliable conclusions.

In contrast, CPDAG-level evaluation strikes a good balance between evaluation efficiency and validity. Unlike full DAG constructing via causal discovery, CPDAG-level evaluation does not require the orientation of all edges, making it a more tractable yet still meaningful metric of structural fidelity. This is supported by the fact that Markov equivalent SCMs serve as minimal I-MAPs (Agrawal et al.,

2018) of the joint distribution factorisation $p(\mathcal{X}) = \prod_{j=1}^{D+1} p(\boldsymbol{x}_j \mid \text{pa}\,(\boldsymbol{x}_j))$, and no causal directions can be further removed. In other words, the CPDAG-level evaluation can retain sufficient real-world semantics for practical use cases, such as the physical laws in Figure 1. Therefore, TabStruct evaluates structural fidelity at the CPDAG level, balancing semantic richness with computational feasibility. More details on the rationale for CPDAG-level evaluation are provided in Appendix C.

### 3.3 GLOBAL UTILITY: SCM-FREE METRIC FOR GLOBAL STRUCTURAL FIDELITY

**Motivation.** The CI scores introduced in Section 3.2 require access to a ground-truth SCM to enumerate the CI statements $\mathcal{C}_{\text{global}}$. However, for real-world datasets, such an SCM is typically unavailable or even non-identifiable, thereby precluding direct evaluation of structural fidelity. Following prior work on tabular foundation models (Hollmann et al., 2025), we adopt an "SCM-inspired and real-data-validated" paradigm to address such a limitation. Specifically, we propose global utility as an SCM-free metric for global structural fidelity.

**Utility per variable.** Given an evaluation dataset $\mathcal{D}$, we treat each variable $\boldsymbol{x}_j \in \mathcal{X}$ as a prediction target. An ensemble of multiple downstream predictors is trained to predict $\boldsymbol{x}_j$ using the remaining variables $\mathcal{X} \setminus \{\boldsymbol{x}_j\}$ as inputs, following a standard supervised learning setup. The predictive performance on $\mathcal{D}_{\text{test}}$ is denoted as $\text{Perf}_j(\mathcal{D})$, measured using *balanced accuracy* for categorical variables and *root mean square error (RMSE)* for numerical variables. We define the utility of variable $\boldsymbol{x}_j$ as the relative performance achieved on evaluation data compared to reference data:

$$\text{Utility}_j(\mathcal{D}) \coloneqq \begin{cases} \text{Perf}_j(\mathcal{D}_{\text{ref}})^{-1}\text{Perf}_j(\mathcal{D}), & \text{if } \boldsymbol{x}_j \text{ is categorical,} \\ \text{Perf}_j(\mathcal{D})^{-1}\text{Perf}_j(\mathcal{D}_{\text{ref}}), & \text{if } \boldsymbol{x}_j \text{ is numerical.} \end{cases} \tag{4}$$

Utility offers a unified perspective for interpreting downstream performance across mixed variable types: $\text{Utility}_j \geq 1$ indicates that downstream predictors trained on $\mathcal{D}$ perform on par with or better than those trained on $\mathcal{D}_{\text{ref}}$ for predicting $\boldsymbol{x}_j$, whereas $\text{Utility}_j < 1$ implies a loss in predictive power. To mitigate the potential bias from a specific downstream predictor, we ensemble nine different predictors with AutoGluon (Erickson et al., 2020). Full technical details are in Appendix B.

**Local utility.** We define the utility of the prediction target $y$, $\text{Utility}_{D+1}(\mathcal{D})$, as local utility, which aligns with the standard metric used to assess the ML efficacy of tabular data generators. The theoretical (Section 3.2) and empirical (Section 4.2) analysis showcases a strong correlation between the local CI score ($\text{CI}\,(\mathcal{C}_{\text{local}}, \mathcal{D})$) and the local utility ($\text{Utility}_{D+1}(\mathcal{D})$), suggesting that local utility is an effective measurement of the local structure around target $y$.

**Global utility.** Building on the heuristics from local utility and local structure, we further examine the relationship between global utility and global structure. We define the global utility as $\text{Global Utility}(\mathcal{D}) \coloneqq \frac{1}{D+1}\sum_{j=1}^{D+1} \text{Utility}_j(\mathcal{D})$. We hypothesise that aggregating the utility across all features can be strongly correlated with the global CI score (i.e., $\text{CI}\,(\mathcal{C}_{\text{global}}, \mathcal{D})$), as global utility is grounded in the observation that a high-fidelity generator should enable accurate conditional prediction of each variable from the others – an idea closely tied to the Markov blanket in SCMs (Fu & Desmarais, 2010; Gao & Ji, 2016). Indeed, our experiments reveal a strong correlation between global CI and global utility (Section 4.2), supporting that global utility serves as an effective and practical metric for evaluating global structural fidelity in the absence of ground-truth SCMs.

**Bias mitigation in global utility.** In contrast to inherently biased local utility, the proposed global utility treats all features fairly. Specifically, we consider predicting each variable associated with different tasks (e.g., binary classification, multi-class classification, regression, etc.). A change in magnitude in predictive performance can reflect different task difficulties depending on the target variable and its type (Feurer et al., 2022; Wistuba et al., 2015; Yogatama & Mann, 2014; Grandini et al., 2020). Consequently, absolute performance scores and their variances are not directly comparable across variables, and aggregating these scores may obscure meaningful differences across tasks (Grinsztajn et al., 2022). To address this, global utility follows the standard practice (Feurer et al., 2022; Grinsztajn et al., 2022) to aggregate normalised utility scores (Equation (4)), providing a more unified perspective on performance across heterogeneous tasks (Section 4.2 and Appendix E.2).

## 4 EXPERIMENTS

We evaluate 13 tabular generators on 29 datasets by focusing on four research questions:

- **Validity of Benchmark Framework (Q1, Section 4.1, and Appendix E.1)**: Can the proposed evaluation framework yield valid evaluation results regarding generator performance?
- **Validity of Global Utility (Q2, Section 4.2, and Appendix E.2)**: Can global utility serve as an effective metric for structural fidelity when ground-truth causal structures are unavailable?
- **Structural Fidelity of Generators (Q3, Section 4.3, and Appendix E.3)**: Can existing tabular generators accurately capture the data structures across both SCM and real-world datasets?
- **Practicability of Global Utility (Q4, Section 4.4, and Appendix E.4)**: Can global utility provide stable and computationally feasible evaluation results for structural fidelity?

**SCM datasets.** To reduce the gap between causal structures in SCM and real-world data, we select six expert-validated SCM datasets with 7-64 features. All SCM datasets are publicly available from `bnlearn` (Scutari, 2011). Full dataset descriptions are provided in Appendix D.

**Real-world datasets.** We observe that many existing generators achieve near-perfect performance on commonly used benchmark datasets (Shi et al., 2025; Zhang et al., 2023), suggesting that these datasets offer limited discriminative power. To address this, we select 14 classification datasets from the hard TabZilla suite (McElfresh et al., 2024), containing 846-98,050 samples and 6-145 features. We further select nine challenging regression datasets, containing 345-22,784 samples and 6-82 features. Full dataset descriptions are available in Appendix D.

**Benchmark generators.** TabStruct includes 13 existing tabular data generation methods of nine different categories. In addition, we include $\mathcal{D}_{\text{ref}}$, where the reference data is used directly for evaluation. Full implementation details are in Appendix D.3.

**Experimental setup.** For each dataset of $N$ samples, we perform nested cross-validation with repeated shuffle, and the details are available in Appendix D.2. Specifically, we first split the dataset into train and test sets (80% train and 20% test), and further split the train set into a training split ($\mathcal{D}_{\text{ref}}$) and a validation split (90% training and 10% validation). For classification datasets, we perform stratified splitting to preserve the class distribution. We shuffle the dataset to repeat the splitting 10 times, summing up to 10 runs per dataset. All benchmark generators are trained on $\mathcal{D}_{\text{ref}}$, and each generator produces a synthetic dataset with $N_{\text{ref}}$ samples. We tune the parameterised generators using Optuna (Akiba et al., 2019) to minimise their average validation loss across 10 repeated runs. Each generator is given at most two hours to complete a single repeat. The reported results are averaged by default over 10 repeats. We aggregate results across all SCM or real-world datasets because the findings are consistent across classification and regression tasks (Appendix E.2). Specifically, we use the average distance to the minimum (ADTM) metric via affine renormalisation between the top-performing and worse-performing models (Grinsztajn et al., 2022; McElfresh et al., 2024; Hollmann et al., 2025; Margeloiu et al., 2024; Jiang et al., 2024). We further provide the detailed configurations (Appendix D) and raw results (Appendix F).

### 4.1 VALIDITY OF BENCHMARK FRAMEWORK (Q1)

**The benchmark results effectively evaluate data quality.** Table 2 demonstrates that all metrics effectively distinguish between high- and low-quality data. Specifically, except for privacy-related metrics, the reference data ($\mathcal{D}_{\text{ref}}$) consistently achieves the highest scores. This is expected, as $\mathcal{D}_{\text{ref}}$ is the ground truth and should score highly on metrics of density estimation, ML efficacy, and structural fidelity. In contrast, privacy metrics reward greater differences from the ground truth to indicate stronger privacy preservation. Since $\mathcal{D}_{\text{ref}}$ is identical to the ground truth, it naturally scores poorly for privacy. These results show that the selected metrics provide appropriate evaluations for data quality. Therefore, we consider the evaluation results to be valid and meaningful for analysis.

**Structural fidelity is complementary to conventional evaluation dimensions, rather than interchangeable.** On SCM datasets, Figure 3 (left) shows that none of the existing evaluation metrics exhibits a strong correlation with global CI. Notably, SMOTE and BN tend to outperform other models by a clear margin in density estimation. However, their performance degrades greatly when it comes to capturing the global structure of tabular data, as reflected by global CI, consistent with our motivating example in Figure 1. This discrepancy reveals the limitations of conventional evaluation dimensions and underscores the need to incorporate structural fidelity for inter-feature causal structures.

Table 2: **Benchmark results of 13 tabular generators on 29 datasets.** We report the normalised mean $\pm$ std metric values across datasets. "N/A" denotes that a specific metric is not applicable. We highlight the First, Second and Third best performances for each metric. For visualisation, we abbreviate "conditional independence" as "CI". The results show that the Top-3 methods in Global CI and Global utility are largely consistent between SCM and real-world datasets. This alignment suggests that the selected SCM datasets represent real-world causal structure, and global utility can serve as an effective metric for global structural fidelity when ground-truth SCM is unavailable.

| Generator | Density Estimation | | | | Privacy Preservation | | ML Efficacy | Structural Fidelity | | |
| --- | --- | --- | --- | --- | --- | --- | --- | --- | --- | --- |
| | Shape $\uparrow$ | Trend $\uparrow$ | $\alpha$-precision $\uparrow$ | $\beta$-recall $\uparrow$ | DCR $\uparrow$ | $\delta$-Presence $\uparrow$ | Local utility $\uparrow$ | Local CI $\uparrow$ | Global CI $\uparrow$ | Global utility $\uparrow$ |
| **SCM datasets** | | | | | | | | | | |
| $\mathcal{D}_{\text{ref}}$ | $1.00_{\pm 0.00}$ | $1.00_{\pm 0.00}$ | $1.00_{\pm 0.00}$ | $1.00_{\pm 0.00}$ | $0.00_{\pm 0.00}$ | $0.00_{\pm 0.00}$ | $0.99_{\pm 0.01}$ | $0.89_{\pm 0.10}$ | $1.00_{\pm 0.00}$ | $0.99_{\pm 0.01}$ |
| SMOTE | $\mathbf{0.82}_{\pm 0.09}$ | $\mathbf{0.85}_{\pm 0.06}$ | $0.60_{\pm 0.17}$ | $\mathbf{0.83}_{\pm 0.01}$ | $0.21_{\pm 0.09}$ | $0.01_{\pm 0.01}$ | $\mathbf{0.92}_{\pm 0.07}$ | $\mathbf{0.82}_{\pm 0.12}$ | $0.30_{\pm 0.11}$ | $0.39_{\pm 0.09}$ |
| BN | $\mathbf{0.80}_{\pm 0.09}$ | $0.73_{\pm 0.10}$ | $\mathbf{0.78}_{\pm 0.10}$ | $0.32_{\pm 0.08}$ | $\mathbf{0.65}_{\pm 0.16}$ | $0.07_{\pm 0.05}$ | $0.41_{\pm 0.17}$ | $0.23_{\pm 0.12}$ | $0.35_{\pm 0.20}$ | $0.49_{\pm 0.24}$ |
| TVAE | $0.59_{\pm 0.10}$ | $0.59_{\pm 0.14}$ | $0.65_{\pm 0.14}$ | $0.36_{\pm 0.06}$ | $\mathbf{0.70}_{\pm 0.10}$ | $0.13_{\pm 0.11}$ | $0.78_{\pm 0.13}$ | $0.50_{\pm 0.21}$ | $0.40_{\pm 0.09}$ | $0.70_{\pm 0.11}$ |
| GOGGLE | $0.46_{\pm 0.16}$ | $0.50_{\pm 0.13}$ | $0.47_{\pm 0.20}$ | $0.36_{\pm 0.09}$ | $0.55_{\pm 0.06}$ | $\mathbf{0.38}_{\pm 0.19}$ | $0.53_{\pm 0.06}$ | $0.42_{\pm 0.27}$ | $0.14_{\pm 0.03}$ | $0.24_{\pm 0.08}$ |
| CTGAN | $0.46_{\pm 0.14}$ | $0.50_{\pm 0.16}$ | $0.71_{\pm 0.13}$ | $0.34_{\pm 0.08}$ | $0.52_{\pm 0.11}$ | $0.19_{\pm 0.15}$ | $\mathbf{0.80}_{\pm 0.11}$ | $0.61_{\pm 0.08}$ | $0.08_{\pm 0.04}$ | $0.26_{\pm 0.10}$ |
| NFlow | $0.31_{\pm 0.15}$ | $0.26_{\pm 0.10}$ | $0.31_{\pm 0.21}$ | $0.15_{\pm 0.09}$ | $\mathbf{0.73}_{\pm 0.14}$ | $\mathbf{0.51}_{\pm 0.13}$ | $0.10_{\pm 0.05}$ | $0.09_{\pm 0.07}$ | $0.09_{\pm 0.07}$ | $0.12_{\pm 0.07}$ |
| ARF | $\mathbf{0.75}_{\pm 0.14}$ | $\mathbf{0.71}_{\pm 0.11}$ | $\mathbf{0.79}_{\pm 0.09}$ | $0.36_{\pm 0.09}$ | $0.50_{\pm 0.13}$ | $0.09_{\pm 0.07}$ | $0.57_{\pm 0.04}$ | $0.21_{\pm 0.09}$ | $0.35_{\pm 0.11}$ | $0.68_{\pm 0.11}$ |
| TabDDPM | $0.62_{\pm 0.11}$ | $0.60_{\pm 0.12}$ | $0.64_{\pm 0.19}$ | $\mathbf{0.39}_{\pm 0.09}$ | $0.44_{\pm 0.19}$ | $0.14_{\pm 0.05}$ | $0.29_{\pm 0.06}$ | $0.17_{\pm 0.08}$ | $\mathbf{0.69}_{\pm 0.08}$ | $\mathbf{0.80}_{\pm 0.05}$ |
| TabSyn | $0.50_{\pm 0.16}$ | $0.48_{\pm 0.17}$ | $0.59_{\pm 0.14}$ | $0.31_{\pm 0.11}$ | $0.45_{\pm 0.14}$ | $\mathbf{0.32}_{\pm 0.21}$ | $0.76_{\pm 0.05}$ | $\mathbf{0.70}_{\pm 0.06}$ | $\mathbf{0.70}_{\pm 0.04}$ | $\mathbf{0.76}_{\pm 0.06}$ |
| TabDiff | $0.69_{\pm 0.11}$ | $0.62_{\pm 0.15}$ | $0.75_{\pm 0.09}$ | $0.36_{\pm 0.09}$ | $0.50_{\pm 0.14}$ | $0.13_{\pm 0.03}$ | $\mathbf{0.80}_{\pm 0.06}$ | $0.58_{\pm 0.14}$ | $\mathbf{0.57}_{\pm 0.15}$ | $\mathbf{0.75}_{\pm 0.07}$ |
| TabEBM | $0.67_{\pm 0.12}$ | $0.57_{\pm 0.15}$ | $\mathbf{0.76}_{\pm 0.04}$ | $0.27_{\pm 0.09}$ | $0.55_{\pm 0.22}$ | $0.14_{\pm 0.06}$ | $0.59_{\pm 0.05}$ | $0.50_{\pm 0.19}$ | $0.26_{\pm 0.11}$ | $0.30_{\pm 0.08}$ |
| NRGBoost | $0.65_{\pm 0.10}$ | $0.50_{\pm 0.15}$ | $0.61_{\pm 0.14}$ | $0.26_{\pm 0.07}$ | $0.53_{\pm 0.12}$ | $0.28_{\pm 0.21}$ | $0.75_{\pm 0.01}$ | $\mathbf{0.64}_{\pm 0.05}$ | $0.11_{\pm 0.05}$ | $0.16_{\pm 0.02}$ |
| GReaT | $0.62_{\pm 0.09}$ | $0.59_{\pm 0.07}$ | $0.62_{\pm 0.10}$ | $\mathbf{0.38}_{\pm 0.07}$ | $0.52_{\pm 0.07}$ | $0.18_{\pm 0.05}$ | $0.27_{\pm 0.09}$ | $0.17_{\pm 0.04}$ | $0.16_{\pm 0.05}$ | $0.25_{\pm 0.08}$ |
| **Real-world datasets** | | | | | | | | | | |
| $\mathcal{D}_{\text{ref}}$ | $1.00_{\pm 0.00}$ | $1.00_{\pm 0.00}$ | $1.00_{\pm 0.00}$ | $1.00_{\pm 0.00}$ | $0.00_{\pm 0.00}$ | $0.00_{\pm 0.00}$ | $0.96_{\pm 0.06}$ | N/A | N/A | $0.99_{\pm 0.01}$ |
| SMOTE | $\mathbf{0.61}_{\pm 0.13}$ | $\mathbf{0.87}_{\pm 0.05}$ | $\mathbf{0.81}_{\pm 0.11}$ | $\mathbf{0.77}_{\pm 0.01}$ | $0.19_{\pm 0.09}$ | $0.02_{\pm 0.02}$ | $\mathbf{0.91}_{\pm 0.07}$ | N/A | N/A | $0.41_{\pm 0.04}$ |
| BN | $\mathbf{0.66}_{\pm 0.11}$ | $\mathbf{0.72}_{\pm 0.09}$ | $\mathbf{0.86}_{\pm 0.09}$ | $\mathbf{0.30}_{\pm 0.04}$ | $0.48_{\pm 0.16}$ | $0.07_{\pm 0.08}$ | $0.38_{\pm 0.16}$ | N/A | N/A | $0.44_{\pm 0.25}$ |
| TVAE | $0.45_{\pm 0.20}$ | $0.50_{\pm 0.14}$ | $0.55_{\pm 0.20}$ | $0.18_{\pm 0.04}$ | $\mathbf{0.68}_{\pm 0.18}$ | $0.29_{\pm 0.18}$ | $0.70_{\pm 0.06}$ | N/A | N/A | $0.53_{\pm 0.13}$ |
| GOGGLE | $0.41_{\pm 0.15}$ | $0.47_{\pm 0.14}$ | $0.57_{\pm 0.16}$ | $0.26_{\pm 0.07}$ | $0.50_{\pm 0.11}$ | $\mathbf{0.35}_{\pm 0.18}$ | $0.46_{\pm 0.04}$ | N/A | N/A | $0.21_{\pm 0.06}$ |
| CTGAN | $0.29_{\pm 0.18}$ | $0.53_{\pm 0.14}$ | $0.56_{\pm 0.14}$ | $0.11_{\pm 0.05}$ | $0.51_{\pm 0.13}$ | $\mathbf{0.30}_{\pm 0.24}$ | $0.70_{\pm 0.06}$ | N/A | N/A | $0.13_{\pm 0.06}$ |
| NFlow | $0.38_{\pm 0.19}$ | $0.28_{\pm 0.16}$ | $0.52_{\pm 0.15}$ | $0.07_{\pm 0.04}$ | $\mathbf{0.64}_{\pm 0.14}$ | $\mathbf{0.42}_{\pm 0.25}$ | $0.10_{\pm 0.06}$ | N/A | N/A | $0.14_{\pm 0.12}$ |
| ARF | $\mathbf{0.61}_{\pm 0.11}$ | $0.58_{\pm 0.12}$ | $\mathbf{0.83}_{\pm 0.10}$ | $0.21_{\pm 0.04}$ | $0.48_{\pm 0.14}$ | $0.05_{\pm 0.04}$ | $0.54_{\pm 0.07}$ | N/A | N/A | $0.56_{\pm 0.12}$ |
| TabDDPM | $0.43_{\pm 0.16}$ | $0.49_{\pm 0.18}$ | $0.54_{\pm 0.22}$ | $0.26_{\pm 0.09}$ | $0.42_{\pm 0.19}$ | $0.27_{\pm 0.18}$ | $0.27_{\pm 0.06}$ | N/A | N/A | $\mathbf{0.72}_{\pm 0.08}$ |
| TabSyn | $0.44_{\pm 0.14}$ | $0.51_{\pm 0.16}$ | $0.62_{\pm 0.18}$ | $0.24_{\pm 0.08}$ | $0.51_{\pm 0.12}$ | $0.24_{\pm 0.14}$ | $\mathbf{0.76}_{\pm 0.08}$ | N/A | N/A | $\mathbf{0.73}_{\pm 0.07}$ |
| TabDiff | $0.54_{\pm 0.15}$ | $0.52_{\pm 0.16}$ | $0.69_{\pm 0.12}$ | $0.22_{\pm 0.07}$ | $0.57_{\pm 0.15}$ | $0.20_{\pm 0.13}$ | $\mathbf{0.78}_{\pm 0.03}$ | N/A | N/A | $\mathbf{0.73}_{\pm 0.07}$ |
| TabEBM | $0.59_{\pm 0.15}$ | $\mathbf{0.65}_{\pm 0.08}$ | $0.79_{\pm 0.04}$ | $\mathbf{0.30}_{\pm 0.10}$ | $\mathbf{0.58}_{\pm 0.10}$ | $0.14_{\pm 0.03}$ | $0.63_{\pm 0.11}$ | N/A | N/A | $0.35_{\pm 0.11}$ |
| NRGBoost | $0.54_{\pm 0.12}$ | $0.49_{\pm 0.13}$ | $0.62_{\pm 0.16}$ | $0.20_{\pm 0.07}$ | $0.51_{\pm 0.15}$ | $0.22_{\pm 0.13}$ | $0.74_{\pm 0.05}$ | N/A | N/A | $0.16_{\pm 0.05}$ |
| GReaT | $0.47_{\pm 0.10}$ | $0.49_{\pm 0.13}$ | $0.57_{\pm 0.14}$ | $0.26_{\pm 0.08}$ | $0.52_{\pm 0.11}$ | $0.27_{\pm 0.15}$ | $0.23_{\pm 0.07}$ | N/A | N/A | $0.20_{\pm 0.06}$ |

## 4.2 VALIDITY OF GLOBAL UTILITY (Q2)

**Global utility serves as an effective metric for global structural fidelity.** Table 2 and Figure 3 (left) demonstrate a strong monotonic correlation between global utility and global CI scores ($r_s = 0.84$, $p < 0.001$). To ensure the generalisability of global utility, we extend our evaluation scope, incorporating more complex SCM datasets (Appendix E.2), a wider range of existing metrics (Appendix E.2), and additional downstream tasks (Appendix E.4). Across all settings, global utility consistently exhibits a substantially stronger correlation with global CI than any other metric. Appendix E.2 further shows that global utility more closely aligns with global CI in the induced generator rankings. We would like to emphasise that the high correlation between global CI and global utility is an empirical finding rather than a formal proof of a connection between the two metrics. This observation primarily aims to offer users actionable and empirically grounded insights into tabular data generation. The strong correlation and consistent generator ranking suggest that global utility offers a robust, SCM-free approach for assessing global structural fidelity.

**Local utility is not always the golden standard, due to its bias towards the local structure.** We further examine the correlation between local utility and local CI, which only considers the local structure associated with the prediction target. As shown in Figure 3 (left), local utility exhibits a strong correlation with local CI ($r_s = 0.78$, $p < 0.001$), but a much weaker correlation with global CI ($r_s = 0.14$, $p < 0.001$). The results indicate that local utility may overlook the holistic data structure, while global utility provides a more comprehensive evaluation of structural fidelity.

## 4.3 STRUCTURAL FIDELITY OF GENERATORS (Q3)

**Structure learning methods struggle with tabular data generation.** One surprising finding is that BN and GOGGLE do not demonstrate strong performance in structural fidelity, despite their inductive bias towards learning tabular data structures. This observation aligns with prior work (Tu et al., 2024; Zeng et al., 2022), which highlights that current causal discovery algorithms often struggle when the number of features exceeds 10 – our benchmark datasets have features from 6

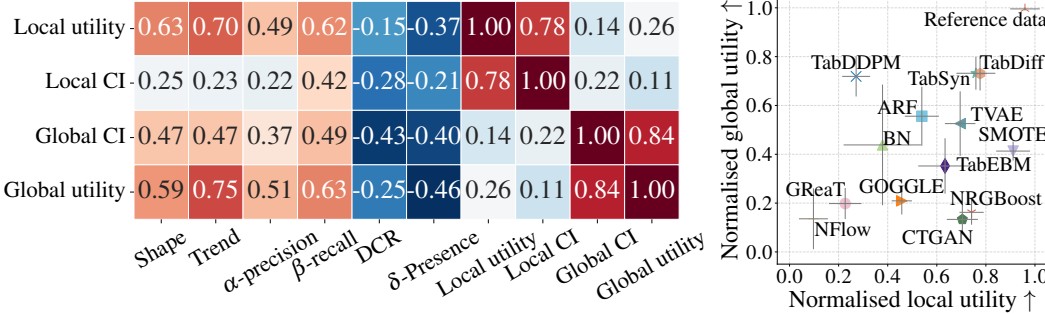

Figure 3: **Left:** Spearman's rank correlation heatmap based on metric values on six SCM datasets. Global utility correlates strongly with global CI, suggesting that global utility can effectively assess global structural fidelity without resorting to SCMs. **Right:** Mean normalised local utility vs. mean normalised global utility on 23 real-world datasets. SMOTE prioritises local utility, whereas TabDiff and TabSyn generally achieve a balanced preservation of both global and local data structures.

up to 145. Furthermore, GOGGLE exhibits notable performance degradation when prior knowledge about the data structure is missing (Liu et al., 2023). The results underscore the limitations of existing causal discovery methods in recovering precise causal structures from real-world data, further justifying our evaluation at the CPDAG level.

**Diffusion models generally capture the global structure well.** As reported in Table 2 and Figure 3 (right), diffusion-based models consistently achieve the highest scores in global structural fidelity: the Top-3 methods are TabDDPM, TabSyn, and TabDiff across both SCM and real-world datasets. We attribute their strong performance to the inherent learning principle of diffusion models for learning permutation-invariant conditional distributions of each feature. At the training stage, since noise is added independently to each feature, the diffusion network is optimised at every denoising step to reconstruct each feature simultaneously by conditioning on others. For instance, TabDDPM and TabDiff implement this principle within each feature type, and TabSyn applies it across all features. Moreover, diffusion models impose no ordering of features. This results in efficient computation (Figure 4) and permutation-invariant conditional distributions, a property that aligns naturally with the structure of tabular data. These theoretical properties align with the conditional independence analysis in Section 3.2, thus confirming that diffusion models are capable of capturing global structure.

**Language models remain limited in learning tabular data structure.** Table 2 shows that the autoregressive model GReaT, even with the help of large language models, fails to outperform even the simple baselines like SMOTE and TVAE. Although token-wise likelihood training is a well-established approach for sequential modalities like text and time series, its underlying assumptions misalign with the permutation-invariant nature of tabular data. An autoregressive generator needs to linearise the feature set and then factorise the joint distribution as $\prod_{j=1}^{d} p(\mathbf{x}_{\pi(j)} \mid \mathbf{x}_{\pi(<j)})$, where $\pi$ denotes a chosen ordering of features. Any fixed ordering $\pi$ can introduce directional bias. For instance, the bias could harm the estimation of $p(\boldsymbol{x}_j \mid \mathcal{X} \setminus \{\boldsymbol{x}_j\})$ when $j$ appears early in the sequence. While GReaT attempts to mitigate this issue by randomising $\pi$ when fine-tuning large language models, randomising $\pi$ does not resolve the fundamental misalignment and can even constrain the performance of autoregressive tabular generators (Appendix E.3).

## 4.4 PRACTICABILITY OF GLOBAL UTILITY (Q4)

**Global utility is robust and stable.** Appendix B.2 and Appendix E.4 show that global utility yields stable generator rankings across both nine tuned predictors ("Full-tuned") and three untuned ones ("Tiny-default"). In contrast, local utility necessitates nine tuned predictors ("Full-tuned") for reliable results. These findings align with the conceptual design of the utility metrics. We note that local utility focuses mainly on the predictive performance of a single target variable, making it susceptible to feature-specific bias, which results in unstable generator rankings across different predictor configurations. In contrast, global utility aggregates performance across all variables, thereby mitigating feature-specific effects and enhancing robustness.

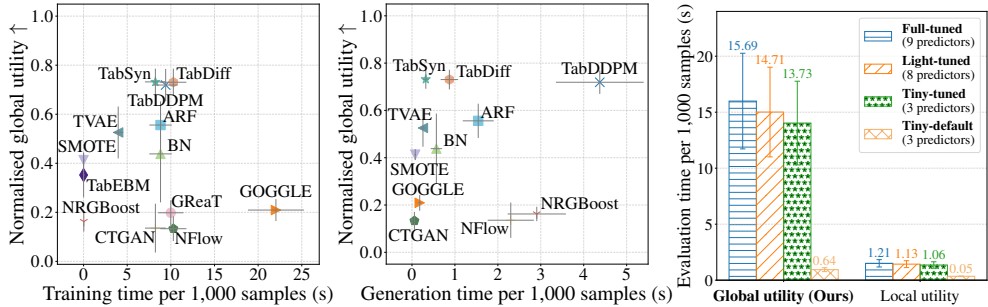

Figure 4: **Computation efficiency on 23 real-world datasets. Left:** Median training time per 1,000 samples vs. mean normalised global utility. **Middle:** Median generation time per 1,000 samples vs. mean normalised global utility. We exclude the outliers (TabEBM and GReaT) due to their long generation time (over 30s). **Right:** Median evaluation time. Because global utility yields stable generator rankings across downstream predictors (Appendix E.4), computing global utility can be highly efficient with only a small ensemble of predictors (i.e., Tiny-default).

**Global utility provides efficient evaluations of structural fidelity.** In practice, we are often interested in identifying the most promising model before fine-tuning it for optimal performance. Global utility supports this by reducing both the tuning burden and the dependency on the number of predictors, while still yielding stable and informative rankings. As illustrated in Figure 4 (right), computing global utility with "Tiny-default" takes only 0.64s per 1000 samples, while local utility requires nearly double the time ("Full-tuned" with 1.21s) for comparable reliability.

**Limitations and future work.** While our proposed global utility is a robust and effective metric for assessing global structural fidelity, it is an empirical measurement of the likely SCMs behind the data at hand. However, developing a theoretically provable structural fidelity metric for real-world tabular data is highly challenging, as ground-truth causal structures are rarely available, even precluding the possibility of theoretical validation. This is in line with several open challenges in the field – particularly the lack of causal discovery methods that can reliably infer the governing SCMs of real-world tabular datasets (Kaddour et al., 2022; Tu et al., 2024; Glymour et al., 2019; Nastl & Hardt, 2024). Despite substantial research efforts, recent work (Nastl & Hardt, 2024) shows that even state-of-the-art causal discovery methods often perform poorly on real-world data and may mislead users. Therefore, we propose global utility primarily as an empirical lens for evaluating tabular data structures. Bridging the gap between theoretical assumptions and real-world causal structures will require advances in causal modelling. As TabStruct library is freely available, its development will be an ongoing, community-driven endeavour. Therefore, TabStruct will continue to evolve with advances in causal modelling. We believe that the open-source nature of TabStruct will help drive progress in theoretical foundations for real-world tabular data challenges. More discussion on future work is in appendix E.5 and Appendix E.6.

## 5 CONCLUSION

We present TabStruct, a principled benchmark for tabular data generators along with both structural fidelity and conventional dimensions. To address the challenge of assessing structural fidelity in the absence of ground-truth SCMs, we introduce global utility – a novel, SCM-free metric that enables unbiased and holistic evaluation for tabular data structure.

In our large-scale study of 13 generators across 29 datasets, we find that existing evaluation methods often favour models that capture local causal interactions while neglecting global structure. Our results show that the four evaluation dimensions are complementary, offering practical guidance for selecting suitable generators across diverse applications. We further observe that diffusion models, due to their permutation-invariant generation process, offer valuable insights into the fundamental representation learning of tabular data. TabStruct is an ongoing effort. As such, it will continue to evolve with additional datasets, generators, and evaluation metrics – both through our engagement and contributions from the community. We envision that the open-source nature of TabStruct will help drive progress in high-fidelity tabular generative modelling.

## ETHICS STATEMENT

This paper proposes integrating structural fidelity as a core evaluation dimension alongside conventional metrics for assessing tabular data generators. Specifically, we introduce global utility, a novel metric that evaluates the structural fidelity of synthetic tabular data without requiring access to the ground-truth causal structures. Furthermore, we present TabStruct, a comprehensive benchmark for tabular data generation that spans a wide evaluation scope – comprising 13 generators from nine distinct categories, evaluated on 29 datasets. Our benchmark results highlight that structural fidelity is an important yet previously underexplored evaluation dimension. It effectively captures whether generated data preserves the underlying causal structures present in real-world tabular datasets, serving as a valuable complement to existing evaluation dimensions.

This is particularly critical for tabular modalities, where visual inspection of data authenticity is not feasible, unlike in text or image domains (Van Breugel & Van Der Schaar, 2024; Zhao et al., 2023). By providing a unified benchmark that incorporates both conventional metrics and structural fidelity, TabStruct has the potential to foster more reliable and transparent development of generative models. This can benefit multiple domains that rely on tabular data, such as healthcare (Jiang et al., 2024; Bespalov et al., 2016; Morford et al., 2011) and scientific research (Margeloiu et al., 2024), where understanding the structural fidelity of generated data is crucial.

The impact of our work extends to enabling broader machine learning applications in data-scarce domains. For instance, it can facilitate robust data analysis in clinical contexts where data collection is limited (Margeloiu et al., 2024; Chawla et al., 2002; McLachlan et al., 2018). Enhancing the fidelity of synthetic data may promote the adoption of more advanced machine learning approaches. TabStruct could further facilitate safer data sharing in privacy-sensitive contexts (Jordon et al., 2018; Hu et al., 2024; Stoian et al., 2025; Alami et al., 2020; Ciecierski-Holmes et al., 2022), support reproducible research through synthetic benchmarks, and broaden access to machine learning capabilities in low-resource or data-scarce scenarios.

## REPRODUCIBILITY STATEMENT

Our study is conducted entirely within a reproducible setting. As detailed in Appendix D, all benchmark datasets are publicly available and widely adopted in the machine learning literature (McElfresh et al., 2024; Scutari, 2011). We do not use, include, or release any newly collected or proprietary data. In addition, the employed tabular generative models and benchmark metrics are not tailored to any specific demographic or domain-sensitive dataset. Full implementation details are available in Appendix D and the associated codebase (`https://github.com/SilenceX12138/TabStruct`). Furthermode, we release TabStruct as an open-source library to support transparency, reproducibility, and further community-driven development. We welcome community contributions that prioritise safety, fairness, and inclusivity in the future evolution of the benchmark.

## ACKNOWLEDGEMENTS

The authors would like to express their gratitude to Prof. Carl Henrik Ek for insightful discussions on structure learning, and to Prof. Ferenc Huszár and Dr. Ruibo Tu for their enlightening perspectives on causal machine learning, and to Dr. Andrei Margeloiu for his thoughtful feedback during the early stages of the project. XJ acknowledges the generous support of the Google PhD Fellowship. MJ and NS acknowledge the support of the U.S. Army Medical Research and Development Command of the Department of Defense; through the FY22 Breast Cancer Research Program of the Congressionally Directed Medical Research Programs, Clinical Research Extension Award GRANT13769713. Opinions, interpretations, conclusions, and recommendations are those of the authors and are not necessarily endorsed by the Department of Defense.

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

# Appendix

## TabStruct: Measuring Structural Fidelity of Tabular Data

## Table of Contents

# A    SUMMARY OF RELATED WORK

As a supplement to Section 2, we provide a detailed summary of related work on tabular data generation. We begin by outlining the conventional evaluation dimensions for tabular generators (Appendix A.1). We then highlight the importance of assessing structural fidelity in the evaluation of such models (Appendix A.2). We further summarise existing tabular data generators (Appendix A.3). Finally, we present a comprehensive and quantitative comparison of the evaluation scope covered by TabStruct versus prior work, including both benchmarks and model studies (Appendix A.4).

## A.1    CONVENTIONAL EVALUATION DIMENSIONS

**Density estimation** assesses the discrepancy between the distributions of reference and synthetic data, considering both marginal (i.e., low-order) and joint (i.e., high-order) distributions (Hansen et al., 2023; Kim et al., 2023; McCarter, 2024; Solatorio & Dupriez, 2023; Pang et al., 2024). A generator may achieve high performance on low-order metrics by sampling each feature independently, thereby ignoring inter-feature dependencies. While high-order metrics aim to measure sample-level similarity, they still fall short of explicitly revealing whether the synthetic data preserves the underlying causal structures present in the reference data.

Following prior studies (Hansen et al., 2023; Shi et al., 2025; Zhang et al., 2023), we evaluate density estimation using four metrics of two categories: (i) Low-order: *Shape* and *Trend* (Wüst, 2011). Shape measures the synthetic data's ability to replicate each column's marginal density. Trend assesses its capacity to capture correlations between different columns. (ii) High-order: $\alpha$-*precision* and $\beta$-*recall* (Alaa et al., 2022). $\alpha$-*precision* quantifies the similarity between the reference and synthetic data, and $\beta$-*recall* assesses the diversity of the synthetic data.

**Privacy preservation** evaluates the trade-off between the utility of synthetic data in downstream tasks and the risk of privacy leakage (Margeloiu et al., 2024; Gulati & Roysdon, 2023; Truda, 2023; Jordon et al., 2018; Zhang et al., 2021; McKenna et al., 2021; 2019). However, this dimension is often tailored to specific tasks (e.g., classification and regression), and as such, it does not directly evaluate the structural fidelity of tabular data. Consequently, privacy preservation alone cannot comprehensively assess a generator's ability to capture the fundamental characteristics of tabular data, such as causal structures.

Following prior studies (Margeloiu et al., 2024; Kotelnikov et al., 2023; Zhao et al., 2021), we measure privacy preservation using two metrics: (i) *median Distance to Closest Record* (DCR) (Zhao et al., 2021), where a higher DCR indicates that synthetic data is less likely to be directly copied from the reference data; (ii) $\delta$-*Presence* (Qian et al., 2024). We note that some implementations of $\delta$-Presence interpret smaller values as indicative of better privacy preservation; however, we adapt the implementation provided by Synthcity (Qian et al., 2024), wherein larger values correspond to improved privacy preservation. Furthermore, we note that Synthcity implementation can exhibit instability due to inherent numerical issues.

**ML efficacy** measures the performance gap observed when replacing reference data with synthetic data in downstream tasks. This metric is inherently task-specific and can be heavily biased by the choice of predictive models and target variables. A useful parallel can be drawn from image generation: Mixup (Psaroudakis & Kollias, 2022) enhances training data by interpolating between real samples, often improving downstream task performance. However, it simultaneously distorts the spatial structure of images, producing visually unrealistic outputs (Mumuni & Mumuni, 2022). As illustrated in Figure 1, assessing the authenticity of synthetic tabular data is far more difficult than in image domains. Consequently, synthetic data that performs well in downstream tasks may still fail to preserve important causal structures of the reference data. This example shows that ML efficacy, while useful for specific tasks, cannot serve as a holistic measure of a tabular data generator.

Following prior studies (Xu et al., 2019; Margeloiu et al., 2024; Seedat et al., 2024), we adopt the "train-on-synthetic, test-on-real" strategy for quantifying ML efficacy of synthetic data. To mitigate the bias from downstream models, we evaluate the utility with the performance of an ensemble of nine predictors (i.e., AutoGluon-full (Erickson et al., 2020) and TabPFN (Hollmann et al., 2025)). Specifically, the downstream models include three standard baselines: Logistic Regression (LR) (Cox, 1958), KNN (Fix, 1985) and MLP (Gorishniy et al., 2021); five tree-based methods: Random Forest (RF) (Breiman, 2001), Extra Trees (Erickson et al., 2020), LightGBM (Ke et al., 2017),

CatBoost (Prokhorenkova et al., 2018), and XGBoost (Chen & Guestrin, 2016); and a PFN method: TabPFN (Hollmann et al., 2025).

Furthermore, as noted in prior work (Kotelnikov et al., 2023), tuning downstream models does affect the relative rankings of tabular generators under ML efficacy. Therefore, to draw generalisable conclusions, we perform hyperparameter tuning for all nine predictors, and the technical details are provided in Appendix D.

## A.2 STRUCTURAL FIDELITY OF TABULAR DATA

As illustrated in Figure 1, one of the key desiderata for faithful synthetic tabular data is the preservation of causal structures present in real data. Prior work (Tu et al., 2024) primarily assesses structural fidelity using toy datasets, as existing metrics (Chen et al., 2023a; Spirtes et al., 2001) typically assume access to the ground-truth SCMs – a condition that is seldom satisfied and arguably infeasible for most real-world datasets (Kaddour et al., 2022; Glymour et al., 2019; Zhou et al., 2024; Nastl & Hardt, 2024).

To bridge this gap, we introduce *global utility*, an SCM-free metric that quantifies how well a generator preserves the causal structure of real data. Global utility provides a complementary perspective to conventional metrics, enabling a more holistic assessment of synthetic tabular data.

## A.3 TABULAR DATA GENERTOR

The common paradigm for tabular data generation is to adapt Generative Adversarial Networks (GANs) and Variational Autoencoders (VAEs) (Xu et al., 2019). For instance, TableGAN (Park et al., 2018) employs a convolutional neural network to optimise the label quality, and TVAE (Xu et al., 2019) is a variant of VAE for tabular data. However, these methods learn the joint distribution and thus cannot preserve the stratification of the reference data (Margeloiu et al., 2024). CTGAN (Xu et al., 2019) refines the generation to be class-conditional. The recent ARF (Watson et al., 2023) is an adversarial variant of random forest for density estimation, and GOGGLE (Liu et al., 2023) enhances VAE by learning relational structure with a Graph Neural Network (GNN). Another emerging direction is the use of denoising diffusion models (Kotelnikov et al., 2023; Zhang et al., 2023; Shi et al., 2025). For instance, TabDDPM (Kotelnikov et al., 2023) demonstrates that diffusion models can approximate typical distributions of tabular data. In addition, several energy-based models have recently been proposed for tabular data generation, such as TabEBM (Margeloiu et al., 2024) and NRGBoost (Bravo, 2025). These models aim to improve synthetic data quality by learning energy-based representations of the data distribution.

In a broader context, there is growing interest in adapting Large Language Models (LLMs) for tabular data generation (Fang et al., 2024; Seedat et al., 2024; Borisov et al., 2023). For example, GReaT fine-tunes GPT-2 to generate realistic tabular data, while CLLM leverages the domain knowledge embedded in LLMs during generation. However, most state-of-the-art LLMs do not disclose their pretraining data, raising concerns about data contamination — i.e., whether the reference data (even the test data) has been included during pretraining (Fang et al., 2024; Margeloiu et al., 2024), which can undermine fair comparisons between tabular generators. To ensure fairness and reproducibility, TabStruct excludes models based on proprietary or undisclosed LLMs, such as GPT-4 (Seedat et al., 2024). We restrict our evaluation to models built on fully open-source LLMs, such as GReaT, thereby mitigating concerns related to data contamination. We would like to emphasise that, although TabStruct excludes certain LLM-based tabular generators to ensure fair and uncontaminated benchmarking, researchers and practitioners are encouraged to integrate their own LLM-based models.

We acknowledge that some models exist beyond those currently implemented in TabStruct. We note that TabStruct offers unified APIs that support up to nine distinct categories of tabular generators (one of the widest scopes to date shown in Table 4), enabling broad compatibility for most tabular generators. Therefore, beyond its current evaluation scope, TabStruct functions as a standardised and extensible benchmarking framework. It is designed to accommodate future methods, promoting continued development and evaluation within a consistent and reproducible environment.

## A.4 EVALUATION SCOPE COMPARISON

Table 3 and Table 4 present a comparative analysis of TabStruct against prior studies on the evaluation of tabular generative models. TabStruct considers four key evaluation dimensions: density estimation, privacy preservation, ML efficacy, and structural fidelity. In addition, it supports all nine categories of

tabular generators, offering a more comprehensive and holistic overview of the current landscape of generative modelling for tabular data.

While we acknowledge that the CauTabBench framework is, in principle, scalable to datasets with higher dimensions than those reported in its original study, we emphasise that the specific causal discovery methods it employs may not be practically scalable in real-world scenarios. For instance, prior work (Zanga et al., 2022) has highlighted the substantial computational overhead associated with causal discovery algorithms such as PC. Empirically, we observe that the vanilla PC algorithm used in CauTabBench may require up to 168 hours (i.e., 7 days) to process datasets with more than 50 features. Consequently, several metrics within CauTabBench may be computationally infeasible for the real-world datasets considered in TabStruct, suggesting that CauTabBench would require additional technical optimisation for practical deployment. Moreover, recent studies (Nastl & Hardt, 2024; Zanga et al., 2022) have shown that even state-of-the-art causal discovery methods often perform unreliably on real-world data, potentially leading to misleading conclusions. We also observe such pitfalls of existing causal discovery methods in the considered datasets (Appendix E.2). Thus, applying CauTabBench in practice presents challenges not only in terms of scalability but also in reliability. In contrast, TabStruct offers a novel and practical contribution by providing an SCM-free lens through which to assess causal structures in tabular data.

Table 3: **Comparison of considered tabular datasets between TabStruct and prior studies.** TabStruct introduces a novel benchmark designed for the holistic evaluation of tabular generative models, with particular emphasis on evaluating the underlying structure of tabular data. It offers a diverse suite of datasets spanning both classification and regression tasks, thereby supporting comprehensive and structure-aware evaluation across varied use cases.

| Paper | Venue | Structural Fidelity | Classification | | | | | Regression | | | |
| | | | # Datasets | Mixed features | # Sample range | # Feature range | # Class range | # Datasets | Mixed features | # Sample range | # Feature range |
|---|---|---|---|---|---|---|---|---|---|---|---|
| **Model studies** | | | | | | | | | | | |
| CTGAN (Xu et al., 2019) | NeurIPS 2019 | ✗ | 5 | ✓ | 48,842-4,000,000 | 14-54 | 2-7 | 1 | ✓ | 39,644-39,644 | 48-48 |
| TVAE (Xu et al., 2019) | NeurIPS 2019 | ✗ | 5 | ✓ | 48,842-4,000,000 | 14-54 | 2-7 | 1 | ✓ | 39,644-39,644 | 48-48 |
| NFLOW (Durkan et al., 2019) | NeurIPS 2019 | ✗ | 4 | ✓ | 130,065-2,075,259 | 6-43 | 2-2 | ✗ | ✗ | ✗ | ✗ |
| ARF (Watson et al., 2023) | AISTATS 2023 | ✗ | 5 | ✓ | 48,842-4,000,000 | 14-54 | 2-7 | ✗ | ✗ | ✗ | ✗ |
| GOGGLE (Liu et al., 2023) | ICLR 2023 | ✗ | 4 | ✓ | 569-581,012 | 12-168 | 2-7 | ✗ | ✗ | ✗ | ✗ |
| GReaT (Borisov et al., 2023) | ICLR 2023 | ✗ | 5 | ✓ | 954-101,766 | 6-47 | 2-3 | 1 | ✓ | 20,640-20,640 | 8-8 |
| STaSy (Kim et al., 2023) | ICLR 2023 | ✗ | 13 | ✓ | 1,473-284,807 | 9-57 | 2-7 | 2 | ✓ | 39,644-43,824 | 12-48 |
| TabDDPM (Kotelnikov et al., 2023) | ICML 2023 | ✗ | 10 | ✓ | 768-130,064 | 8-50 | 2-4 | 6 | ✓ | 1,338-197,080 | 8-51 |
| CoDi (Lee et al., 2023) | ICML 2023 | ✗ | 12 | ✓ | 1,000-45,211 | 10-28 | 2-7 | 3 | ✓ | 740-1,036 | 6-21 |
| TabSyn (Zhang et al., 2023) | ICLR 2024 | ✗ | 4 | ✓ | 12,330-48,842 | 11-25 | 2-2 | 2 | ✓ | 39,644-43,824 | 12-48 |
| CLLM (Seedat et al., 2024) | ICML 2024 | ✗ | 7 | ✓ | 20-200 | 12-29 | Unknown (private data) | ✗ | ✗ | ✗ | ✗ |
| TabEBM (Margeloiu et al., 2024) | NeurIPS 2024 | ✗ | 8 | ✓ | 20-500 | 7-77 | 2-26 | ✗ | ✗ | ✗ | ✗ |
| NRGBoost (Bravo, 2025) | ICLR 2025 | ✗ | 3 | ✓ | 10,000-116,202 | 12-50 | 2-7 | 3 | ✓ | 835-9,146 | 8-9 |
| TabDiff (Shi et al., 2025) | ICLR 2025 | ✗ | 5 | ✓ | 12,330-101,766 | 11-36 | 2-3 | 2 | ✓ | 39,644-43,824 | 12-48 |
| **Benchmarks** | | | | | | | | | | | |
| Hansen et al. (2023) | NeurIPS 2023 | ✗ | 11 | ✓ | 7608-71,090 | 7-26 | 2-2 | ✗ | ✗ | ✗ | ✗ |
| Synthcity (Qian et al., 2024) | NeurIPS 2023 | ✗ | ✗ | Unknown | ✗ | ✗ | ✗ | ✗ | ✗ | ✗ | ✗ |
| SynMeter (Du & Li, 2024) | arXiv | ✗ | 8 | ✓ | 1,941-48,842 | 11-31 | 2-7 | 4 | ✓ | 1,338-39,644 | 7-60 |
| CauTabBench (Tu et al., 2024) | arXiv | ✗ | ✗ | ✗ | ✗ | ✗ | ✗ | ✗ | ✗ | ✗ | ✗ |
| Livieris et al. (2024) | IFIP 2024 | ✗ | 2 | ✓ | 5,456-20,757 | 18-24 | 3-5 | ✗ | ✗ | ✗ | ✗ |
| SynthEval (Lautrup et al., 2025) | DMKD 2025 | ✗ | 1 | ✓ | 1,385-1,385 | 28-28 | 4 | ✗ | ✗ | ✗ | ✗ |
| Kapar et al. (2025) | arXiv | ✗ | 2 | ✓ | 12,960-48,842 | 8-15 | 2-3 | ✗ | ✗ | ✗ | ✗ |
| **TabStruct (Ours)** | ICLR 2026 | ✓ | **17** | ✓ | **846-100,000** | **6-145** | **2-100** | **12** | ✓ | **345-100,000** | **6-82** |

Table 4: **Comparison of considered tabular generative models between TabStruct and prior studies.** TabStruct encompasses nine distinct categories of tabular generators, enabling a comprehensive and systematic comparison across a broad spectrum of generative approaches.

| Paper | Venue | # Generators | Interpolation | Bayesian Network | Tabular Generative Models | | | | | | |
| --- | --- | --- | --- | --- | --- | --- | --- | --- | --- | --- | --- |
| | | | | | GAN | VAE | Normalising Flows | Tree | Diffusion | EBM | Language Model |
| **Model studies** | | | | | | | | | | | |
| CTGAN (Xu et al., 2019) | NeurIPS 2019 | 7 | ✗ | ✓ | ✓ | ✓ | ✗ | ✗ | ✗ | ✗ | ✗ |
| TVAE (Xu et al., 2019) | NeurIPS 2019 | 7 | ✗ | ✓ | ✓ | ✓ | ✗ | ✗ | ✗ | ✗ | ✗ |
| NFLOW (Durkan et al., 2019) | NeurIPS 2019 | 10 | ✗ | ✗ | ✗ | ✓ | ✓ | ✓ | ✗ | ✗ | ✗ |
| ARF (Watson et al., 2023) | AISTATS 2023 | 6 | ✗ | ✗ | ✓ | ✓ | ✗ | ✗ | ✗ | ✗ | ✗ |
| GOGGLE (Liu et al., 2023) | ICLR 2023 | 7 | ✗ | ✓ | ✓ | ✓ | ✓ | ✗ | ✗ | ✗ | ✓ |
| GReaT (Borisov et al., 2023) | ICLR 2023 | 4 | ✗ | ✗ | ✓ | ✓ | ✗ | ✗ | ✗ | ✗ | ✗ |
| STaSy (Kim et al., 2023) | ICLR 2023 | 8 | ✓ | ✗ | ✓ | ✓ | ✓ | ✗ | ✓ | ✗ | ✗ |
| TabDDPM (Kotelnikov et al., 2023) | ICML 2023 | 6 | ✗ | ✗ | ✓ | ✓ | ✗ | ✗ | ✓ | ✗ | ✓ |
| CoDi (Lee et al., 2023) | ICML 2023 | 9 | ✓ | ✗ | ✓ | ✓ | ✓ | ✗ | ✓ | ✗ | ✗ |
| TabSyn (Zhang et al., 2023) | ICLR 2024 | 9 | ✓ | ✗ | ✓ | ✓ | ✗ | ✗ | ✓ | ✗ | ✓ |
| CLLM (Seedat et al., 2024) | ICML 2024 | 7 | ✓ | ✗ | ✓ | ✓ | ✓ | ✓ | ✓ | ✗ | ✓ |
| TabEBM (Margeloiu et al., 2024) | NeurIPS 2024 | 9 | ✗ | ✗ | ✓ | ✓ | ✓ | ✗ | ✓ | ✓ | ✗ |
| NRGBoost (Bravo, 2025) | ICLR 2025 | 6 | ✗ | ✗ | ✗ | ✗ | ✓ | ✗ | ✗ | ✓ | ✗ |
| TabDiff (Shi et al., 2025) | ICLR 2025 | 9 | ✗ | ✗ | ✓ | ✓ | ✗ | ✗ | ✓ | ✗ | ✓ |
| **Benchmarks** | | | | | | | | | | | |
| Hansen et al. (2023) | NeurIPS 2023 | 5 | ✗ | ✓ | ✓ | ✓ | ✓ | ✗ | ✓ | ✗ | ✗ |
| Synthcity (Qian et al., 2024) | NeurIPS 2023 | 6 | ✗ | ✗ | ✓ | ✓ | ✗ | ✓ | ✓ | ✗ | ✗ |
| SynMeter (Du & Li, 2024) | arXiv | 8 | ✗ | ✗ | ✓ | ✓ | ✗ | ✗ | ✓ | ✗ | ✓ |
| CauTabBench (Tu et al., 2024) | arXiv | 7 | ✗ | ✗ | ✓ | ✓ | ✗ | ✗ | ✓ | ✗ | ✓ |
| Livieris et al. (2024) | IFIP 2024 | 5 | ✓ | ✓ | ✓ | ✗ | ✗ | ✗ | ✗ | ✗ | ✗ |
| SynthEval (Lautrup et al., 2025) | DMKD 2025 | 5 | ✓ | ✓ | ✓ | ✗ | ✗ | ✓ | ✗ | ✗ | ✗ |
| Kapar et al. (2025) | arXiv | 6 | ✗ | ✗ | ✓ | ✓ | ✗ | ✓ | ✓ | ✗ | ✗ |
| **TabStruct (Ours)** | ICLR 2026 | **13** | ✓ | ✓ | ✓ | ✓ | ✓ | ✓ | ✓ | ✓ | ✓ |

# B    DESIGNS OF STRUCTURAL FIDELITY METRICS

In this section, we detail the design and computation of structural fidelity metrics. We first detail the computation of Conditional Independence scores (Appendix B.1), and then detail the computation of the proposed global utility score (Appendix B.2).

## B.1    CONDITIONAL INDEPENDENCE (CI) SCORES

### B.1.1    DERIVING CI STATEMENTS FROM A CAUSAL GRAPH

**Goal.** For each pair of distinct variables $(\boldsymbol{x}_j, \boldsymbol{x}_k)$, our objective is to construct: (i) a family of *d-separation sets* $S_{j,k}$ such that $\boldsymbol{x}_j \perp\!\!\!\perp \boldsymbol{x}_k \mid S_{j,k}$, and (ii) a family of *d-connection sets* $\widehat{S}_{j,k}$ such that $\boldsymbol{x}_j \not\perp\!\!\!\perp \boldsymbol{x}_k \mid \widehat{S}_{j,k}$.

**Notations.** We introduce the following notations, which will be used in the derivation of conditional independence (CI) statements:

- Let $\mathcal{G} := (\mathcal{X}, E)$ denote a directed acyclic graph (DAG), where the node set $\mathcal{X} := \{\boldsymbol{x}_1, \ldots, \boldsymbol{x}_D, \boldsymbol{x}_{D+1}\}$ consists of the variables introduced in Section 3.
- An undirected path $\mathcal{P}$ in $\mathcal{G}$ is a sequence of distinct nodes $\langle v_1, \ldots, v_L \rangle$ such that for each edge on the path, $(v_\ell, v_{\ell+1}) \in E$ or $(v_{\ell+1}, v_\ell) \in E$, and each $v_\ell \in \mathcal{X}$.
- A non-endpoint node $v_\ell$ on $\mathcal{P}$ is a *collider* iff the adjacent edges on $\mathcal{P}$ converge head-to-head at $v_\ell$ (i.e. $\rightarrow v_\ell \leftarrow$ in the induced subpath).
- For disjoint subsets $\{\boldsymbol{x}_j\}, \{\boldsymbol{x}_k\}, S \subseteq \mathcal{X}$, a path $\mathcal{P}$ is said to be *blocked by* $S$ if **either**: (i) $\mathcal{P}$ includes a non-collider that is in $S$, **or** (ii) $\mathcal{P}$ includes a collider such that neither the collider nor any of its descendants is in $S$.
- The variables $\boldsymbol{x}_j$ and $\boldsymbol{x}_k$ are *d-separated* by $S_{j,k}$ (denoted $\boldsymbol{x}_j \perp\!\!\!\perp \boldsymbol{x}_k \mid S_{j,k}$) if every path between $\boldsymbol{x}_j$ and $\boldsymbol{x}_k$ is blocked by $S_{j,k}$.

**Procedures.** The derivations of CI statements are fully programmatic (Spohn, 1980; Dawid, 1979; Constantinou & Dawid, 2017). For each pair of variables $(\boldsymbol{x}_j, \boldsymbol{x}_k)$, we enumerate all subsets $S \subseteq \mathcal{X} \setminus \{\boldsymbol{x}_j, \boldsymbol{x}_k\}$ and apply the d-separation test (Tu et al., 2024; Spirtes et al., 2001) to the triple $(\boldsymbol{x}_j, \boldsymbol{x}_k, S)$. If the test returns true, we add $S$ to the set $S_{j,k}$. Once the d-separation sets are identified, we derive the corresponding d-connection sets by selectively removing elements from the $S_{j,k}$ sets. The full procedure is detailed in Algorithm 1.

### B.1.2    COMPUTE CI SCORES ON TABULAR DATA

We compute CI scores according to Equation (3), where the key step is to select an appropriate conditional independence test for different types of features. For categorical datasets (i.e., all variable are categorical), we employ the chi-square test of independence (McHugh, 2013). For numerical datasets (i.e., all variables are numerical), we use partial correlation based on the Pearson correlation coefficient (Baba et al., 2004). For mixed datasets (i.e., mixed variable types), we utilise a residualisation-based conditional independence test (Ankan & Textor, 2023; Li & Shepherd, 2010; Muller & Peterson, 1984). We implement all conditional independence tests using pgmpy (Ankan & Textor, 2024), an open-source Python library for causal and probabilistic inference. By default, the significance level is set to 0.01 (i.e., the p-value is 0.01).

## B.2    GLOBAL UTILITY SCORE

### B.2.1    DOWNSTREAM PREDICTOR CONFIGURATIONS

To compute the utility per feature as defined in Equation (4), we need to evaluate the performance of downstream predictors when predicting the variable $\boldsymbol{x}_j$, which requires selecting an appropriate set of predictors. As discussed in Section 3, the utility per feature is inherently affected by the inductive biases of downstream models. For instance, KNN tends to perform better when the number of classes is large (Jiang et al., 2024), whereas XGBoost often performs well on skewed target distributions (McElfresh et al., 2024). To mitigate such biases, we employ an ensemble of nine

---

**Algorithm 1** Derive complete CI statements

---

**Input:** DAG $\mathcal{G}$ over nodes $\mathcal{X} = \{\boldsymbol{x}_1, \ldots, \boldsymbol{x}_{D+1}\}$
**Output:** Full CI statements $\mathcal{C}_{\text{global}}$
$\mathcal{C}_{\text{global}} \leftarrow \varnothing$      `// initialise output`
**foreach** *unordered pair* $(j, k) \in \{(a, b) \mid 1 \le a < b \le D + 1\}$ **do**
     $\mathcal{S}_{j,k} \leftarrow \varnothing$      `// reset container`
     **foreach** $S \subseteq \mathcal{X} \setminus \{\boldsymbol{x}_j, \boldsymbol{x}_k\}$ **do**
         **if** *d-separation_test*$(\boldsymbol{x}_j, \boldsymbol{x}_k, S)$ **then**
             $\mathcal{S}_{j,k} \leftarrow \mathcal{S}_{j,k} \cup \{S\}$      `// store separator`
             $\mathcal{C}_{\text{global}} \leftarrow \mathcal{C}_{\text{global}} \cup \{(\boldsymbol{x}_j \perp\!\!\!\perp \boldsymbol{x}_k \mid S)\}$ `// record conditional independence`
         **end**
     **end**
     **foreach** $S \in \mathcal{S}_{j,k}$ **do**
         **foreach** $v \in S$ **do**
             $\widehat{S} \leftarrow S \setminus \{v\}$      `// candidate d-connection set`
             **if *not*** *d-separation_test*$(\boldsymbol{x}_j, \boldsymbol{x}_k, \widehat{S})$ **then**
                 $\mathcal{C}_{\text{global}} \leftarrow \mathcal{C}_{\text{global}} \cup \{(\boldsymbol{x}_j \not\perp\!\!\!\perp \boldsymbol{x}_k \mid \widehat{S})\}$      `// record conditional`
                 `dependence`
             **end**
         **end**
     **end**
**end**
**return** $\mathcal{C}_{global}$      `// complete CI statements`

---

predictors with distinct inductive biases. Specifically, we use the widely adopted "AutoGluon-full" (Erickson et al., 2020), which includes eight predictors, and supplement it with the competitive TabPFN (Hollmann et al., 2025).

Furthermore, as shown in prior work (Kotelnikov et al., 2023), tuning downstream predictors can impact the relative rankings of tabular data generators. To account for this, we allocate a time budget of one hour per feature for tuning the full ensemble. We refer to this configuration (i.e., using all nine tuned predictors) as "Full-tuned".

### B.2.2 PRUNING THE ENSEMBLE OF DOWNSTREAM PREDICTORS

In addition to the "Full-tuned" setup, we define three alternative configurations of downstream predictors. These four configurations are summarised below:

- **Full-tuned**: A *tuned* ensemble of nine predictors: Logistic Regression (LR), KNN, MLP, Random Forest, Extra Trees, LightGBM, CatBoost, XGBoost, TabPFN;

- **Light-tuned**: A *tuned* ensemble of eight predictors: Logistic Regression (LR), MLP, Random Forest, Extra Trees, LightGBM, CatBoost, XGBoost, TabPFN;

- **Tiny-tuned**: A *tuned* ensemble of three predictors: KNN, XGBoost, TabPFN;

- **Tiny-default**: An *untuned* ensemble of three predictors: KNN, XGBoost, TabPFN.

An important observation is that tuning the downstream predictors does improve the absolute performance of the utility per feature. However, we find that *global utility* is more robust to the choice of downstream predictors than *local utility*. Specifically, when the ensemble is reduced from nine to three predictors, the relative rankings of tabular generators under global utility remain consistent, whereas the rankings under local utility fluctuate notably. For instance, under local utility, CTGAN ranks second with "Full-tuned", but drops to 10th with "Tiny-default".

We attribute this robustness to the fairness inherent in the design of global utility – each variable is treated equally as a prediction target, thereby reducing the bias towards any specific decision boundary (i.e., downstream predictor). This design helps to mitigate the effect of predictor-specific biases. Full experimental results are provided in Appendix E.4.

**Practical guidance for computing local and global utility.** For a comprehensive and fair evaluation, TabStruct reports all results under the "Full-tuned" configuration. For local utility, we strongly recommend using the "Full-tuned" configuration. Using a less robust setup may lead to unstable rankings and potentially misleading conclusions about generator performance. In contrast, Appendix E.4 demonstrates that global utility remains consistent even under the "Tiny-default" configuration, as both "Full-tuned" and "Tiny-default" settings produce identical relative rankings across 13 tabular generators. Therefore, we recommend using "Tiny-default" when computing global utility for model selection, particularly in scenarios where computational efficiency is a priority.

## C   RATIONALES FOR EVALUATION FRAMEWORK DESIGN

### C.1   STRUCTURAL PRIOR FOR TABULAR DATA

The underlying structure of tabular data has long been an open research question (Kitson et al., 2023; Hollmann et al., 2025; Müller et al., 2022; Haavelmo, 1944; Wang & Sun, 2022; Ucar et al., 2021; Zhu et al., 2023; Cui et al., 2024; Chen et al., 2023b; Levin et al., 2023). For other modalities like textual data, it is natural to characterise their structure as autoregressive, guided by human knowledge (Yang, 2019). Therefore, pretraining paradigms aligned with the autoregressive structure, such as next-token prediction (Achiam et al., 2023), have proven successful in textual generative modelling. In contrast, heterogeneous tabular data does not naturally lend itself to human interpretation, making a structural prior for such data generally elusive.

Recent studies (Hollmann et al., 2025; Müller et al., 2022) on tabular foundation predictors have begun to shed light on the underlying structure of tabular data. TabPFN (Hollmann et al., 2025) is a tabular foundation predictor pretrained on 100 million "synthetic" tabular datasets. These datasets are "synthetic" because they do not incorporate real-world semantics: they are produced with randomly constructed structural causal models (SCM). Remarkably, despite not being explicitly trained on any real-world dataset, TabPFN is able to outperform an ensemble of strong baseline predictors, which have been fine-tuned on each individual classification task. The exceptional performance of TabPFN suggests that the SCMs used to construct the pretraining datasets, despite lacking real-world semantics, effectively reflect the structural information encoded in real-world tabular data. However, it is important to note that this does not imply SCMs can fully capture the underlying structure of all tabular data, as no definitive theoretical guarantees have been made yet in the tabular domain. Instead, TabPFN demonstrates that the causal relationships between features, as modelled by SCMs, serve as an empirically effective structural prior for a substantial proportion of real-world tabular data.

As the success of LLMs primarily stems from their ability to leverage the autoregressive nature of textual data, we argue that a robust tabular data generation process should be able to capture the unique causal structures within the tabular data. More specifically, generating data aligned with the causal structures in reference data could provide valuable insights into the open research question of how to effectively leverage the structural information inherent in tabular data.

### C.2   CPDAG-LEVEL EVALUATION OF STRUCTURAL FIDELITY

Prior studies (Tu et al., 2024; Spirtes et al., 2001) typically evaluate the causal structure alignment at three different levels: (i) skeleton level, (ii) Markov equivalence class level, and (iii) causal graph level.

**Skeleton level is limited in capacity.** At the skeleton level, all causal directions are ignored, resulting in a loss of information about the causal relationships between features. For instance, the causal skeleton is unable to reflect encoded physical laws. Consider the physical system illustrated in Figure 1: the ground-truth causal path from $\rho$ to $F_{\text{Earth}}$ is $\rho \rightarrow m_{\text{A}} \rightarrow F_{\text{Earth}}$. This encodes a meaningful interpretation of physical law: given $m_{\text{A}}$, changing $\rho$ should *not* affect the gravitational force acting on ball A. However, if all directions are removed from the causal path, the resulting skeleton allows for alternative paths, such as $\rho \rightarrow m_{\text{A}} \leftarrow F_{\text{Earth}}$, which share the same undirected structure but imply contradictory physical laws. In this case, the alternative path suggests that, given $m_{\text{A}}$, changing $\rho$ *would* affect the gravitational force, which is incorrect. Therefore, we choose not to evaluate structural fidelity at the skeleton level due to its inability to capture reliable causal relationships across variables.

**Causal graph level necessitates efficient and accurate causal discovery methods, which remains an open research question.** At the causal graph level, structural fidelity is assessed by comparing the directed acyclic graphs (DAGs) of the reference and synthetic datasets, accounting for both the skeleton and the causal directions of edges. In principle, this level provides the most fine-grained evaluation of structural fidelity. However, current causal discovery methods struggle to recover accurate DAGs from observational tabular data (Nastl & Hardt, 2024). Section 4 and Appendix F demonstrate such limitations – where Bayesian Network (BN) performs poorly in generating high-quality synthetic data – suggesting that existing causal discovery tools are inadequate for learning precise causal graphs.

This limitation is well-documented in the literature: recovering perfect DAGs from tabular data remains an unresolved problem for current algorithms (Zanga et al., 2022; Kaddour et al., 2022; Nastl & Hardt, 2024). This limitation further supports our argument that CauTabBench provides limited insights for real-world datasets. While CauTabBench attempts to evaluate structural fidelity by applying causal discovery methods to infer a "pseudo" causal graph from real-world data, the absence of ground-truth (GT) causal structures makes such evaluations unreliable. Without access to a known GT, it is impossible to assess the validity of the inferred graphs. Moreover, the poor empirical performance of BN suggests that these pseudo causal graphs may not be accurate.

Moreover, evaluating at the DAG level requires running causal discovery algorithms on both the reference and synthetic datasets. Employing a specific causal discovery algorithm may introduce evaluation bias – analogous to how utility scores are affected by the choice of predictor models. To reduce this bias, one would need to ensemble multiple causal discovery methods. However, unlike downstream predictors, causal discovery algorithms are often computationally expensive. For instance, the DAGMA algorithm (Bello et al., 2022) takes over 24 hours to recover a causal graph from a dataset with more than 100 features on our machine (Intel(R) Xeon(R) CPU @ 2.20GHz, 64 cores), due to the exponential scaling of its computation cost with dimensionality.

**CPDAG-level evaluation strikes a good balance between evaluation efficiency and validity.** Unlike full DAG constructing via causal discovery, CPDAG-level evaluation does not require the orientation of all edges, making it a more tractable yet still meaningful metric of structural fidelity. A CPDAG represents the Markov equivalence class of a DAG, preserving essential causal relationships while greatly reducing computational overhead. This is supported by the fact that Markov equivalent SCMs serve as minimal I-MAPs (Agrawal et al., 2018) of the joint distribution factorisation $p(\mathcal{X}) = \prod_{j=1}^{D+1} p(\boldsymbol{x}_j \mid \mathrm{pa}(\boldsymbol{x}_j))$, and no causal directions can be further removed. Therefore, the CPDAG-level evaluation provides a lens to interpret the fidelity of the tabular data. As illustrated in Figure 1, CPDAGs retain sufficient real-world semantics for practical use cases. Therefore, TabStruct evaluates structural fidelity at the CPDAG level, balancing semantic richness with computational feasibility.

It is important to note that even reference datasets do not guarantee CI scores of 1. This is analogous to ML efficacy, where even reference data cannot ensure perfect downstream utility (e.g., balanced accuracy = 1 or RMSE = 0). However, as shown in Section 4 and Appendix F, conditional independence (CI) tests generally provide valid and reliable evaluation results. Specifically, CI tests yield consistently high scores on reference datasets, indicating their ability to distinguish between high- and low-quality datasets and thus produce meaningful fidelity assessments.

# D    REPRODUCIBILITY

## D.1    BENCHMARK DATASETS

### D.1.1    SCM DATASETS

To accurately quantify structural fidelity, the reference data should be paired with ground-truth causal structures. To this end, we construct benchmark SCM datasets using structural causal models (SCMs) that have been validated by human experts (Scutari, 2011). All 11 SCM datasets are publicly available, with further details provided in Table 5, Table 6, and Table 7. By default, throughout this work, references to "six SCM datasets" refer to those listed in Table 5 and Table 6.

Table 5: Details of three SCM classification datasets from bnlearn (Scutari, 2011).

| Dataset | Domain | # Samples ($N$) | # Features ($D$) | $N/D$ | # Numerical | # Categorical | # Classes | # Samples per class (Min) | # Samples per class (Max) |
|---|---|---|---|---|---|---|---|---|---|
| Hailfinder | Meteorology | 100,000 | 56 | 1785.71 | 0 | 56 | 3 | 25,048 | 44,200 |
| Insurance | Economics | 100,000 | 27 | 3703.70 | 0 | 27 | 4 | 1,648 | 56,361 |
| Sangiovese | Agriculture | 100,000 | 15 | 6666.67 | 14 | 1 | 16 | 5,659 | 6,841 |

Table 6: Details of three SCM regression datasets from bnlearn (Scutari, 2011).

| Dataset | Domain | # Samples ($N$) | # Features ($D$) | $N/D$ | # Numerical | # Categorical |
|---|---|---|---|---|---|---|
| Healthcare | Medicine | 100,000 | 7 | 14285.71 | 4 | 3 |
| MAGIC-IRRI | Life Science | 100,000 | 64 | 1562.50 | 64 | 0 |
| MEHRA | Meteorology | 100,000 | 24 | 4166.67 | 20 | 4 |

Table 7: Details of five classification datasets with large SCMs from bnlearn (Scutari, 2011).

| Dataset | Domain | # Samples ($N$) | # Features ($D$) | $N/D$ | # Numerical | # Categorical | # Classes | # Samples per class (Min) | # Samples per class (Max) |
|---|---|---|---|---|---|---|---|---|---|
| ANDES | Education | 100,000 | 223 | 448.43 | 0 | 223 | 2 | 23,000 | 77,000 |
| Diabetes | Life Science | 100,000 | 413 | 242.13 | 0 | 413 | 4 | 1,000 | 86,000 |
| Link | Life Science | 100,000 | 724 | 138.12 | 0 | 724 | 4 | 24,961 | 25.037 |
| Pathfinder | Medicine | 100,000 | 109 | 917.43 | 0 | 109 | 4 | 8,000 | 68,000 |
| PIGS | Life Science | 100,000 | 441 | 226.76 | 0 | 441 | 3 | 24,769 | 49,988 |

Human validation ensures that the causal structures are realistic, thereby increasing the likelihood that TabStruct's benchmark results can generalise to other real-world datasets where ground-truth SCMs are not available. We note that this is a core difference between TabStruct and prior studies (Tu et al., 2024; Hollmann et al., 2025): rather than relying on toy SCM datasets lacking real-world semantics, TabStruct introduces one of the first comprehensive benchmarks for tabular generative models, based on datasets with expert-validated causal structures, mixed feature types, and more than 10 features.

We outline the process of building the reference SCM datasets as follows. Firstly, we use ground-truth SCMs with realistic and expert-validated structures. Secondly, we perform prior sampling on these SCMs: root nodes are randomly initialised, and their values are propagated through the causal graph. A single sample is generated by recording the node values after propagation, with each propagation producing one sample. Thirdly, this process is repeated until sufficient samples are obtained. In TabStruct, we set $N_{\text{full}} =$ 100,000. By following this procedure, we construct full datasets $\mathcal{D}_{\text{full}}$ with accessible and well-defined causal structures. The pseudocode is in Algorithm 2.

### D.1.2    REAL-WORLD DATASETS

To demonstrate the generalisability of the proposed global utility and TabStruct, we further select 23 challenging real-world datasets from the open-source TabZilla benchmark (McElfresh et al., 2024), the OpenML repository (https://www.openml.org/search?type=data&sort=runs), and the UCI repository (https://archive.ics.uci.edu/datasets). All datasets are publicly available, with further details provided in Table 8 and Table 9.

---

**Algorithm 2** Constructing full SCM datasets

---

**Input:** Ground-truth structural causal model, $M = \langle \mathcal{X}, \mathcal{G}, \mathcal{F}, \mathcal{E} \rangle$, number of samples $N_{\text{full}}$ (default to 100,000 samples)

**Output:** Full SCM dataset $\mathcal{D}_{\text{full}} = \left\{ (\mathbf{x}^{(i)}, y^{(i)}) \right\}_{i=1}^{N_{\text{full}}}$

**Pre-processing** $\pi \leftarrow \texttt{TopologicalSort}(\mathcal{G})$      `// topological order of the variables`

$\mathcal{D}_{\text{full}} \leftarrow \texttt{InitDataset}()$     `// Initialise an empty dataset`

**for** $i \leftarrow 1$ **to** $N_{\text{full}}$ **do**
  **for** $j \in \pi$ **do**
    **if** $\text{pa}(\boldsymbol{x}_j) = \varnothing$ **then**
      $x_j^{(i)} \leftarrow \texttt{Sample}(\boldsymbol{\epsilon}_j)$     `// root node: random initialisation`
    **else**
      $x_j^{(i)} \leftarrow f_j\big( \{ x_k^{(i)} : \boldsymbol{x}_k \in \text{pa}(\boldsymbol{x}_j) \}, \boldsymbol{\epsilon}_j \big)$     `// Propagate through SCM`
    **end**
  **end**
  $\texttt{Append}\big( \mathcal{D}_{\text{full}}, (x_1^{(i)}, \ldots, x_{D+1}^{(i)}) \big)$     `// Add the new sample to the SCM dataset`
**end**
**return** $\mathcal{D}_{\text{full}}$

---

The dataset selection follows three main criteria: Firstly, the datasets are non-trivial, meaning that generative models cannot easily achieve evaluation results comparable to those obtained from the reference data. Secondly, the datasets originate from diverse domains. For example, "Credit-g" pertains to business applications, whereas "Plants" relates to biological studies. Thirdly, the datasets were not part of the meta-validation stage for TabPFN, reducing the likelihood that their causal structures were implicitly leaked during the development or pretraining of TabPFN.

Table 8: Details of 14 real-world classification datasets.

| Dataset | Domain | Source | ID | # Samples $(N)$ | # Features $(D)$ | $N/D$ | # Numerical | # Categorical | # Classes | # Samples per class (Min) | # Samples per class (Max) |
|---|---|---|---|---|---|---|---|---|---|---|---|
| Ada | Economics | OpenML | 1043 | 4,562 | 48 | 95.04 | 47 | 1 | 2 | 1,132 | 3,430 |
| Characters | Images | OpenML | 1459 | 10,218 | 8 | 1277.25 | 7 | 1 | 10 | 600 | 1,416 |
| Credit-g | Economics | OpenML | 46378 | 1,000 | 21 | 47.62 | 7 | 14 | 2 | 300 | 700 |
| Electricity | Economics | OpenML | 151 | 45,312 | 9 | 5034.67 | 7 | 2 | 2 | 19,237 | 26,075 |
| Higgs | Physics | OpenML | 4532 | 98,050 | 29 | 3381.03 | 28 | 1 | 2 | 46,223 | 51,827 |
| Jasmine | Life Science | OpenML | 41143 | 2,984 | 145 | 20.58 | 8 | 137 | 2 | 1,492 | 1,492 |
| Nomao | Economics | OpenML | 45078 | 34,465 | 119 | 289.62 | 89 | 30 | 2 | 9,844 | 24,621 |
| Phoneme | Language | OpenML | 1489 | 5,404 | 6 | 900.67 | 5 | 1 | 2 | 1,586 | 3,818 |
| Plants | Life Science | OpenML | 1493 | 1,599 | 65 | 24.60 | 64 | 1 | 100 | 15 | 16 |
| QSAR | Chemistry | OpenML | 1494 | 1,055 | 42 | 25.12 | 41 | 1 | 2 | 356 | 699 |
| SpeedDating | Social Science | OpenML | 40536 | 8,378 | 121 | 69.24 | 59 | 62 | 2 | 1,380 | 6,998 |
| Splice | Life Science | OpenML | 46 | 3,190 | 61 | 52.30 | 0 | 61 | 3 | 767 | 1,655 |
| Vehicle | Transportation | OpenML | 54 | 846 | 19 | 44.53 | 18 | 1 | 4 | 199 | 218 |
| Zernike | Images | OpenML | 22 | 2,000 | 48 | 41.67 | 47 | 1 | 10 | 200 | 200 |

Table 9: Details of nine real-world regression datasets.

| Dataset | Domain | Source | ID | # Samples $(N)$ | # Features $(D)$ | $N/D$ | # Numerical | # Categorical |
|---|---|---|---|---|---|---|---|---|
| Ailerons | Physics | OpenML | 296 | 13,750 | 41 | 335.37 | 41 | 0 |
| California | Economics | OpenML | 43939 | 20,640 | 10 | 2064.00 | 9 | 1 |
| Elevators | Physics | OpenML | 216 | 16,599 | 19 | 873.63 | 19 | 0 |
| H16 | Economics | OpenML | 574 | 22,784 | 17 | 1340.24 | 17 | 0 |
| Liver | Medicine | OpenML | 8 | 345 | 6 | 57.50 | 6 | 0 |
| Sales | Economics | OpenML | 42092 | 21,613 | 20 | 1080.65 | 18 | 2 |
| Space | Demographics | OpenML | 507 | 3,107 | 7 | 443.86 | 7 | 0 |
| Superconductivity | Chemistry | UCI | 464 | 21,263 | 82 | 259.30 | 82 | 0 |
| Wine | Life Scienece | UCI | 186 | 6,497 | 12 | 541.42 | 12 | 0 |

## D.2 DATA PROCESSING

**Data splitting (Figure 5).** For each dataset of $N$ samples, we first split it into train and test sets (80% train and 20% test). We further split the train set into a training split ($\mathcal{D}_{\mathrm{ref}}$) and a validation split (90% training and 10% validation). For classification datasets, stratification is preserved during data splitting. We repeat the splitting 10 times, summing up to 10 runs per dataset. All benchmark generators are trained on $\mathcal{D}_{\mathrm{ref}}$, and each generator produces a synthetic dataset with $N_{\mathrm{ref}}$ samples. For classification, the synthetic data preserves the stratification of the reference data.

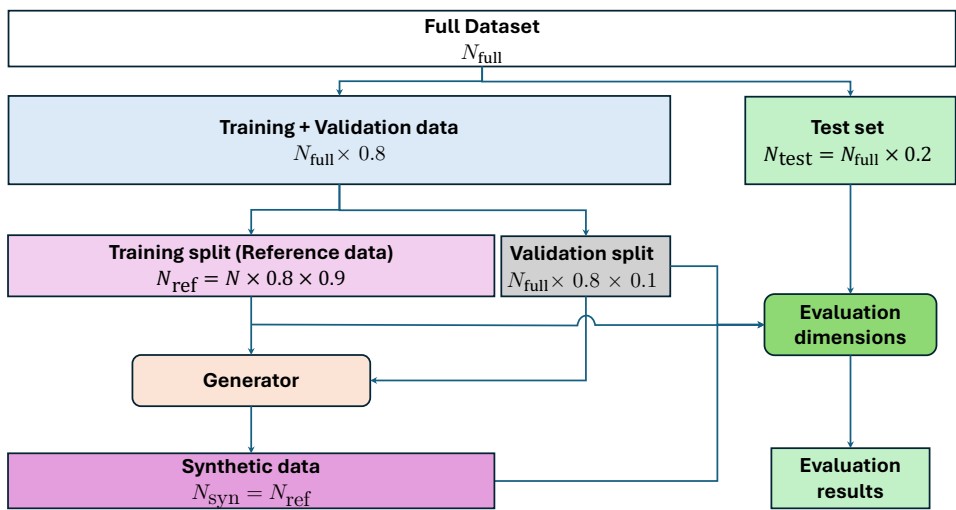

Figure 5: Data splitting strategies for benchmarking tabular data generators.

**Feature preprocessing for generators.** Following the procedures presented in prior work (McElfresh et al., 2024; Grinsztajn et al., 2022), we perform preprocessing in three steps. Firstly, we impute the missing values with the mean value for numerical features and the mode value for categorical features. We then compute the required statistics with training data and then transform the training split. For categorical features, we convert them into one-hot encodings. An exception is TabDiff, which tends to perform better with ordinal encoding for categorical features. For numerical features, we perform Z-score normalisation. We compute each feature's mean and standard deviation in the training data and then transform the training samples to have a mean of zero and a variance of one for each feature. Finally, we apply the same transformation to the validation and test data before conducting evaluations.

**Feature preprocessing for downstream predictors.** The synthetic data produced by generators is inversely transformed back to the original feature space before being passed to the downstream predictors. In other words, the AutoGluon models receive input data in the original, unprocessed feature space, allowing them to apply their own model-specific preprocessing strategies.

## D.3 IMPLEMENTATIONS OF BENCHMARK GENERATORS

TabStruct includes 13 existing tabular data generation methods of nine different categories: (i) a standard interpolation method SMOTE (Chawla et al., 2002); (ii) a structure learning method Bayesian Network (BN) (Qian et al., 2024); (iii) two Variational Autoencoders (VAE) based methods TVAE (Xu et al., 2019) and GOGGLE (Liu et al., 2023); (iv) a Generative Adversarial Networks (GAN) method CTGAN (Xu et al., 2019); (v) a normalising flow model Neural Spine Flows (NFLOW) (Durkan et al., 2019); (vi) a tree-based method Adversarial Random Forests (ARF) (Watson et al., 2023); (vii) three diffusion models: TabDDPM (Kotelnikov et al., 2023), TabSyn (Zhang et al., 2023), TabDiff (Shi et al., 2025); (viii) two energy-based models: TabEBM (Margeloiu et al., 2024) and NRGBoost (Bravo, 2025); and (ix) a Large Language Model (LLM) based method GReaT (Borisov et al., 2023).

Following prior work (Kotelnikov et al., 2023; Hansen et al., 2023), we tune the parametrised generators to ensure a fair comparison. Specifically, we use Optuna (Akiba et al., 2019) to optimise each generator by minimising its average validation loss across 10 repeated runs. Each generator

is given at most two hours to complete a single repeat. Importantly, to mitigate bias introduced by specific evaluation metrics, we tune each generator based on its own objective function rather than any external metric. Different from prior work (Du & Li, 2024), this approach ensures that each model is evaluated under conditions aligned with its intended optimisation direction. The technical details and hyperparameter search space for each method are described below.

**SMOTE** is an interpolation-based oversampling technique (Chawla et al., 2002), which generates synthetic samples by interpolating between existing minority class instances. We employ the open-source implementation of SMOTE provided by Imbalanced-learn (Lemaître et al., 2017), where the number of nearest neighbours $k$ can be specified. Unless stated otherwise, we use the default setting of $k = 5$.

**Bayesian Network (BN)** is a probabilistic graphical model used to represent and reason about the dependence relationships between features (Qian et al., 2024; Hansen et al., 2023). It consists of two main components: (i) a causal discovery model to construct a directed acyclic graph (DAG), where features and the target serve as nodes, and their dependencies are represented as edges; (ii) a parameter estimation mechanism to quantify the dependence relationships. Following prior work (Hansen et al., 2023), the causal discovery method is selected from Hill Climbing Search (Koller, 2009), the Peter-Clark algorithm (Koller, 2009; Spirtes et al., 2001), LiNGAM (Shimizu, 2014), LiM (Zeng et al., 2022), DAGMA (Bello et al., 2022), DCD (Prashant et al., 2024), AutoCD (Chan et al., 2024), and Chow-Liu or Tree-augmented Naive Bayes (Chow & Liu, 1968; Friedman et al., 1997). We empirically find that AutoCD generally achieves the highest structural fidelity, and thus we build a parametrised BN with AutoCD and maximum likelihood estimation.

Table 10: Hyperparameter search space of BN.

| Hyperparameter | Range |
|---|---|
| struct_learning_score | {"k2", "bdeu", "bic", "bds"} |

**TVAE** is a variational autoencoder (VAE) designed for tabular data (Xu et al., 2019). TVAE employs mode-specific normalisation to handle the complex distributions of numerical features. To address the class imbalance problem, TVAE conditions on specific categorical features during generation.

Table 11: Hyperparameter search space of TVAE.

| Hyperparameter | Range |
|---|---|
| encoder_n_layers_hidden | $[1, 5]$ |
| encoder_n_units_hidden | $[50, 500]$ |
| encoder_nonlin | $\{\text{relu}, \text{leaky\_relu}, \text{tanh}, \text{elu}\}$ |
| n_units_embedding | $[50, 500]$ |
| decoder_n_layers_hidden | $[1, 5]$ |
| decoder_n_units_hidden | $[50, 500]$ |
| decoder_nonlin | $\{\text{relu}, \text{leaky\_relu}, \text{tanh}, \text{elu}\}$ |
| n_iter | $[100, 1000]$ |
| lr | $[10^{-4}, 10^{-3}]$ (log) |
| weight_decay | $[10^{-4}, 10^{-3}]$ (log) |

**GOGGLE** is a VAE-based tabular data generator designed to model the dependence relationships between features (Liu et al., 2023). GOGGLE proposes to learn an adjacency matrix to model the dependence relationships between features. However, TabStruct and prior benchmarks (Margeloiu et al., 2024; Zhang et al., 2023; Shi et al., 2025) all show that the downstream utility of GOGGLE is limited. We hypothesise that this stems from the challenge of learning accurate structures of tabular data. The inherent structure learning mechanism in GOGGLE fails to capture accurate conditional independence relationships between features, which could thus lead to low-quality synthetic data. We further observe that the official implementation of GOGGLE (Qian et al., 2024) can be unstable and may fail to converge when fitted on large tabular datasets (e.g., more than 10,000 samples).

Table 12: Hyperparameter search space of GOGGLE.

| Hyperparameter | Range |
|---|---|
| encoder_dim | $[32, 128]$ |
| encoder_l | $[1, 5]$ |
| decoder_dim | $[32, 128]$ |
| decoder_arch | $\{gcn, het, sage\}$ |
| n_iter | $[100, 500]$ |
| learning_rate | $[10^{-4}, 5 \times 10^{-3}]$ (log) |
| weight_decay | $[10^{-4}, 10^{-3}]$ (log) |
| alpha | $[0.0, 1.0]$ |
| beta | $[0.0, 1.0]$ |
| iter_opt | $\{True, False\}$ |
| threshold | $[0.0, 1.0]$ |

**CTGAN** is a conditional generative adversarial network (GAN) designed for tabular data (Xu et al., 2019). CTGAN leverages PacGAN (Lin et al., 2018) framework to mitigate mode collapse. In addition, CTGAN employs the same mode-specific normalisation technique as TVAE.

Table 13: Hyperparameter search space of CTGAN.

| Hyperparameter | Range |
|---|---|
| generator_n_layers_hidden | $[1, 4]$ |
| generator_n_units_hidden | $[50, 150]$ |
| generator_nonlin | $\{relu, leaky\_relu, tanh, elu\}$ |
| discriminator_n_layers_hidden | $[1, 4]$ |
| discriminator_n_units_hidden | $[50, 150]$ |
| discriminator_nonlin | $\{relu, leaky\_relu, tanh, elu\}$ |
| n_iter | $[100, 1000]$ |
| discriminator_n_iter | $[1, 5]$ |
| lr | $[10^{-4}, 10^{-3}]$ (log) |
| weight_decay | $[10^{-4}, 10^{-3}]$ (log) |

**NFlow** is a normalisation flow model designed for tabular data generation (Durkan et al., 2019). NFlow incorporates neural splines as a drop-in replacement for affine or additive transformations in coupling and autoregressive layers, which assists in the modelling of tabular data.

Table 14: Hyperparameter search space of NFlow.

| Hyperparameter | Range |
|---|---|
| n_layers_hidden | $[1, 10]$ |
| n_units_hidden | $[10, 100]$ |
| linear_transform_type | $\{lu, permutation, svd\}$ |
| base_transform_type | $\{affine\text{-}coupling,$ quadratic-coupling, rq-coupling, affine-autoregressive, quadratic-autoregressive, rq-autoregressive$\}$ |
| dropout | $[0.0, 0.2]$ |
| batch_norm | $\{False, True\}$ |
| lr | $[2 \times 10^{-4}, 10^{-3}]$ (log) |
| n_iter | $[100, 5000]$ |

**ARF** is a tree-based model for tabular data generation (Watson et al., 2023). ARF employs a recursive adaptation of unsupervised random forests for joint density estimation by iteratively refining synthetic data distributions using adversarial training principles.

Table 15: Hyperparameter search space of ARF.

| Hyperparameter | Range |
|---|---|
| num_trees | $\{10, 20, \ldots, 100\}$ |
| delta | $\{0, 2, \ldots, 50\}$ |
| max_iters | $[1, 5]$ |
| early_stop | $\{\text{True}, \text{False}\}$ |
| min_node_size | $\{2, 4, \ldots, 20\}$ |

**TabDDPM** is a diffusion-based model for tabular data generation (Kotelnikov et al., 2023). TabDDPM introduces two core diffusion processes: (i) Gaussian noise for numerical features and (ii) multinomial diffusion with categorical noise for categorical features. TabDDPM directly concatenates numerical and categorical features as the input and output of the denoising function. We further note that the official implementation of TabDDPM (research, 2023) can be unstable and may fail to converge when fitted on tabular data with mixed feature types.

Table 16: Hyperparameter search space of TabDDPM.

| Hyperparameter | Range |
|---|---|
| n_iter | $[10^3, 10^4]$ |
| lr | $[10^{-5}, 10^{-1}]$ (log) |
| weight_decay | $[10^{-4}, 10^{-3}]$ (log) |
| num_timesteps | $[10, 10^3]$ |

**TabSyn** is a diffusion-based model for tabular data generation (Zhang et al., 2023). It synthesises tabular data by employing a diffusion model within the latent space of a variational autoencoder (VAE). TabSyn supports a wide range of data types by mapping them into a unified representation space and explicitly modelling inter-column dependencies.

Table 17: Hyperparameter search space of TabSyn.

| Hyperparameter | Range |
|---|---|
| vae.num_epochs | $[100, 1000]$ |
| vae.max_beta | $[10^{-3}, 10^{-2}]$ (log) |
| vae.min_beta | $[10^{-5}, 10^{-4}]$ (log) |
| vae.lambd | $[0.1, 1.0]$ |
| vae.num_layers | $[1, 4]$ |
| vae.d_token | $[1, 8]$ |
| vae.n_head | $[1, 4]$ |
| vae.factor | $[1, 64]$ |
| vae.lr | $[10^{-4}, 10^{-2}]$ (log) |
| vae.wd | $[0, 10^{-2}]$ (log) |
| tabsyn.num_epochs | $[100, 500]$ |
| tabsyn.lr | $[10^{-4}, 10^{-2}]$ (log) |
| tabsyn.wd | $[0, 10^{-2}]$ (log) |

**TabDiff** is a diffusion-based model for tabular data generation (Shi et al., 2025). It introduces a joint diffusion framework capable of capturing the mixed-type distributions inherent in tabular data within a single model. In particular, TabDiff utilises a joint continuous-time diffusion process and leverages a transformer architecture to handle both numerical and categorical variables.

Table 18: Hyperparameter search space of TabDiff.

| Hyperparameter | Range |
|---|---|
| batch_size | $\{512, 1024, 2048, 4096, 8192\}$ |
| c_lambda | $[0.1, 10.0]$ |
| check_val_every | $\{10, 20, 30, 40, 50\}$ |
| closs_weight_schedule | $\{$"constant", "anneal", "linear"$\}$ |
| d_lambda | $[0.1, 10.0]$ |
| ema_decay | $[0.9, 0.9999]$ |
| factor | $[0.1, 0.99]$ |
| lr | $[10^{-5}, 10^{-2}]$ (log) |
| lr_scheduler | $\{$"reduce_lr_on_plateau", "cosine", "none"$\}$ |
| reduce_lr_patience | $\{10, 30, 50, 70\}$ |
| steps | $\{100, 200, 300, 500\}$ |
| weight_decay | $[0, 10^{-2}]$ (log) |

**TabEBM** is an energy-based model for tabular data generation (Margeloiu et al., 2024). It transforms a pretrained tabular predictor into a set of class-specific generators. While the original paper only provides TabEBM implementation for classification tasks, we extend its applicability in TabStruct to regression tasks by treating all reference samples as a single class, and then performing sampling over the energy surface.

Table 19: Hyperparameter search space of TabEBM.

| Hyperparameter | Range |
|---|---|
| starting_point_noise_std | $[10^{-4}, 10^{-1}]$ (log) |
| sgld_step_size | $[10^{-3}, 10^{-1}]$ (log) |
| sgld_noise_std | $[10^{-4}, 10^{-1}]$ (log) |
| sgld_steps | $\{50, 100, 200, 500\}$ |

**NRGBoost** is an energy-based model for tabular data generation (Bravo, 2025). It is trained by maximising a local second-order approximation to the log-likelihood at each stage of the boosting process. NRGBoost is shown to offer generally good discriminative performance and competitive sampling performance compared to more specialised alternatives.

Table 20: Hyperparameter search space of NRGBoost.

| Hyperparameter | Range |
|---|---|
| num_trees | $\{1, 5, 10, 20, 50\}$ |
| shrinkage | $[0.01, 0.3]$ |
| max_leaves | $\{32, 64, 128, 256, 512\}$ |
| max_ratio_in_leaf | $[1, 5]$ |
| num_model_samples | $\{10{,}000, 40{,}000, 80{,}000, 160{,}000\}$ |
| p_refresh | $[0.01, 0.3]$ |
| num_chains | $\{4, 8, 16, 32\}$ |
| burn_in | $\{50, 100, 200, 500\}$ |

**GReaT** leverages large language models (LLMs) to generate synthetic tabular data (Borisov et al., 2023). GReaT converts each sample into a sentence and fine-tunes the language model to capture the sentence-level distributions. Additionally, GReaT shuffles the order of features to mitigate the permutation variance in sentence-level distributions.

Table 21: Hyperparameter search space of GReaT.

| Hyperparameter | Range |
|----------------|-------|
| n_iter | $\{100,\ 300,\ 500,\ 1000\}$ |
| learning_rate | $[10^{-4},\ 10^{-2}]$ (log) |
| weight_decay | $[10^{-5},\ 10^{-2}]$ (log) |

### D.4 HYPERPARAMETER TUNING FOR DOWNSTREAM PREDICTORS

As discussed in Appendix B.2, we employ AutoGluon's built-in tuning functionality for training the ensemble predictors. For each variable, the ensemble predictor is allocated one hour of tuning budget per repeat, resulting in a total of 10 hours per variable for each dataset. We note that TabPFN is not integrated into the employed version of AutoGluon. However, the default configuration of TabPFN already demonstrates competitive performance (Hollmann et al., 2025), and thus, we use its default hyperparameters across all of our experiments.

### D.5 AGGREGATION OF EVALUATION RESULTS

The reported results are averaged by default over 10 repeats. We aggregate results across all SCM or real-world datasets because the findings are consistent across classification and regression tasks. Specifically, we use the average distance to the minimum (ADTM) metric (Grinsztajn et al., 2022; McElfresh et al., 2024; Hollmann et al., 2025; Margeloiu et al., 2024; Jiang et al., 2024) via affine renormalisation between the top-performing and worse-performing models.

### D.6 SOFTWARE AND COMPUTING RESOURCES

**Software implementation.** *(i) For generators:* We implemented SMOTE with Imbalanced-learn (Lemaître et al., 2017), an open-source Python library for imbalanced datasets with an MIT license. For TabSyn and TabEBM, we used their open-source implementations with an Apache-2.0 license. For TabDiff and NRGBoost, we used their open-source implementations with an MIT license. For other benchmark generators, we used their open-source implementations in Synthcity (Qian et al., 2024), a library for generating and evaluating synthetic tabular data with an Apache-2.0 license. *(ii) For downstream predictors:* We implemented TabPFN with its open-source implementation (`https://github.com/automl/TabPFN`). We implemented the other eight downstream predictors (i.e., Logistic Regression, KNN, MLP, Random Forest, Extra Trees, LightGBM, CatBoost, and XGBoost) with their open-source implementation in scikit-learn (Pedregosa et al., 2011) and AutoGluon (Erickson et al., 2020), an open-source Python library under an Apache-2.0 license. *(iii) For data preprocessing and evaluation:* All data handling, including data loading and preprocessing, was performed with TabCamel (Jiang, 2025a), an open-source Python library for tabular data management. All data-quality evaluation was performed with TabEval (Jiang, 2025b) and Synthcity (Qian et al., 2024), a comprehensive open-source Python framework for evaluating tabular data. *(iv) For result analysis and visualisation:* All numerical plots and graphics have been generated using Matplotlib 3.7 (Hunter, 2007), a Python-based plotting library with a BSD license. The icons for evaluation dimensions in Figure 2 are from `https://icons8.com/`.

We ensure the consistency and reproducibility of experimental results by implementing a uniform pipeline using PyTorch Lightning, an open-source library under an Apache-2.0 license. We further fixed the random seeds for data loading and evaluation throughout the training and evaluation process. This ensured that all benchmark models in TabStruct were trained and evaluated on the same set of samples. The experimental environment settings, including library dependencies, are specified in the open-source library for reference and reproduction purposes.

**Computing Resources.** All the experiments were conducted on a machine equipped with an NVIDIA A100 GPU with 80GB memory and an Intel(R) Xeon(R) CPU (at 2.20GHz) with 64 cores. The operating system used was Ubuntu 20.04.5 LTS.

# E    EXTENDED ANALYSIS AND DISCUSSION

## E.1    EXTENDED ANALYSIS ON VALIDITY OF BENCHMARK FRAMEWORK

**The benchmark datasets present a genuine challenge for existing generators.** As detailed in Section 4, we select challenging, contamination-free real-world datasets, ensuring that they are non-trivial for existing tabular data generators. Table 2 illustrates that, unlike prior studies (Shi et al., 2025; Zhang et al., 2023; Margeloiu et al., 2024), no generator can easily match $\mathcal{D}_{\text{ref}}$ on our benchmark datasets. This confirms that the selected datasets offer a more informative and realistic assessment of generator capabilities.

**Detection score (C2ST) is relatively limited in measuring global structural fidelity.** Following prior work Zhang et al. (2023), we compute the detection score (C2ST) using logistic regression classifier. Higher C2ST scores indicate better performance, i.e., synthetic data that is harder to distinguish from real data. We select three SCM classification datasets (Table 5): Hailfinder, Insurance, and Sangiovese. Table 22 shows that C2ST exhibits a weaker correlation with global CI compared to global utility. The relatively low correlation between C2ST and global CI is consistent with the trends observed in other sample-level metrics, including $\alpha$-precision and $\beta$-recall. Although these sample-level metrics are designed to capture high-order interactions across features, they fail to explicitly attend to inter-feature causal interactions, limiting their ability to reflect the underlying causal structures. This further supports the effectiveness of global utility in assessing global structural fidelity of tabular data.

Table 22: **Spearman's rank correlation with global CI on three SCM datasets.** We **bold** the highest correlation with Global CI. Global utility correlates strongly with global CI (p < 0.001), demonstrating the validity of global utility.

|                        | Global CI ↑ |
| ---------------------- | ----------- |
| $\alpha$-precision     | 0.38        |
| $\beta$-recall         | 0.47        |
| C2ST                   | 0.50        |
| **Global utility (Ours)** | **0.83**  |

## E.2    EXTENDED ANALYSIS ON VALIDITY OF GLOBAL UTILITY

**The evaluation results are consistent across classification and regression datasets of different domains.** In Table 23, we present per-dataset evaluation results for both local and global utility. SMOTE remains one of the most competitive methods for capturing local structure, and diffusion models consistently rank among the top-3 for modelling global data structure. These findings indicate that the proposed "utility per variable" metric is stable and provides a unified lens for interpreting evaluation results across both classification and regression datasets.

**Global utility provides similar generator rankings as global CI.** Figure 6 demonstrates that the rankings of generators under global utility closely align with those under global CI. Notably, the Top-3 methods are identical across both metrics: TabSyn, TabDDPM, and TabDiff. In contrast, when using local utility, the Top-3 methods shift to SMOTE, CTGAN, and TabDiff. This reveals a great discrepancy between the rankings produced by global CI and those from the local utility. In comparison, the proposed global utility yields rankings consistent with global CI, indicating that global utility is an effective metric when ground-truth SCM is unavailable. Consequently, global utility serves as an informative metric for evaluating global structural fidelity.

In addition to the correlation analysis of metric values, we compute Spearman's rank correlation between the generator rankings induced by three metrics: local utility, global CI, and global utility. As shown in Table 24, across 13 generators evaluated on six SCM datasets, generator rankings induced by global CI and global utility exhibit a strong correlation ($r_s = 0.95$, p < 0.001), whereas local utility shows substantially weaker alignment with the other two metrics.

We further analyse the rank correlations among the top-5 generators according to global CI: Tab-DDPM, TabSyn, TabDiff, TVAE, and ARF. When restricting the analysis to the top-5 generators

Table 23: **Top-3 tabular generators across the TabStruct benchmark suite.** For each dataset, we report the Top-3 tabular generators in terms of both local and global utility. For visualisation, we abbreviate "Classification" as "Class.", and 'Regression' as "Reg.". The results indicate that while SMOTE remains a simple yet effective approach for ML efficacy, diffusion models demonstrate stronger capability in capturing the holistic structure of tabular data.

| Dataset | | # Samples ($N$) | # Features ($D$) | $N/D$ | Local utility | | | Global utility | | |
|---|---|---|---|---|---|---|---|---|---|---|
| | | | | | 1st | 2nd | 3rd | 1st | 2nd | 3rd |
| **SCM datasets** | | | | | | | | | | |
| *Class.* | Hailfinder | 100,000 | 56 | 1785.71 | SMOTE | CTGAN | NRGBoost | TabDDPM | TabSyn | TabDiff |
| | Insurance | 100,000 | 27 | 3703.70 | SMOTE | TabEBM | TVAE | TabDDPM | TabDiff | TabSyn |
| | Sangiovese | 100,000 | 15 | 6666.67 | SMOTE | CTGAN | TabEBM | TabDDPM | TabSyn | TVAE |
| *Reg.* | Healthcare | 100,000 | 7 | 14285.71 | SMOTE | TabDiff | TabSyn | BN | ARF | TabDDPM |
| | MAGIC-IRRI | 100,000 | 64 | 1562.50 | SMOTE | TVAE | TabSyn | TVAE | TabDDPM | TabSyn |
| | MEHRA | 100,000 | 24 | 4166.67 | SMOTE | TabSyn | GOGGLE | TabDDPM | TabDiff | TabSyn |
| **Real-world datasets** | | | | | | | | | | |
| *Class.* | Ada | 4,562 | 48 | 95.04 | SMOTE | TabEBM | TabDiff | TVAE | TabDDPM | ARF |
| | Characters | 10,218 | 8 | 1277.25 | SMOTE | TabEBM | ARF | TabDDPM | TabSyn | TabDiff |
| | Credit-g | 1,000 | 21 | 47.62 | SMOTE | TabEBM | TabDiff | TabSyn | TabDiff | TabDDPM |
| | Electricity | 45,312 | 9 | 5034.67 | SMOTE | TabEBM | TabDiff | TabDDPM | TabDiff | ARF |
| | Higgs | 98,050 | 29 | 3381.03 | SMOTE | CTGAN | TabEBM | TabDDPM | TabSyn | TabDiff |
| | Jasmine | 2,984 | 145 | 20.58 | SMOTE | TVAE | TabSyn | TabSyn | TabDiff | TabDDPM |
| | Nomao | 34,465 | 119 | 289.62 | SMOTE | CTGAN | TVAE | TabDiff | TVAE | TabDDPM |
| | Phoneme | 5,404 | 6 | 900.67 | SMOTE | TabEBM | NRGBoost | TabDDPM | TabSyn | TabDiff |
| | Plants | 1,599 | 65 | 24.60 | SMOTE | TabEBM | NRGBoost | TabDDPM | TabSyn | TabDiff |
| | QSAR | 1,055 | 42 | 25.12 | SMOTE | TabEBM | NRGBoost | TabSyn | TabDDPM | TabDiff |
| | SpeedDating | 8,378 | 121 | 69.24 | SMOTE | TabEBM | TVAE | TabDDPM | TabSyn | TabDiff |
| | Splice | 3,190 | 61 | 52.30 | SMOTE | TVAE | CTGAN | TabSyn | TabDiff | TabDDPM |
| | Vehicle | 846 | 19 | 44.53 | SMOTE | TabEBM | TabSyn | TabSyn | TabDDPM | TabDiff |
| | Zernike | 2,000 | 48 | 41.67 | SMOTE | TabEBM | TVAE | TabSyn | TabDDPM | TabDiff |
| *Reg.* | Ailerons | 13,750 | 41 | 335.37 | SMOTE | TabDiff | TabSyn | TabDiff | TabDDPM | TabSyn |
| | California | 20,640 | 10 | 2064.00 | SMOTE | TabSyn | TabDiff | TabDDPM | TabSyn | TabDiff |
| | Elevators | 16,599 | 19 | 873.63 | SMOTE | TabDiff | TabSyn | TabDDPM | TabDiff | TabSyn |
| | H16 | 22,784 | 17 | 1340.24 | SMOTE | TabDiff | CTGAN | BN | TabDDPM | TabDiff |
| | Liver | 345 | 6 | 57.50 | TabDiff | TabSyn | SMOTE | ARF | TabDiff | TabSyn |
| | Sales | 21,613 | 20 | 1080.65 | SMOTE | TabDiff | TabSyn | TabDiff | TabSyn | TVAE |
| | Space | 3,107 | 7 | 443.86 | SMOTE | TabSyn | TabDiff | BN | TabDDPM | TabSyn |
| | Superconductivity | 21,263 | 82 | 259.30 | SMOTE | TabDiff | TabSyn | BN | TabDiff | TabSyn |
| | Wine | 6,497 | 12 | 541.42 | SMOTE | TabSyn | TabDiff | TabDiff | TabSyn | TabDDPM |

based on global CI, Table 25 shows the correlation between global CI and global utility remains high ($r_s = 0.92$, p $< 0.001$). This suggests that global utility maintains a consistent ranking even among high-performing generators, supporting its robustness in discerning top-performing models.

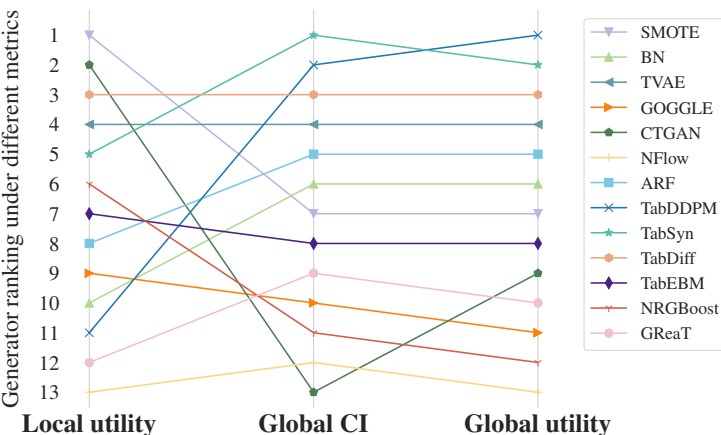

Figure 6: **Rank comparison of 13 tabular data generators across three evaluation metrics on six SCM datasets.** Compared to local utility, global CI and global utility rankings are relatively consistent, suggesting that global utility can serve as an effective metric for global structural fidelity.

**Global utility consistently aligns with global CI on large-scale SCMs.** To further validate the applicability of global utility in high-dimensional settings, we select five additional datasets (Table 7)

Table 24: **Spearman's rank correlation based on generator rankings on six SCM datasets.** Global utility induces generator rankings correlating strongly with global CI, showing the alignment between global utility and global CI ($p < 0.001$).

|  | Local utility | Global CI | Global utility |
|---|---|---|---|
| Local utility | – | – | – |
| Global CI | 0.29 | – | – |
| Global utility | 0.35 | 0.95 | – |

Table 25: **Spearman's rank correlation based on generator rankings of the top-5 generators (by global CI) on six SCM datasets.** Global utility maintains a consistent ranking even among high-performing generators, showing its robustness in discerning top-performing models ($p < 0.001$).

|  | Local utility | Global CI | Global utility |
|---|---|---|---|
| Local utility | – | – | – |
| Global CI | 0.38 | – | – |
| Global utility | 0.32 | 0.92 | – |

with large-scale SCMs from the `bnlearn` repository (Lemaître et al., 2017). Table 26 demonstrates that global utility remains strongly correlated with global CI across both these large SCM datasets and the six smaller ones discussed in Section 4. The results provide further empirical evidence that global utility reliably captures the global structural fidelity of tabular data.

Table 26: **Spearman's rank correlation with global CI on SCM datasets.** We **bold** the highest correlation with Global CI. Global utility correlates strongly with global CI across 11 SCM datasets ($p < 0.001$), demonstrating the generalisability of global utility.

| Dataset | Shape | Trend | $\alpha$-precision | $\beta$-recall | DCR | $\delta$-presence | Local utility | Local CI | **Global utility (Ours)** |
|---|---|---|---|---|---|---|---|---|---|
| 5 datasets with large SCMs | 0.26 | 0.24 | 0.31 | 0.33 | -0.25 | -0.22 | 0.13 | 0.11 | **0.81** |
| 11 SCM datasets (6 in Section 4 + 5 large) | 0.40 | 0.30 | 0.36 | 0.46 | -0.30 | -0.26 | 0.13 | 0.19 | **0.83** |

**Normalised utility is important for providing balanced and consistent evaluation across columns.** To further assess the impact of normalisation, we compare global utility computed using absolute predictive scores versus relative (i.e., normalised) scores. As shown in Table 27, using unnormalised scores leads to a substantially weaker correlation with global CI. Specifically, Spearman's drops from 0.84 to 0.57. This finding supports that the normalisation design plays an important role in improving the alignment with causal structures. Furthermore, global utility based on absolute scores fails to produce stable generator rankings across different predictor configurations. Specifically, the Top-5 generators under Full-tuned and Light-tuned settings share only one generator in common when computing global utility with absolute scores. This further supports that the normalisation enables global utility to deliver more robust and consistent evaluations across predictor configurations.

**Metrics requiring explicit causal discovery remain limited in evaluating the structural fidelity of tabular data.** We further examine the relationship between global CI and several metrics used in CauTabBench, which rely on causal discovery algorithms to infer SCMs from observed data. Specifically, we construct two dataset collections, A and B, each comprising three SCM datasets: A={Hailfinder, Insurance, MEHRA} and B={Sangiovese, Healthcare, MAGIC-IRRI}. For datasets in A, we compute our global utility metric alongside three CauTabBench metrics: skeleton-F1, direction-ACC, and direction-F1. For datasets in B, we compute the global CI. We then calculate Spearman's rank correlation between global CI and each of the other metrics. Table 28 shows that global utility exhibits a substantially stronger correlation with global CI than the metrics from CauTabBench. The relatively weaker correlation of CauTabBench metrics is likely due to their dependence on causal discovery algorithms. For instance, skeleton-F1 uses PC algorithm to recover causal graphs from synthetic tabular data. However, PC algorithms could suffer notable performance degradation as the number of features increases (Zanga et al., 2022; Zeng et al., 2022). This observation aligns with broader findings regarding the limitations of existing causal discovery

Table 27: **Spearman's rank correlation with global CI on SCM datasets.** We **bold** the highest correlation with Global CI. Global utility with normalised utility correlates strongly with global CI, showing that normalisation helps global utility to provide balanced evaluation across columns.

|  | Global CI $\uparrow$ |
| --- | --- |
| Global utility (absolute performance) | 0.57 |
| **Global utility (relative performance)** | **0.84** |

methods on real-world tabular datasets (Nastl & Hardt, 2024). These results suggest that global utility offers a more robust, SCM-free approach for evaluating global structural fidelity of tabular data.

Table 28: **Spearman's rank correlation with global CI across six SCM datasets.** We **bold** the highest correlation with Global CI. Global utility exhibits a substantially stronger correlation with global CI compared to the CauTabBench metrics ($p < 0.001$), which rely on causal discovery.

|  | Global CI $\uparrow$ |
| --- | --- |
| skeleton-F1 | 0.42 |
| direction-ACC | 0.44 |
| direction-F1 | 0.44 |
| **Global utility (Ours)** | **0.84** |

**Metrics for evaluating multi-table interactions are insufficient for structural fidelity within a single table.** Prior work (Pang et al., 2024; Solatorio & Dupriez, 2023), such as ClavaDDPM (Pang et al., 2024), which models relational databases, proposes the use of machine learning efficacy (MLE) to assess how well a generator preserves inter-table relationships. These studies primarily focus on relational structures across multiple tables, whereas TabStruct is designed to evaluate inter-feature causal relationships within a single table. Consequently, the prior studies do not explicitly establish a direct connection between MLE and the underlying causal structures of a single table. To quantitatively assess such a distinction, we evaluate the correlation between MLE and global CI using the same experimental setup as for global utility. We strictly follow the MLE evaluation procedure proposed in ClavaDDPM, following its official implementation (Pang et al., 2024). As shown in Table 29, both MLE-R2 and MLE-F1 exhibit relatively weak correlations with global CI, suggesting that multi-table relational metrics are less suitable for evaluating inter-feature causal interactions in single-table scenarios.

Table 29: **Spearman's rank correlation with global CI across six SCM datasets.** We **bold** the highest correlation with global CI. Global utility generally shows a stronger correlation with global CI compared to the metrics designed for multi-table settings ($p < 0.001$).

|  | Global CI $\uparrow$ |
| --- | --- |
| MLE-R2 | 0.40 |
| MLE-F1 | 0.44 |
| **Global utility (Ours)** | **0.84** |

**Global utility provides stable results with synthetic data of equal size to reference data.** Across six SCM datasets (Table 5 and Table 6), we fix $N_{ref}$ while varying the ratio $N_{ref} : N_{syn}$. We evaluate three representative tabular generators: SMOTE, TabSyn, and TabDDPM, which achieve the best results in local CI, global CI, and global utility, respectively. Table 30 shows that When $N_{syn} < N_{syn}$, global utility generally increases with the sample size of $\mathcal{D}_{syn}$. Once the condition $\mathcal{D}_{syn} \geq \mathcal{D}_{syn}$ is met, global utility tends to stabilise. This observation further validates the robustness of global utility score and supports our design rationale for using equal-sized $\mathcal{D}_{syn}$ and $\mathcal{D}_{ref}$ in the evaluation framework.

Table 30: **Global utility scores under different synthetic sample sizes.** We **bold** the highest global utility score for each generator. In general, global utility tends to saturate when the synthetic sample size reaches or exceeds that of the reference data.

| $N_{\text{ref}} : N_{\text{syn}}$ | SMOTE | TabSyn | TabDDPM |
|---|---|---|---|
| 5:1 | 0.13 | 0.62 | 0.64 |
| 3:1 | 0.25 | 0.72 | 0.74 |
| 1:1 | **0.39** | **0.76** | 0.80 |
| 1:3 | **0.39** | 0.75 | 0.79 |
| 1:5 | 0.38 | **0.76** | **0.81** |

### E.3 EXTENDED ANALYSIS ON STRUCTURAL FIDELITY OF GENERATORS

**Column order can have a notable impact on autoregressive tabular generators.** Autoregressive generators model the data distribution by linearising features according to a column order $\pi$. For tabular data, the ideal ordering $\pi^*$ corresponds to the topological order derived from the true SCM. However, since $\pi^*$ is typically unavailable in practice, using a mismatched $\pi$ may compromise structural fidelity. Although prior work (Borisov et al., 2023) attempts to improve robustness by finetuning LLMs on randomly permuted column orders, such approaches are computationally expensive (e.g., we observe that GReaT often fail to converge on datasets with more than 50 features) and do not explicitly align the model with the true causal structure of the dataset. For instance, if the random ordering $\pi$ happens to reverse the topological order encoded by the ground-truth causal structure, the autoregressive model is forced to learn spurious conditional independence across features, thereby harming the learned global structure. To investigate the impact of directional bias, we conduct a proof-of-concept experiment on six SCM datasets. Specifically, we introduce "GReaT-sort", a variant of GReaT finetuned using the ground-truth topological order extracted from each SCM. In this setup, GReaT-sort and the original GReaT share identical model configurations, and the only difference lies in the column order employed during finetuning. As shown in Table 31, GReaT-sort consistently outperforms GReaT across all datasets by a clear margin, suggesting that the mismatched bias in column ordering constrains the performance of autoregressive tabular generators.

Table 31: **Global utility of GReaT and GReaT-sort on six SCM datasets.** We **bold** the highest performance for each dataset. GReaT-sort consistently achieves higher global utility than GReaT, indicating that aligning column order with the underlying causal structure can effectively improve the performance of autoregressive tabular models.

| Generator | Hailfinder | Insurance | Sangiovese | Healthcare | MAGIC-IRRI | MEHRA |
|---|---|---|---|---|---|---|
| GReaT | $0.29_{\pm 0.28}$ | $0.32_{\pm 0.27}$ | $0.31_{\pm 0.28}$ | $0.19_{\pm 0.24}$ | $0.41_{\pm 0.24}$ | $0.26_{\pm 0.15}$ |
| GReaT-sorted | $\mathbf{0.43}_{\pm 0.23}$ | $\mathbf{0.49}_{\pm 0.14}$ | $\mathbf{0.40}_{\pm 0.23}$ | $\mathbf{0.43}_{\pm 0.22}$ | $\mathbf{0.49}_{\pm 0.13}$ | $\mathbf{0.31}_{\pm 0.12}$ |

**Interpolation and energy-based methods tend to prioritise local structure over global structure.** Figure 3 (right) shows that the interpolation method (e.g., SMOTE) and energy-based models (e.g., TabEBM and NRGBoost) can effectively capture local structure, yet perform poorly when modelling global structure. These two families of methods share a common trait in their generation process: they generate new samples from class-specific reference data. For example, in classification tasks, SMOTE interpolates between samples of the same class, and TabEBM samples from a class-specific energy surface. As a result, the generated samples are inevitably biased towards local structure.

**A high global utility score typically reflects consistently high utility across individual features.** As shown in Figure 7, generators with relatively high global utility scores (Type 1), such as TabSyn and TabDiff, tend to achieve balanced utility across features. This is largely because the global utility assigns equal importance to each feature, thereby reducing bias towards any particular one. As a result, generators with high global utility are less prone to overfitting a limited subset of features, thus achieving balanced performance across features. In contrast, generators with lower global utility scores generally fall into two categories (Type 2 and Type 3). The models of Type 2, including SMOTE and CTGAN, often achieve high utility on the target feature (i.e., local utility) but perform poorly on the others. This pattern primarily stems from their target-specific model design, which

inherently biases them towards the target feature. This is consistent with Figure 3 (Right), which shows that such models prioritise capturing the local structure of the target feature at the expense of the global structure. The models of Type 3 typically underperform across all features. We attribute it to their misaligned architectures for the unique characteristics of tabular data. A representative example is GReaT, which attempts to leverage domain knowledge from LLMs for tabular data generation. However, the mismatch between the textual modality of LLMs and the heterogeneous nature of tabular data undermines their ability to model tabular structures effectively.

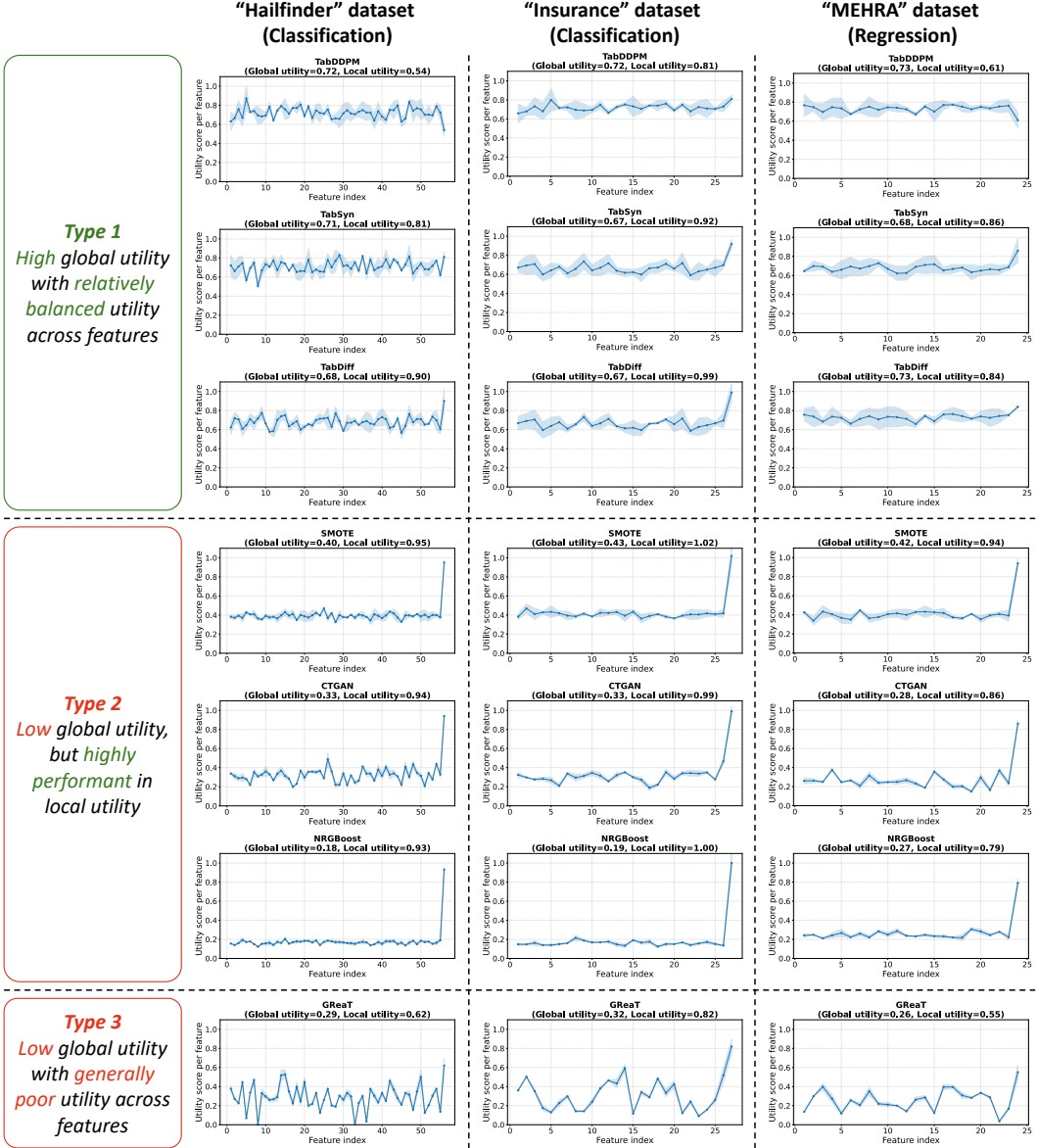

Figure 7: **Utility score distribution across features.** For visual clarity, we present seven representative generators and report their utility scores for each feature, accompanied by standard deviations. The results reveal that generators generally fall into three distinct categories: Generators with high global utility scores tend to exhibit balanced performance across all features. In contrast, those with relatively low global utility scores either display strong local utility or show generally poor performance across features.

### E.4 EXTENDED ANALYSIS ON PRACTICABILITY OF GLOBAL UTILITY

**Global utility remains stable across different downstream predictors.** Figure 8 shows that the relative rankings of tabular generators are consistent even when the number of downstream predictors is reduced from nine to three. In contrast, local utility is far more sensitive to the choice of predictors: its rankings fluctuate greatly even when simply reducing from nine to eight predictors. The instability of local utility stems from its bias towards the prediction target, which may introduce unfair bias towards specific types of predictors. For example, KNN tends to perform better when the number of classes is large (Jiang et al., 2024), while XGBoost typically favours skewed target distributions (McElfresh et al., 2024). Since local utility evaluates performance on a single feature, such biases are amplified, yielding unstable rankings even after ensembling different predictors. In contrast, global utility aggregates performance across all features, diluting predictor-specific biases and producing more robust generator rankings.

**Global utility is stable regardless of hyperparameter tuning.** Figure 8 shows that global utility provides consistent rankings of tabular generators regardless of whether downstream predictors are tuned. We note that this does not imply that tuning is unnecessary. In line with prior work (Kotelnikov et al., 2023; McElfresh et al., 2024; Du & Li, 2024), we also observe that tuning improves absolute performance. However, tuning has a negligible effect on the *relative* rankings under global utility. In contrast, local utility necessitates tuning to guarantee reliable results. Such robustness further reflects the core rationale of global utility: by not focusing on a single feature, it avoids introducing feature-specific biases and is therefore less susceptible to variation caused by tuning for a particular downstream prediction target. Such robustness further supports the rationale for using global utility as a stable and unbiased evaluation metric.

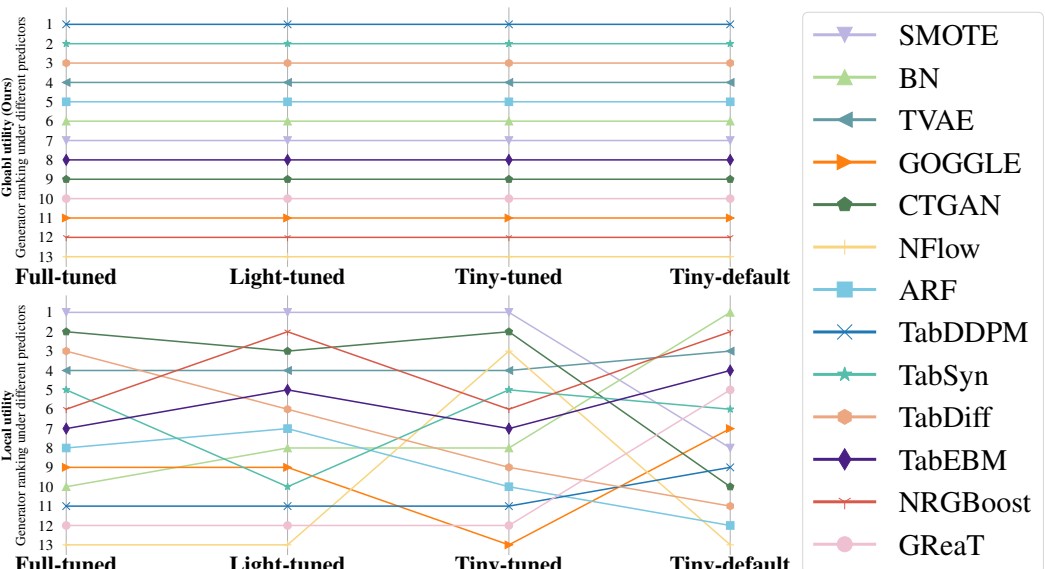

Figure 8: **Comparison of ranking stability between global utility and local utility on 23 real-world datasets, evaluated using different downstream predictors.** The proposed global utility produces consistent generator rankings across downstream predictors. In contrast, local utility necessitates a large set of tuned downstream predictors (i.e., Full-tuned) to yield meaningful rankings. As a result, global utility can achieve high computational efficiency with only a small ensemble of default predictors (i.e., Tiny-default in Figure 4).

**Global utility has the potential to evaluate global structural fidelity in the presence of latent confounders.** We present a proof-of-concept experiment to investigate the behaviour of global utility on datasets containing unobserved confounders, using three SCM datasets: Sangiovese, Healthcare, and MEHRA. Given an SCM, we introduce $n_{\text{conf}}$ exogenous latent confounders $U = \{U_1, \ldots, U_{n_{\text{conf}}}\}$, where each $U_k \sim \mathcal{N}(0, 1)$ is associated with a child set $S_k \subset \mathcal{X}$ such that $|S_k| = m_k \in \{2, 3, 4\}$. For each variable $x_j \in S_k$, we modify its structural function via $x_j = f_j(\text{pa}(x_j), \epsilon_j) + \sum_{k:j \in S_k} \lambda_{kj} U_k$, where the $U_k$ are exogenous and mutually independent, and $\lambda_{kj} \sim \text{Unif}(0.5, 1.5)$ controls the strength of confounding. This process yields a new causal graph denoted $\mathcal{G}_{\text{DAG}}^{\text{Conf}}$. We then marginalise out the

unobserved confounders to convert $\mathcal{G}_{\text{DAG}}^{\text{Conf}}$ into a maximal ancestral graph (MAG), $\mathcal{G}_{\text{MAG}}$, from which we derive conditional independence (CI) relationships using m-separation (Sadeghi & Lauritzen, 2014). Based on Equation (3), we compute "Global CI (MAG)", a variant of standard global CI, using the derived CI relations on the MAG. Global utility maintains stable performance (Table 32) and a strong correlation with Global CI (MAG) (Table 33) across different numbers of unobserved confounders. These results highlight the robustness and generalisability of global utility, and they suggest its potential for evaluating global structural fidelity in datasets containing latent confounding.

Table 32: **Benchmark results on three SCM datasets with injected unobserved confounders, aggregated over 13 tabular generators.** Global utility provides stable performance evaluations across varying numbers of unobserved confounders, demonstrating its robustness in assessing tabular data structures with latent confounding.

| $n_{\text{conf}}$ | Local CI (MAG) | Local utility | Global CI (MAG) | **Global utility (Ours)** |
|---|---|---|---|---|
| 1 | $0.84_{\pm 0.03}$ | $0.87_{\pm 0.02}$ | $0.80_{\pm 0.04}$ | $0.79_{\pm 0.03}$ |
| 2 | $0.84_{\pm 0.02}$ | $0.84_{\pm 0.03}$ | $0.80_{\pm 0.03}$ | $0.77_{\pm 0.02}$ |
| 3 | $0.81_{\pm 0.04}$ | $0.85_{\pm 0.02}$ | $0.75_{\pm 0.05}$ | $0.79_{\pm 0.04}$ |
| 4 | $0.80_{\pm 0.03}$ | $0.84_{\pm 0.04}$ | $0.74_{\pm 0.04}$ | $0.75_{\pm 0.03}$ |
| 5 | $0.79_{\pm 0.05}$ | $0.82_{\pm 0.03}$ | $0.71_{\pm 0.04}$ | $0.77_{\pm 0.02}$ |

Table 33: **Spearman's rank correlation with global CI (MAG) on three SCM datasets with injected unobserved confounders.** We **bold** the highest correlation with global CI (MAG). Global utility consistently exhibits a stronger and statistically significant correlation with global CI (MAG) compared to other metrics ($p < 0.001$).

| $n_{\text{conf}}$ | Shape | Trend | $\alpha$-precision | $\beta$-recall | DCR | $\delta$-presence | Local CI (MAG) | Local utility | **Global utility (Ours)** |
|---|---|---|---|---|---|---|---|---|---|
| 1 | 0.26 | 0.30 | 0.41 | 0.24 | -0.28 | -0.23 | 0.22 | 0.14 | **0.76** |
| 2 | 0.22 | 0.30 | 0.41 | 0.26 | -0.17 | -0.16 | 0.14 | 0.18 | **0.72** |
| 3 | 0.22 | 0.31 | 0.40 | 0.27 | -0.25 | -0.16 | 0.18 | 0.19 | **0.76** |
| 4 | 0.22 | 0.31 | 0.39 | 0.25 | -0.26 | -0.15 | 0.15 | 0.19 | **0.75** |
| 5 | 0.25 | 0.33 | 0.35 | 0.26 | -0.28 | -0.16 | 0.16 | 0.16 | **0.72** |

**Global utility is indicative of data utility for downstream causal inference tasks.** We perform a causal inference evaluation across 13 tabular data generators on six SCM datasets. Following the protocols of Chen et al. (2023a) and CauTabBench (Tu et al., 2024), we assess performance by learning SCMs from synthetic data and comparing them against ground-truth SCMs on both interventional and counterfactual inference tasks. For interventional evaluation, we apply 10 interventions per variable and generate 100,000 interventional samples per intervention under both $M_{\text{ref}}$ and $M_{\text{syn}}$. We then compute the interventional mean squared error (I-MSE) by comparing the expected values of the remaining variables. For counterfactual evaluation, given observed data, we apply 10 interventions per variable and generate 100,000 counterfactual samples using both $M_{\text{ref}}$ and $M_{\text{syn}}$. We compute the mean counterfactual values from both SCMs and calculate the counterfactual mean squared error (C-MSE). As shown in Table 34, the top five performing generators in the causal inference evaluation (TabSyn, TabDDPM, TabDiff, TVAE, and ARF) are consistent with those ranking highest in both global CI and global utility. This further supports the utility of global utility as an indicator of global causal structure in tabular data. Table 35 also shows that existing evaluation metrics exhibit considerably weaker correlations with causal inference performance, whereas global utility remains a reliable and effective indicator.

**Global utility provides stable evaluation across different degrees of data availability.** We select six SCM datasets (Table 5 and Table 6) and simulate varying levels of data availability by subsampling to smaller values of $N_{\text{full}}$. The corresponding reference sample size is $N_{\text{ref}} = N_{\text{full}} \times 0.8 \times 0.9$, as illustrated in Figure 5. As shown in Table 36, the proposed global utility metric consistently achieves the highest correlation with global CI across all evaluated sample sizes, clearly outperforming existing evaluation metrics. Notably, this holds even in very low-data scenarios, such as $N_{\text{full}} \leq 500$. These results suggest that global utility serves as a robust and reliable measure for global structural fidelity across a wide range of data availability.

Table 34: **Causal inference check of synthetic data from 13 tabular generators on six SCM datasets.** We **bold** the lowest error for both interventional and counterfactual tasks. Diffusion models generally achieve the best perfomance in downstream causal inference tasks.

| Generator | I-MSE $\downarrow$ | C-MSE $\downarrow$ |
|---|---|---|
| SMOTE | $0.32_{\pm 0.03}$ | $0.45_{\pm 0.04}$ |
| BN | $0.37_{\pm 0.02}$ | $0.51_{\pm 0.05}$ |
| TVAE | $0.16_{\pm 0.02}$ | $0.27_{\pm 0.03}$ |
| GOGGLE | $0.59_{\pm 0.04}$ | $0.75_{\pm 0.05}$ |
| CTGAN | $0.87_{\pm 0.05}$ | $0.90_{\pm 0.04}$ |
| NFlow | $0.97_{\pm 0.03}$ | $0.98_{\pm 0.02}$ |
| ARF | $0.12_{\pm 0.02}$ | $0.23_{\pm 0.03}$ |
| TabDDPM | $0.10_{\pm 0.01}$ | $0.22_{\pm 0.02}$ |
| TabSyn | $\mathbf{0.09}_{\pm 0.01}$ | $\mathbf{0.20}_{\pm 0.02}$ |
| TabDiff | $\mathbf{0.09}_{\pm 0.02}$ | $0.21_{\pm 0.02}$ |
| TabEBM | $0.34_{\pm 0.03}$ | $0.47_{\pm 0.04}$ |
| NRGBoost | $0.55_{\pm 0.04}$ | $0.70_{\pm 0.05}$ |
| GReaT | $0.59_{\pm 0.03}$ | $0.75_{\pm 0.04}$ |

Table 35: **Spearman's rank correlation between causal inference metrics and other metrics on six SCM datasets.** We **bold** the strongest correlation with causal inference performance. Global utility exhibits a strong correlation with causal inference metrics ($p < 0.001$), showing that global has the potential to indicate causal inference evaluations in SCM-free settings.

| | Shape | Trend | $\alpha$-precision | $\beta$-recall | DCR | $\delta$-presence | Local utility | Local CI | Global CI | **Global utility (Ours)** |
|---|---|---|---|---|---|---|---|---|---|---|
| I-MSE $\downarrow$ | -0.32 | -0.33 | -0.35 | -0.19 | 0.23 | 0.15 | -0.21 | -0.17 | -0.80 | **-0.90** |
| C-MSE $\downarrow$ | -0.17 | -0.40 | -0.16 | -0.14 | 0.36 | 0.21 | -0.45 | -0.24 | -0.82 | **-0.83** |

Table 36: **Spearman's rank correlation with global CI across different degrees of data availability.** We **bold** the strongest correlation with global CI for each degree. Global utility consistently correlates strongly with global CI, showing it a stable measure for global structural fidelity given different degrees of data availability ($p < 0.001$).

| $N_{\text{full}}$ | Shape | Trend | $\alpha$-precision | $\beta$-recall | DCR | $\delta$-presence | Local utility | Local CI | **Global utility (Ours)** |
|---|---|---|---|---|---|---|---|---|---|
| 100 | 0.40 | 0.51 | 0.36 | 0.46 | -0.41 | -0.37 | 0.20 | 0.26 | **0.82** |
| 500 | 0.43 | 0.51 | 0.32 | 0.52 | -0.49 | -0.39 | 0.24 | 0.22 | **0.83** |
| 1,000 | 0.43 | 0.46 | 0.35 | 0.53 | -0.49 | -0.42 | 0.22 | 0.19 | **0.87** |
| 5,000 | 0.47 | 0.41 | 0.42 | 0.50 | -0.48 | -0.45 | 0.18 | 0.13 | **0.83** |
| 10,000 | 0.43 | 0.46 | 0.33 | 0.55 | -0.47 | -0.42 | 0.14 | 0.15 | **0.89** |
| 100,000 | 0.47 | 0.47 | 0.37 | 0.49 | -0.43 | -0.40 | 0.14 | 0.22 | **0.84** |

**TabStruct can provide customised results on global structural fidelity evaluation.** We consider two variants of global CI: (i) Global CI (discrete), which is the current global CI reported in Section 4, and (ii) Global CI (continuous). Instead of using binary CI test outcomes as in Equation (3), we compute the average $\alpha$ level at which CI tests fail across all features to obtain a continuous global CI score. Similarly, we consider two variants of global utility: Global utility (continuous), which is the current global utility reported in Section 4, and (ii) Global utility (discrete). Instead of normalised downstream performance in Equation (4), we perform a Wilcoxon signed-rank test ($\alpha = 0.01$) between $\text{Perf}(\mathcal{D}_{\text{ref}})$ and $\text{Perf}(\mathcal{D})$ for each feature, then average the resulting binary outcomes. Table 37 shows that all variants of global CI and global utility, both continuous and discrete, exhibit strong mutual correlations. Notably, their correlation strengths are very similar, ranging between $[0.80, 0.86]$, indicating consistent alignment across formulations, which allows users to select either the continuous or discrete variant for global CI and global utility.

Table 37: **Spearman's rank correlation based on generator rankings on six SCM datasets.** The variants of global CI and global utility stably shows strong correlation, indicating that global utility is an effective and robust measure for global structural fidelity ($p < 0.001$).

|  | Global CI (discrete) | Global CI (continuous) |
|---|---|---|
| Global utility (continuous) | 0.84 | 0.86 |
| Global utility (discrete) | 0.80 | 0.83 |

### E.5 PRACTICAL GUIDANCE

**Evaluation dimensions are complementary, not interchangeable.** Table 2 shows that no single metric is fully indicative of all other metrics. Therefore, researchers and practitioners should select evaluation dimensions that align with the specific objectives of their tasks, rather than relying on a single dimension. If the objective is leakage-free data sharing, the privacy preservation and ML efficacy should be prioritised over density estimation and structural fidelity. Conversely, when the aim is to model a real-world physical system like Figure 1, global structural fidelity should take precedence, because it promotes realistic inter-feature relationships, instead of being distorted towards a single prediction target.

**SMOTE is a simple yet effective method for ML efficacy.** In Table 23, we provide per-dataset guidance for selecting appropriate tabular generators based on ML efficacy. Surprisingly, SMOTE achieves the highest local utility on 28 out of 29 datasets. Despite this strong performance, Table 4 shows that it has been largely overlooked in prior studies (Shi et al., 2025; Bravo, 2025; Xu et al., 2019). We strongly encourage researchers and practitioners to consider SMOTE as a robust baseline in scenarios where ML efficacy is the primary objective and other dimensions, such as privacy, are less critical. For instance, in data augmentation tasks, SMOTE can serve as an effective baseline to compare against.

### E.6 FUTURE WORK

**Investigation using a separate independent set for performance evaluation.** Different from the well-established experimental setup in Section 4, we can modify the data splitting strategy into $\mathcal{D}_{\text{train}} : \mathcal{D}_{\text{test}} : \mathcal{D}_{\text{indep}} : \mathcal{D}_{\text{valid}} = 3 : 3 : 3 : 1$. In this configuration, $\mathcal{D}_{\text{indep}}$ acts as an independent benchmark to assess whether a generator is underfitting or overfitting the training data. Figure 9 shows the proof-of-concept results on three SCM datasets (Table 5). An interesting observation is that SMOTE outperforms $\mathcal{D}_{\text{indep}}$ in the Trend metric but performs much worse in DCR. This indicates that SMOTE generally overfits to the training data $\mathcal{D}_{\text{train}}$, rather than learning truly generalisable distributions. In addition, Figure 9 suggests that simply maximising DCR can degrade performance in other aspects, such as density estimation. Therefore, although some generators demonstrate high DCR scores, they may not be ideal if they severely compromise on other metrics. Therefore, achieving a higher DCR than $\mathcal{D}_{\text{indep}}$ may be a more balanced and practical criterion for acceptable privacy preservation, rather than pushing DCR to its maximum. We further observe that varying the data splitting strategy can have notable impacts on model performance. For instance, TabDiff excels in Trend under the original splitting strategy (Section 4) but underperforms in the new configuration. We attribute this to model-specific properties. As the new strategy allocates fewer samples for training, it implies a potential limitation for fitting tabular generators. Specifically, TabDiff appears more

susceptible to reduced training sample sizes than other benchmark methods. As TabStruct is fully open-source and will continue to evolve with contributions from the community, we believe that incorporating an independent set holds promise for offering a fresh and valuable perspective in assessing the performance of tabular data generators.

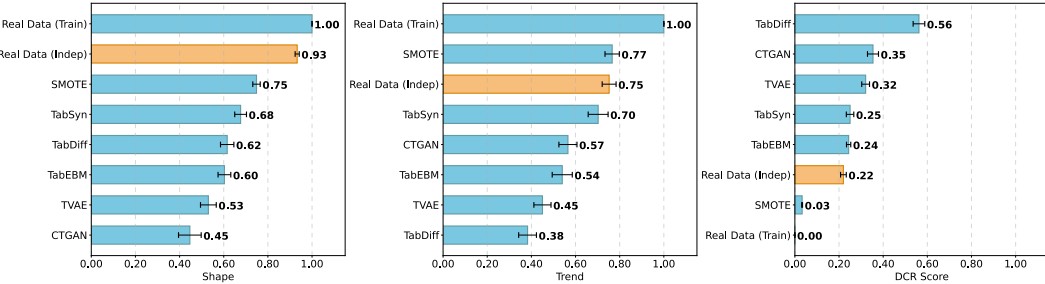

Figure 9: **Benchmark results of six representative tabular generators on three SCM classification datasets with a separate independent set.** The results show that $\mathcal{D}_{\text{Indep}}$, i.e., orange bar, offers a novel and complementary perspective for evaluating the performance of tabular data generators.

**Theoretical justifications for causal modelling of tabular data.** Bridging the gap between empirical metrics on real-world tabular datasets and structural causal models (SCMs) remains a major theoretical challenge in causal machine learning (Nastl & Hardt, 2024; Tu et al., 2024; Zanga et al., 2022). A promising direction for future research lies in developing theoretical underpinnings for the proposed global utility metric. Currently, the proposed global utility serves as an empirically effective metric for structural fidelity, grounded in its correlation with conditional independence (CI) scores. A more rigorous formalisation could help enhance its interpretability in relation to specific causal relationships, and potentially inspire new paradigms for evaluating tabular generators.

**Efficient and accurate causal discovery in real-world scenarios.** A promising direction for future work is the development of more effective causal discovery algorithms for real-world tabular data. In practical scenarios, ground-truth causal graphs are seldom available, and despite progress in constraint-based, score-based, and hybrid approaches, reliably recovering even partial or probabilistic SCMs remains a challenge – particularly in high-dimensional settings (Zeng et al., 2022; Kaddour et al., 2022; Nastl & Hardt, 2024). Nevertheless, incorporating such approximated structures as priors or regularisers in the global utility computation could enhance both its scalability and its fidelity to causal semantics. This would not only enable structural fidelity evaluation on more complex datasets but also improve the robustness of global utility by reducing the influence of spurious statistical associations.

**Structure-aware tabular data generation.** Beyond evaluation, another important avenue for future work is the design of structure-aware tabular data generators that are explicitly optimised for structural fidelity. These models could embed inductive biases or incorporate regularisation objectives that encourage alignment with the conditional independence structure observed in the reference data. This would mark a shift away from conventional likelihood-driven generation toward structure-informed tabular data generation, enabling the generation of data that better complies with domain-specific constraints (e.g., scientific laws in Figure 1).

**Extension to dynamic and temporal data modalities.** While TabStruct already offers broad coverage of static tabular datasets (Appendix A), a promising direction for future work is to extend the framework to support temporal and event-based data, where causal relationships may change over time. Many real-world domains – such as healthcare, finance, and operations research – exhibit longitudinal structures that challenge the assumptions of static SCMs (Borisov et al., 2022). Adapting global utility to reflect time-dependent causal structures would broaden TabStruct's applicability.

# F    EXTENDED EXPERIMENTAL RESULTS

## F.1    EVALUATION RESULTS FOR SCM DATASETS

### F.1.1    CLASSIFICATION DATASETS

Table 38: **Raw benchmark results of 13 tabular generators on "Hailfinder" dataset.** We report the normalised mean $\pm$ std metric values across datasets. We highlight the **First**, **Second** and **Third** best performances for each metric. For visualisation, we abbreviate "conditional independence" as "CI". SMOTE generally achieves the highest performance in capturing local structure (i.e., local utility or local CI), while diffusion models typically excel at capturing global structure (i.e., global CI or global utility).

| Generator | Density Estimation | | | | Privacy Preservation | | ML Efficacy | Structural Fidelity | | |
|---|---|---|---|---|---|---|---|---|---|---|
| | Shape ↑ | Trend ↑ | $\alpha$-precision ↑ | $\beta$-recall ↑ | DCR ↑ | $\delta$-Presence ↑ | Local utility ↑ | Local CI ↑ | Global CI ↑ | Global utility ↑ |
| $\mathcal{D}_{\text{ref}}$ | $1.00_{\pm0.00}$ | $1.00_{\pm0.00}$ | $1.00_{\pm0.00}$ | $1.00_{\pm0.00}$ | $0.00_{\pm0.00}$ | $0.00_{\pm0.00}$ | $1.00_{\pm0.00}$ | $0.97_{\pm0.11}$ | $0.98_{\pm0.00}$ | $1.00_{\pm0.00}$ |
| SMOTE | $0.99_{\pm0.00}$ | $0.98_{\pm0.00}$ | $0.89_{\pm0.00}$ | $0.90_{\pm0.00}$ | $0.00_{\pm0.00}$ | $0.00_{\pm0.00}$ | $0.95_{\pm0.06}$ | $0.89_{\pm0.03}$ | $0.53_{\pm0.10}$ | $0.40_{\pm0.38}$ |
| BN | $0.99_{\pm0.00}$ | $0.98_{\pm0.00}$ | $0.99_{\pm0.00}$ | $0.39_{\pm0.00}$ | $0.64_{\pm0.02}$ | $0.00_{\pm0.00}$ | $0.66_{\pm0.24}$ | $0.49_{\pm0.06}$ | $0.52_{\pm0.09}$ | $0.36_{\pm0.35}$ |
| TVAE | $0.94_{\pm0.00}$ | $0.90_{\pm0.01}$ | $0.86_{\pm0.02}$ | $0.17_{\pm0.03}$ | $0.66_{\pm0.04}$ | $0.01_{\pm0.01}$ | $0.92_{\pm0.04}$ | $0.78_{\pm0.09}$ | $0.61_{\pm0.11}$ | $0.66_{\pm0.10}$ |
| GOGGLE | $0.94_{\pm0.02}$ | $0.89_{\pm0.04}$ | $0.86_{\pm0.06}$ | $0.37_{\pm0.08}$ | $0.43_{\pm0.03}$ | $0.19_{\pm0.41}$ | $0.62_{\pm0.20}$ | $0.53_{\pm0.10}$ | $0.51_{\pm0.08}$ | $0.29_{\pm0.28}$ |
| CTGAN | $0.93_{\pm0.01}$ | $0.90_{\pm0.02}$ | $0.96_{\pm0.02}$ | $0.29_{\pm0.05}$ | $0.61_{\pm0.01}$ | $0.01_{\pm0.01}$ | $0.94_{\pm0.06}$ | $0.73_{\pm0.18}$ | $0.48_{\pm0.06}$ | $0.33_{\pm0.31}$ |
| NFlow | $0.88_{\pm0.01}$ | $0.81_{\pm0.02}$ | $0.74_{\pm0.08}$ | $0.00_{\pm0.00}$ | $0.65_{\pm0.03}$ | $0.02_{\pm0.01}$ | $0.52_{\pm0.06}$ | $0.53_{\pm0.02}$ | $0.53_{\pm0.01}$ | $0.04_{\pm0.04}$ |
| ARF | $0.96_{\pm0.04}$ | $0.93_{\pm0.06}$ | $0.91_{\pm0.08}$ | $0.28_{\pm0.11}$ | $0.54_{\pm0.14}$ | $0.06_{\pm0.16}$ | $0.81_{\pm0.03}$ | $0.57_{\pm0.05}$ | $0.55_{\pm0.03}$ | $0.54_{\pm0.05}$ |
| TabDDPM | $0.90_{\pm0.06}$ | $0.85_{\pm0.08}$ | $0.54_{\pm0.40}$ | $0.23_{\pm0.22}$ | $0.53_{\pm0.09}$ | $0.01_{\pm0.00}$ | $0.54_{\pm0.18}$ | $0.48_{\pm0.06}$ | $0.66_{\pm0.15}$ | $0.72_{\pm0.21}$ |
| TabSyn | $0.81_{\pm0.16}$ | $0.64_{\pm0.30}$ | $0.73_{\pm0.23}$ | $0.22_{\pm0.23}$ | $0.22_{\pm0.24}$ | $1.96_{\pm4.57}$ | $0.81_{\pm0.25}$ | $0.74_{\pm0.17}$ | $0.71_{\pm0.09}$ | $0.71_{\pm0.23}$ |
| TabDiff | $0.97_{\pm0.01}$ | $0.95_{\pm0.02}$ | $0.95_{\pm0.04}$ | $0.36_{\pm0.08}$ | $0.40_{\pm0.05}$ | $0.01_{\pm0.00}$ | $0.90_{\pm0.07}$ | $0.67_{\pm0.16}$ | $0.62_{\pm0.11}$ | $0.68_{\pm0.09}$ |
| TabEBM | $0.94_{\pm0.03}$ | $0.90_{\pm0.04}$ | $0.88_{\pm0.06}$ | $0.34_{\pm0.12}$ | $0.39_{\pm0.14}$ | $0.14_{\pm0.40}$ | $0.91_{\pm0.10}$ | $0.77_{\pm0.15}$ | $0.51_{\pm0.09}$ | $0.30_{\pm0.29}$ |
| NRGBoost | $0.93_{\pm0.03}$ | $0.89_{\pm0.05}$ | $0.86_{\pm0.06}$ | $0.22_{\pm0.23}$ | $0.51_{\pm0.07}$ | $0.02_{\pm0.01}$ | $0.93_{\pm0.07}$ | $0.73_{\pm0.17}$ | $0.47_{\pm0.09}$ | $0.18_{\pm0.27}$ |
| GReaT | $0.94_{\pm0.02}$ | $0.89_{\pm0.04}$ | $0.86_{\pm0.06}$ | $0.37_{\pm0.08}$ | $0.43_{\pm0.03}$ | $0.19_{\pm0.41}$ | $0.62_{\pm0.20}$ | $0.53_{\pm0.10}$ | $0.51_{\pm0.08}$ | $0.29_{\pm0.28}$ |

Table 39: **Raw benchmark results of 13 tabular generators on "Insurance" dataset.** We report the normalised mean $\pm$ std metric values across datasets. We highlight the **First**, **Second** and **Third** best performances for each metric. For visualisation, we abbreviate "conditional independence" as "CI". SMOTE generally achieves the highest performance in capturing local structure (i.e., local utility or local CI), while diffusion models typically excel at capturing global structure (i.e., global CI or global utility).

| Generator | Density Estimation | | | | Privacy Preservation | | ML Efficacy | Structural Fidelity | | |
|---|---|---|---|---|---|---|---|---|---|---|
| | Shape ↑ | Trend ↑ | $\alpha$-precision ↑ | $\beta$-recall ↑ | DCR ↑ | $\delta$-Presence ↑ | Local utility ↑ | Local CI ↑ | Global CI ↑ | Global utility ↑ |
| $\mathcal{D}_{\text{ref}}$ | $1.00_{\pm0.00}$ | $1.00_{\pm0.00}$ | $1.00_{\pm0.00}$ | $1.00_{\pm0.00}$ | $0.00_{\pm0.00}$ | $0.00_{\pm0.00}$ | $1.00_{\pm0.00}$ | $1.00_{\pm0.00}$ | $0.97_{\pm0.01}$ | $1.00_{\pm0.00}$ |
| SMOTE | $0.99_{\pm0.00}$ | $0.99_{\pm0.00}$ | $0.97_{\pm0.00}$ | $0.93_{\pm0.00}$ | $0.00_{\pm0.00}$ | $0.00_{\pm0.00}$ | $1.02_{\pm0.02}$ | $0.86_{\pm0.14}$ | $0.53_{\pm0.06}$ | $0.43_{\pm0.39}$ |
| BN | $0.99_{\pm0.00}$ | $0.97_{\pm0.00}$ | $0.95_{\pm0.00}$ | $0.65_{\pm0.00}$ | $0.48_{\pm0.01}$ | $0.00_{\pm0.00}$ | $0.85_{\pm0.18}$ | $0.66_{\pm0.17}$ | $0.55_{\pm0.08}$ | $0.31_{\pm0.27}$ |
| TVAE | $0.97_{\pm0.00}$ | $0.94_{\pm0.01}$ | $0.93_{\pm0.00}$ | $0.70_{\pm0.01}$ | $0.51_{\pm0.01}$ | $0.00_{\pm0.00}$ | $1.01_{\pm0.03}$ | $0.67_{\pm0.20}$ | $0.58_{\pm0.09}$ | $0.67_{\pm0.12}$ |
| GOGGLE | $0.95_{\pm0.01}$ | $0.92_{\pm0.02}$ | $0.90_{\pm0.03}$ | $0.61_{\pm0.10}$ | $0.29_{\pm0.03}$ | $0.02_{\pm0.03}$ | $0.82_{\pm0.15}$ | $0.57_{\pm0.08}$ | $0.52_{\pm0.05}$ | $0.32_{\pm0.27}$ |
| CTGAN | $0.94_{\pm0.01}$ | $0.91_{\pm0.01}$ | $0.92_{\pm0.04}$ | $0.69_{\pm0.02}$ | $0.36_{\pm0.05}$ | $0.01_{\pm0.00}$ | $0.99_{\pm0.04}$ | $0.76_{\pm0.25}$ | $0.49_{\pm0.03}$ | $0.33_{\pm0.29}$ |
| NFlow | $0.92_{\pm0.02}$ | $0.86_{\pm0.03}$ | $0.79_{\pm0.08}$ | $0.23_{\pm0.07}$ | $0.56_{\pm0.05}$ | $0.02_{\pm0.01}$ | $0.73_{\pm0.13}$ | $0.50_{\pm0.01}$ | $0.51_{\pm0.01}$ | $0.21_{\pm0.19}$ |
| ARF | $0.97_{\pm0.01}$ | $0.95_{\pm0.03}$ | $0.92_{\pm0.02}$ | $0.76_{\pm0.11}$ | $0.33_{\pm0.03}$ | $0.00_{\pm0.00}$ | $0.96_{\pm0.02}$ | $0.58_{\pm0.08}$ | $0.54_{\pm0.03}$ | $0.64_{\pm0.07}$ |
| TabDDPM | $0.91_{\pm0.06}$ | $0.86_{\pm0.09}$ | $0.81_{\pm0.14}$ | $0.59_{\pm0.12}$ | $0.41_{\pm0.12}$ | $0.01_{\pm0.00}$ | $0.81_{\pm0.15}$ | $0.54_{\pm0.08}$ | $0.64_{\pm0.10}$ | $0.72_{\pm0.14}$ |
| TabSyn | $0.87_{\pm0.11}$ | $0.76_{\pm0.20}$ | $0.72_{\pm0.23}$ | $0.36_{\pm0.37}$ | $0.16_{\pm0.17}$ | $0.22_{\pm0.40}$ | $0.92_{\pm0.16}$ | $0.82_{\pm0.19}$ | $0.66_{\pm0.08}$ | $0.67_{\pm0.19}$ |
| TabDiff | $0.98_{\pm0.01}$ | $0.96_{\pm0.02}$ | $0.96_{\pm0.04}$ | $0.64_{\pm0.07}$ | $0.29_{\pm0.03}$ | $0.00_{\pm0.00}$ | $0.99_{\pm0.05}$ | $0.72_{\pm0.18}$ | $0.60_{\pm0.09}$ | $0.67_{\pm0.12}$ |
| TabEBM | $0.98_{\pm0.01}$ | $0.96_{\pm0.02}$ | $0.95_{\pm0.03}$ | $0.42_{\pm0.30}$ | $0.16_{\pm0.17}$ | $0.00_{\pm0.00}$ | $1.02_{\pm0.04}$ | $0.80_{\pm0.21}$ | $0.51_{\pm0.05}$ | $0.34_{\pm0.30}$ |
| NRGBoost | $0.94_{\pm0.03}$ | $0.89_{\pm0.05}$ | $0.85_{\pm0.08}$ | $0.37_{\pm0.35}$ | $0.23_{\pm0.09}$ | $0.02_{\pm0.02}$ | $1.00_{\pm0.04}$ | $0.79_{\pm0.22}$ | $0.49_{\pm0.05}$ | $0.19_{\pm0.22}$ |
| GReaT | $0.95_{\pm0.01}$ | $0.92_{\pm0.02}$ | $0.90_{\pm0.03}$ | $0.61_{\pm0.10}$ | $0.29_{\pm0.03}$ | $0.02_{\pm0.03}$ | $0.82_{\pm0.15}$ | $0.57_{\pm0.08}$ | $0.52_{\pm0.05}$ | $0.32_{\pm0.27}$ |

Table 40: **Raw benchmark results of 13 tabular generators on "Sangiovese" dataset.** We report the normalised mean ± std metric values across datasets. We highlight the First, Second and Third best performances for each metric. For visualisation, we abbreviate "conditional independence" as "CI". SMOTE generally achieves the highest performance in capturing local structure (i.e., local utility or local CI), while diffusion models typically excel at capturing global structure (i.e., global CI or global utility).

| Generator | Density Estimation | | | | Privacy Preservation | | ML Efficacy | Structural Fidelity | | |
|---|---|---|---|---|---|---|---|---|---|---|
| | Shape ↑ | Trend ↑ | $\alpha$-precision ↑ | $\beta$-recall ↑ | DCR ↑ | $\delta$-Presence ↑ | Local utility ↑ | Local CI ↑ | Global CI ↑ | Global utility ↑ |
| $\mathcal{D}_{\mathrm{ref}}$ | $1.00_{\pm 0.00}$ | $1.00_{\pm 0.00}$ | $1.00_{\pm 0.00}$ | $1.00_{\pm 0.00}$ | $0.00_{\pm 0.00}$ | $0.00_{\pm 0.00}$ | $1.00_{\pm 0.00}$ | $0.99_{\pm 0.02}$ | $0.99_{\pm 0.02}$ | $1.00_{\pm 0.00}$ |
| SMOTE | $\mathbf{0.99}_{\pm 0.00}$ | $\mathbf{0.98}_{\pm 0.00}$ | $0.90_{\pm 0.00}$ | $\mathbf{0.88}_{\pm 0.00}$ | $0.23_{\pm 0.02}$ | $0.00_{\pm 0.00}$ | $\mathbf{0.95}_{\pm 0.08}$ | $\mathbf{0.79}_{\pm 0.17}$ | $0.56_{\pm 0.07}$ | $0.41_{\pm 0.38}$ |
| BN | $\mathbf{0.99}_{\pm 0.00}$ | $\mathbf{0.97}_{\pm 0.00}$ | $0.95_{\pm 0.01}$ | $0.30_{\pm 0.00}$ | $\mathbf{0.37}_{\pm 0.02}$ | $0.00_{\pm 0.00}$ | $0.52_{\pm 0.12}$ | $0.53_{\pm 0.07}$ | $0.53_{\pm 0.05}$ | $0.29_{\pm 0.28}$ |
| TVAE | $0.95_{\pm 0.01}$ | $0.94_{\pm 0.00}$ | $0.93_{\pm 0.01}$ | $0.36_{\pm 0.01}$ | $\mathbf{0.43}_{\pm 0.04}$ | $0.01_{\pm 0.00}$ | $0.89_{\pm 0.07}$ | $0.66_{\pm 0.18}$ | $0.62_{\pm 0.15}$ | $\mathbf{0.76}_{\pm 0.15}$ |
| GOGGLE | $0.96_{\pm 0.01}$ | $0.95_{\pm 0.00}$ | $0.93_{\pm 0.02}$ | $\mathbf{0.44}_{\pm 0.07}$ | $0.28_{\pm 0.01}$ | $\mathbf{0.05}_{\pm 0.05}$ | $0.60_{\pm 0.19}$ | $0.56_{\pm 0.08}$ | $0.55_{\pm 0.06}$ | $0.31_{\pm 0.28}$ |
| CTGAN | $0.93_{\pm 0.01}$ | $0.95_{\pm 0.01}$ | $0.95_{\pm 0.01}$ | $0.36_{\pm 0.02}$ | $0.31_{\pm 0.03}$ | $0.01_{\pm 0.00}$ | $\mathbf{0.91}_{\pm 0.09}$ | $0.73_{\pm 0.22}$ | $0.50_{\pm 0.02}$ | $0.30_{\pm 0.28}$ |
| NFlow | $0.89_{\pm 0.03}$ | $0.89_{\pm 0.01}$ | $0.82_{\pm 0.10}$ | $0.15_{\pm 0.04}$ | $0.32_{\pm 0.02}$ | $0.04_{\pm 0.02}$ | $0.41_{\pm 0.02}$ | $0.54_{\pm 0.06}$ | $0.51_{\pm 0.03}$ | $0.20_{\pm 0.17}$ |
| ARF | $0.97_{\pm 0.01}$ | $0.95_{\pm 0.01}$ | $0.93_{\pm 0.02}$ | $0.44_{\pm 0.08}$ | $0.29_{\pm 0.02}$ | $0.04_{\pm 0.05}$ | $0.76_{\pm 0.08}$ | $0.62_{\pm 0.04}$ | $0.59_{\pm 0.03}$ | $0.60_{\pm 0.06}$ |
| TabDDPM | $0.98_{\pm 0.02}$ | $\mathbf{0.97}_{\pm 0.02}$ | $\mathbf{0.96}_{\pm 0.04}$ | $\mathbf{0.45}_{\pm 0.07}$ | $0.18_{\pm 0.09}$ | $0.00_{\pm 0.00}$ | $0.63_{\pm 0.24}$ | $0.61_{\pm 0.15}$ | $\mathbf{0.77}_{\pm 0.11}$ | $\mathbf{0.84}_{\pm 0.19}$ |
| TabSyn | $0.97_{\pm 0.01}$ | $0.96_{\pm 0.01}$ | $\mathbf{0.96}_{\pm 0.04}$ | $0.39_{\pm 0.12}$ | $0.30_{\pm 0.04}$ | $0.01_{\pm 0.00}$ | $0.89_{\pm 0.11}$ | $\mathbf{0.80}_{\pm 0.16}$ | $\mathbf{0.73}_{\pm 0.13}$ | $\mathbf{0.81}_{\pm 0.20}$ |
| TabDiff | $0.97_{\pm 0.01}$ | $0.94_{\pm 0.03}$ | $0.93_{\pm 0.01}$ | $0.34_{\pm 0.18}$ | $0.28_{\pm 0.10}$ | $0.01_{\pm 0.00}$ | $0.85_{\pm 0.09}$ | $0.69_{\pm 0.15}$ | $\mathbf{0.65}_{\pm 0.13}$ | $0.66_{\pm 0.20}$ |
| TabEBM | $0.97_{\pm 0.02}$ | $0.97_{\pm 0.01}$ | $\mathbf{0.95}_{\pm 0.03}$ | $0.36_{\pm 0.15}$ | $\mathbf{0.38}_{\pm 0.11}$ | $0.00_{\pm 0.00}$ | $\mathbf{0.91}_{\pm 0.10}$ | $\mathbf{0.80}_{\pm 0.16}$ | $0.56_{\pm 0.08}$ | $0.30_{\pm 0.27}$ |
| NRGBoost | $\mathbf{0.98}_{\pm 0.02}$ | $0.91_{\pm 0.05}$ | $0.89_{\pm 0.03}$ | $0.29_{\pm 0.23}$ | $0.30_{\pm 0.04}$ | $\mathbf{0.46}_{\pm 0.52}$ | $0.89_{\pm 0.11}$ | $0.76_{\pm 0.19}$ | $0.53_{\pm 0.05}$ | $0.17_{\pm 0.23}$ |
| GReaT | $0.96_{\pm 0.01}$ | $0.95_{\pm 0.00}$ | $0.93_{\pm 0.02}$ | $\mathbf{0.44}_{\pm 0.07}$ | $0.28_{\pm 0.01}$ | $\mathbf{0.05}_{\pm 0.05}$ | $0.60_{\pm 0.19}$ | $0.56_{\pm 0.08}$ | $0.55_{\pm 0.06}$ | $0.31_{\pm 0.28}$ |

### F.1.2 REGRESSION DATASETS

Table 41: **Raw benchmark results of 13 tabular generators on "Healthcare" dataset.** We report the normalised mean ± std metric values across datasets. We highlight the **First**, **Second** and **Third** best performances for each metric. For visualisation, we abbreviate "conditional independence" as "CI". SMOTE generally achieves the highest performance in capturing local structure (i.e., local utility or local CI), while diffusion models typically excel at capturing global structure (i.e., global CI or global utility).

| Generator | Density Estimation | | | | Privacy Preservation | | ML Efficacy | Structural Fidelity | | |
| --- | --- | --- | --- | --- | --- | --- | --- | --- | --- | --- |
| | Shape ↑ | Trend ↑ | α-precision ↑ | β-recall ↑ | DCR ↑ | δ-Presence ↑ | Local utility ↑ | Local CI ↑ | Global CI ↑ | Global utility ↑ |
| $\mathcal{D}_{\text{ref}}$ | $1.00_{\pm 0.00}$ | $1.00_{\pm 0.00}$ | $1.00_{\pm 0.00}$ | $1.00_{\pm 0.00}$ | $0.00_{\pm 0.00}$ | $0.00_{\pm 0.00}$ | $1.00_{\pm 0.00}$ | $0.98_{\pm 0.06}$ | $0.98_{\pm 0.03}$ | $1.00_{\pm 0.00}$ |
| SMOTE | $0.89_{\pm 0.00}$ | $0.99_{\pm 0.00}$ | $1.00_{\pm 0.00}$ | $0.59_{\pm 0.00}$ | $0.00_{\pm 0.00}$ | $0.00_{\pm 0.00}$ | $1.01_{\pm 0.01}$ | $0.74_{\pm 0.10}$ | $0.60_{\pm 0.12}$ | $0.50_{\pm 0.48}$ |
| BN | $0.93_{\pm 0.00}$ | $0.98_{\pm 0.01}$ | $0.98_{\pm 0.00}$ | $0.09_{\pm 0.00}$ | $0.01_{\pm 0.00}$ | $0.00_{\pm 0.00}$ | $0.72_{\pm 0.38}$ | $0.53_{\pm 0.07}$ | $0.65_{\pm 0.09}$ | $0.90_{\pm 0.08}$ |
| TVAE | $0.89_{\pm 0.01}$ | $0.93_{\pm 0.01}$ | $0.96_{\pm 0.01}$ | $0.14_{\pm 0.02}$ | $0.00_{\pm 0.00}$ | $0.02_{\pm 0.01}$ | $0.71_{\pm 0.17}$ | $0.55_{\pm 0.13}$ | $0.58_{\pm 0.13}$ | $0.57_{\pm 0.25}$ |
| GOGGLE | $0.71_{\pm 0.19}$ | $0.77_{\pm 0.19}$ | $0.59_{\pm 0.40}$ | $0.18_{\pm 0.19}$ | $0.01_{\pm 0.00}$ | $24.60_{\pm 40.90}$ | $0.72_{\pm 0.38}$ | $0.67_{\pm 0.14}$ | $0.52_{\pm 0.06}$ | $0.17_{\pm 0.24}$ |
| CTGAN | $0.83_{\pm 0.05}$ | $0.90_{\pm 0.02}$ | $0.89_{\pm 0.07}$ | $0.10_{\pm 0.03}$ | $0.00_{\pm 0.00}$ | $0.21_{\pm 0.42}$ | $0.72_{\pm 0.30}$ | $0.67_{\pm 0.13}$ | $0.52_{\pm 0.05}$ | $0.15_{\pm 0.17}$ |
| NFlow | $0.83_{\pm 0.03}$ | $0.83_{\pm 0.05}$ | $0.88_{\pm 0.10}$ | $0.17_{\pm 0.10}$ | $0.02_{\pm 0.03}$ | $0.05_{\pm 0.05}$ | $0.36_{\pm 0.18}$ | $0.48_{\pm 0.05}$ | $0.50_{\pm 0.03}$ | $0.14_{\pm 0.18}$ |
| ARF | $0.89_{\pm 0.00}$ | $0.99_{\pm 0.00}$ | $1.00_{\pm 0.00}$ | $0.47_{\pm 0.01}$ | $0.00_{\pm 0.00}$ | $0.00_{\pm 0.00}$ | $0.61_{\pm 0.31}$ | $0.49_{\pm 0.05}$ | $0.64_{\pm 0.10}$ | $0.85_{\pm 0.10}$ |
| TabDDPM | $0.85_{\pm 0.05}$ | $0.92_{\pm 0.03}$ | $0.96_{\pm 0.01}$ | $0.38_{\pm 0.02}$ | $0.00_{\pm 0.00}$ | $0.04_{\pm 0.06}$ | $0.41_{\pm 0.26}$ | $0.53_{\pm 0.09}$ | $0.72_{\pm 0.10}$ | $0.82_{\pm 0.18}$ |
| TabSyn | $0.83_{\pm 0.08}$ | $0.86_{\pm 0.12}$ | $0.81_{\pm 0.20}$ | $0.19_{\pm 0.19}$ | $0.06_{\pm 0.08}$ | $1.84_{\pm 5.70}$ | $0.84_{\pm 0.20}$ | $0.69_{\pm 0.12}$ | $0.71_{\pm 0.11}$ | $0.78_{\pm 0.21}$ |
| TabDiff | $0.88_{\pm 0.03}$ | $0.83_{\pm 0.14}$ | $0.96_{\pm 0.02}$ | $0.21_{\pm 0.16}$ | $0.02_{\pm 0.04}$ | $0.04_{\pm 0.06}$ | $0.86_{\pm 0.17}$ | $0.67_{\pm 0.14}$ | $0.69_{\pm 0.12}$ | $0.81_{\pm 0.18}$ |
| TabEBM | $0.85_{\pm 0.05}$ | $0.85_{\pm 0.11}$ | $0.95_{\pm 0.01}$ | $0.18_{\pm 0.19}$ | $0.12_{\pm 0.12}$ | $0.08_{\pm 0.07}$ | $0.41_{\pm 0.34}$ | $0.58_{\pm 0.02}$ | $0.58_{\pm 0.05}$ | $0.44_{\pm 0.15}$ |
| NRGBoost | $0.83_{\pm 0.07}$ | $0.80_{\pm 0.15}$ | $0.92_{\pm 0.05}$ | $0.18_{\pm 0.19}$ | $0.07_{\pm 0.07}$ | $0.19_{\pm 0.15}$ | $0.72_{\pm 0.38}$ | $0.68_{\pm 0.13}$ | $0.52_{\pm 0.06}$ | $0.16_{\pm 0.25}$ |
| GReaT | $0.85_{\pm 0.05}$ | $0.89_{\pm 0.06}$ | $0.89_{\pm 0.08}$ | $0.21_{\pm 0.16}$ | $0.05_{\pm 0.04}$ | $0.09_{\pm 0.08}$ | $0.34_{\pm 0.28}$ | $0.51_{\pm 0.06}$ | $0.52_{\pm 0.06}$ | $0.19_{\pm 0.24}$ |

Table 42: **Raw benchmark results of 13 tabular generators on "MAGIC-IRRI" dataset.** We report the normalised mean ± std metric values across datasets. We highlight the **First**, **Second** and **Third** best performances for each metric. For visualisation, we abbreviate "conditional independence" as "CI". SMOTE generally achieves the highest performance in capturing local structure (i.e., local utility or local CI), while diffusion models typically excel at capturing global structure (i.e., global CI or global utility).

| Generator | Density Estimation | | | | Privacy Preservation | | ML Efficacy | Structural Fidelity | | |
| --- | --- | --- | --- | --- | --- | --- | --- | --- | --- | --- |
| | Shape ↑ | Trend ↑ | α-precision ↑ | β-recall ↑ | DCR ↑ | δ-Presence ↑ | Local utility ↑ | Local CI ↑ | Global CI ↑ | Global utility ↑ |
| $\mathcal{D}_{\text{ref}}$ | $1.00_{\pm 0.00}$ | $1.00_{\pm 0.00}$ | $1.00_{\pm 0.00}$ | $1.00_{\pm 0.00}$ | $0.00_{\pm 0.00}$ | $0.00_{\pm 0.00}$ | $1.00_{\pm 0.00}$ | $0.99_{\pm 0.02}$ | $0.99_{\pm 0.02}$ | $1.00_{\pm 0.00}$ |
| SMOTE | $0.96_{\pm 0.00}$ | $1.00_{\pm 0.00}$ | $0.36_{\pm 0.00}$ | $0.98_{\pm 0.00}$ | $0.40_{\pm 0.01}$ | $0.00_{\pm 0.00}$ | $0.98_{\pm 0.02}$ | $0.57_{\pm 0.03}$ | $0.54_{\pm 0.12}$ | $0.41_{\pm 0.29}$ |
| BN | $0.94_{\pm 0.01}$ | $0.99_{\pm 0.00}$ | $0.69_{\pm 0.10}$ | $0.51_{\pm 0.13}$ | $0.39_{\pm 0.09}$ | $0.54_{\pm 1.03}$ | $0.86_{\pm 0.12}$ | $0.52_{\pm 0.02}$ | $0.59_{\pm 0.06}$ | $0.66_{\pm 0.25}$ |
| TVAE | $0.91_{\pm 0.01}$ | $0.99_{\pm 0.00}$ | $0.51_{\pm 0.02}$ | $0.70_{\pm 0.02}$ | $0.53_{\pm 0.03}$ | $0.00_{\pm 0.00}$ | $0.98_{\pm 0.02}$ | $0.53_{\pm 0.04}$ | $0.58_{\pm 0.06}$ | $0.86_{\pm 0.11}$ |
| GOGGLE | $0.73_{\pm 0.23}$ | $0.99_{\pm 0.00}$ | $0.31_{\pm 0.33}$ | $0.55_{\pm 0.10}$ | $0.56_{\pm 0.14}$ | $6.46_{\pm 12.39}$ | $0.95_{\pm 0.07}$ | $0.54_{\pm 0.04}$ | $0.45_{\pm 0.09}$ | $0.33_{\pm 0.19}$ |
| CTGAN | $0.92_{\pm 0.02}$ | $0.99_{\pm 0.00}$ | $0.73_{\pm 0.10}$ | $0.52_{\pm 0.07}$ | $0.42_{\pm 0.04}$ | $0.01_{\pm 0.00}$ | $0.97_{\pm 0.03}$ | $0.54_{\pm 0.04}$ | $0.47_{\pm 0.06}$ | $0.43_{\pm 0.27}$ |
| NFlow | $0.90_{\pm 0.01}$ | $0.99_{\pm 0.00}$ | $0.27_{\pm 0.02}$ | $0.47_{\pm 0.02}$ | $0.46_{\pm 0.02}$ | $0.04_{\pm 0.02}$ | $0.83_{\pm 0.10}$ | $0.49_{\pm 0.01}$ | $0.46_{\pm 0.06}$ | $0.33_{\pm 0.19}$ |
| ARF | $0.99_{\pm 0.00}$ | $0.99_{\pm 0.00}$ | $0.83_{\pm 0.00}$ | $0.20_{\pm 0.00}$ | $0.44_{\pm 0.03}$ | $0.00_{\pm 0.00}$ | $0.88_{\pm 0.12}$ | $0.51_{\pm 0.01}$ | $0.57_{\pm 0.07}$ | $0.76_{\pm 0.16}$ |
| TabDDPM | $0.97_{\pm 0.02}$ | $1.00_{\pm 0.00}$ | $0.81_{\pm 0.20}$ | $0.51_{\pm 0.13}$ | $0.26_{\pm 0.18}$ | $0.00_{\pm 0.00}$ | $0.86_{\pm 0.12}$ | $0.51_{\pm 0.03}$ | $0.63_{\pm 0.06}$ | $0.83_{\pm 0.17}$ |
| TabSyn | $0.96_{\pm 0.01}$ | $1.00_{\pm 0.00}$ | $0.75_{\pm 0.14}$ | $0.54_{\pm 0.10}$ | $0.44_{\pm 0.03}$ | $0.01_{\pm 0.00}$ | $0.97_{\pm 0.03}$ | $0.55_{\pm 0.03}$ | $0.63_{\pm 0.06}$ | $0.83_{\pm 0.17}$ |
| TabDiff | $0.97_{\pm 0.02}$ | $1.00_{\pm 0.00}$ | $0.80_{\pm 0.19}$ | $0.51_{\pm 0.13}$ | $0.44_{\pm 0.01}$ | $0.00_{\pm 0.00}$ | $0.97_{\pm 0.03}$ | $0.55_{\pm 0.03}$ | $0.64_{\pm 0.05}$ | $0.82_{\pm 0.17}$ |
| TabEBM | $0.96_{\pm 0.01}$ | $0.99_{\pm 0.01}$ | $0.79_{\pm 0.18}$ | $0.32_{\pm 0.33}$ | $0.30_{\pm 0.15}$ | $0.01_{\pm 0.00}$ | $0.81_{\pm 0.14}$ | $0.52_{\pm 0.01}$ | $0.54_{\pm 0.03}$ | $0.37_{\pm 0.24}$ |
| NRGBoost | $0.97_{\pm 0.03}$ | $0.99_{\pm 0.00}$ | $0.77_{\pm 0.16}$ | $0.36_{\pm 0.29}$ | $0.41_{\pm 0.03}$ | $0.01_{\pm 0.01}$ | $0.91_{\pm 0.12}$ | $0.55_{\pm 0.04}$ | $0.48_{\pm 0.07}$ | $0.34_{\pm 0.19}$ |
| GReaT | $0.94_{\pm 0.01}$ | $0.99_{\pm 0.00}$ | $0.66_{\pm 0.05}$ | $0.55_{\pm 0.09}$ | $0.42_{\pm 0.02}$ | $0.55_{\pm 1.03}$ | $0.85_{\pm 0.11}$ | $0.50_{\pm 0.02}$ | $0.49_{\pm 0.08}$ | $0.41_{\pm 0.24}$ |

Table 43: **Raw benchmark results of 13 tabular generators on "MEHRA" dataset.** We report the normalised mean ± std metric values across datasets. We highlight the **First**, **Second** and **Third** best performances for each metric. For visualisation, we abbreviate "conditional independence" as "CI". SMOTE generally achieves the highest performance in capturing local structure (i.e., local utility or local CI), while diffusion models typically excel at capturing global structure (i.e., global CI or global utility).

| Generator | Density Estimation | | | | Privacy Preservation | | ML Efficacy | Structural Fidelity | | |
| --- | --- | --- | --- | --- | --- | --- | --- | --- | --- | --- |
| | Shape ↑ | Trend ↑ | α-precision ↑ | β-recall ↑ | DCR ↑ | δ-Presence ↑ | Local utility ↑ | Local CI ↑ | Global CI ↑ | Global utility ↑ |
| $\mathcal{D}_{\text{ref}}$ | $1.00_{\pm 0.00}$ | $1.00_{\pm 0.00}$ | $1.00_{\pm 0.00}$ | $1.00_{\pm 0.00}$ | $0.00_{\pm 0.00}$ | $0.00_{\pm 0.00}$ | $1.00_{\pm 0.00}$ | $0.96_{\pm 0.03}$ | $0.96_{\pm 0.01}$ | $1.00_{\pm 0.00}$ |
| SMOTE | $0.97_{\pm 0.00}$ | $0.94_{\pm 0.02}$ | $0.89_{\pm 0.00}$ | $0.81_{\pm 0.00}$ | $0.03_{\pm 0.01}$ | $0.00_{\pm 0.01}$ | $0.94_{\pm 0.06}$ | $0.55_{\pm 0.03}$ | $0.54_{\pm 0.05}$ | $0.42_{\pm 0.29}$ |
| BN | $0.88_{\pm 0.04}$ | $0.89_{\pm 0.03}$ | $0.91_{\pm 0.06}$ | $0.39_{\pm 0.10}$ | $0.08_{\pm 0.03}$ | $1.31_{\pm 2.58}$ | $0.66_{\pm 0.17}$ | $0.49_{\pm 0.04}$ | $0.55_{\pm 0.04}$ | $0.55_{\pm 0.27}$ |
| TVAE | $0.89_{\pm 0.01}$ | $0.87_{\pm 0.02}$ | $0.95_{\pm 0.02}$ | $0.43_{\pm 0.01}$ | $0.07_{\pm 0.02}$ | $0.22_{\pm 0.62}$ | $0.85_{\pm 0.09}$ | $0.55_{\pm 0.04}$ | $0.54_{\pm 0.04}$ | $0.62_{\pm 0.23}$ |
| GOGGLE | $0.70_{\pm 0.22}$ | $0.80_{\pm 0.11}$ | $0.66_{\pm 0.31}$ | $0.30_{\pm 0.26}$ | $0.10_{\pm 0.05}$ | $12.80_{\pm 21.05}$ | $0.86_{\pm 0.16}$ | $0.54_{\pm 0.04}$ | $0.51_{\pm 0.03}$ | $0.30_{\pm 0.17}$ |
| CTGAN | $0.83_{\pm 0.03}$ | $0.86_{\pm 0.01}$ | $0.97_{\pm 0.02}$ | $0.43_{\pm 0.03}$ | $0.05_{\pm 0.01}$ | $0.07_{\pm 0.16}$ | $0.86_{\pm 0.15}$ | $0.54_{\pm 0.04}$ | $0.50_{\pm 0.02}$ | $0.28_{\pm 0.15}$ |
| NFlow | $0.85_{\pm 0.01}$ | $0.84_{\pm 0.02}$ | $0.90_{\pm 0.08}$ | $0.37_{\pm 0.01}$ | $0.10_{\pm 0.02}$ | $0.63_{\pm 0.39}$ | $0.51_{\pm 0.08}$ | $0.48_{\pm 0.03}$ | $0.50_{\pm 0.01}$ | $0.21_{\pm 0.11}$ |
| ARF | $0.91_{\pm 0.00}$ | $0.91_{\pm 0.02}$ | $0.97_{\pm 0.00}$ | $0.38_{\pm 0.00}$ | $0.06_{\pm 0.01}$ | $0.00_{\pm 0.00}$ | $0.68_{\pm 0.18}$ | $0.48_{\pm 0.03}$ | $0.54_{\pm 0.04}$ | $0.65_{\pm 0.21}$ |
| TabDDPM | $0.88_{\pm 0.03}$ | $0.85_{\pm 0.07}$ | $0.91_{\pm 0.04}$ | $0.47_{\pm 0.08}$ | $0.05_{\pm 0.01}$ | $0.05_{\pm 0.07}$ | $0.61_{\pm 0.16}$ | $0.47_{\pm 0.03}$ | $0.57_{\pm 0.05}$ | $0.73_{\pm 0.26}$ |
| TabSyn | $0.84_{\pm 0.09}$ | $0.86_{\pm 0.06}$ | $0.89_{\pm 0.14}$ | $0.44_{\pm 0.12}$ | $0.07_{\pm 0.03}$ | $11.85_{\pm 22.46}$ | $0.86_{\pm 0.16}$ | $0.55_{\pm 0.03}$ | $0.57_{\pm 0.05}$ | $0.68_{\pm 0.33}$ |
| TabDiff | $0.86_{\pm 0.05}$ | $0.88_{\pm 0.03}$ | $0.95_{\pm 0.01}$ | $0.39_{\pm 0.17}$ | $0.08_{\pm 0.04}$ | $0.06_{\pm 0.07}$ | $0.84_{\pm 0.18}$ | $0.55_{\pm 0.04}$ | $0.57_{\pm 0.05}$ | $0.73_{\pm 0.26}$ |
| TabEBM | $0.85_{\pm 0.06}$ | $0.87_{\pm 0.04}$ | $0.96_{\pm 0.01}$ | $0.28_{\pm 0.28}$ | $0.11_{\pm 0.06}$ | $0.05_{\pm 0.07}$ | $0.64_{\pm 0.15}$ | $0.49_{\pm 0.04}$ | $0.53_{\pm 0.02}$ | $0.32_{\pm 0.19}$ |
| NRGBoost | $0.92_{\pm 0.02}$ | $0.91_{\pm 0.01}$ | $0.84_{\pm 0.14}$ | $0.43_{\pm 0.12}$ | $0.06_{\pm 0.01}$ | $0.06_{\pm 0.07}$ | $0.79_{\pm 0.26}$ | $0.54_{\pm 0.04}$ | $0.51_{\pm 0.02}$ | $0.27_{\pm 0.15}$ |
| GReaT | $0.83_{\pm 0.09}$ | $0.86_{\pm 0.05}$ | $0.77_{\pm 0.20}$ | $0.38_{\pm 0.18}$ | $0.08_{\pm 0.04}$ | $0.17_{\pm 0.19}$ | $0.55_{\pm 0.16}$ | $0.48_{\pm 0.03}$ | $0.50_{\pm 0.03}$ | $0.26_{\pm 0.15}$ |

## F.2 EVALUATION RESULTS FOR REAL-WORLD DATASETS

### F.2.1 CLASSIFICATION DATASETS

Table 44: **Raw benchmark results of 13 tabular generators on "Ada" dataset.** We report the normalised mean $\pm$ std metric values across datasets. We highlight the **First**, **Second** and **Third** best performances for each metric. For visualisation, we abbreviate "conditional independence" as "CI". SMOTE generally achieves the highest performance in capturing local structure (i.e., local utility or local CI), while diffusion models typically excel at capturing global structure (i.e., global CI or global utility).

| Generator | Density Estimation | | | | Privacy Preservation | | ML Efficacy | Structural Fidelity |
|---|---|---|---|---|---|---|---|---|
| | Shape $\uparrow$ | Trend $\uparrow$ | $\alpha$-precision $\uparrow$ | $\beta$-recall $\uparrow$ | DCR $\uparrow$ | $\delta$-Presence $\uparrow$ | Local utility $\uparrow$ | Global utility $\uparrow$ |
| $\mathcal{D}_{\text{ref}}$ | $1.00_{\pm 0.00}$ | $1.00_{\pm 0.00}$ | $1.00_{\pm 0.00}$ | $1.00_{\pm 0.00}$ | $0.00_{\pm 0.00}$ | $0.00_{\pm 0.00}$ | $1.00_{\pm 0.00}$ | $1.00_{\pm 0.00}$ |
| SMOTE | $0.24_{\pm 0.01}$ | $0.99_{\pm 0.00}$ | $0.89_{\pm 0.01}$ | $0.76_{\pm 0.01}$ | $0.04_{\pm 0.03}$ | $0.00_{\pm 0.00}$ | $1.01_{\pm 0.05}$ | $0.47_{\pm 0.38}$ |
| BN | $0.31_{\pm 0.07}$ | $0.97_{\pm 0.02}$ | $0.83_{\pm 0.12}$ | $0.25_{\pm 0.11}$ | $0.25_{\pm 0.07}$ | $0.16_{\pm 0.27}$ | $0.86_{\pm 0.13}$ | $0.36_{\pm 0.28}$ |
| TVAE | $0.23_{\pm 0.01}$ | $0.98_{\pm 0.00}$ | $0.70_{\pm 0.03}$ | $0.22_{\pm 0.01}$ | $0.28_{\pm 0.02}$ | $0.05_{\pm 0.06}$ | $0.96_{\pm 0.09}$ | $0.77_{\pm 0.13}$ |
| GOGGLE | $0.36_{\pm 0.02}$ | $0.97_{\pm 0.02}$ | $0.78_{\pm 0.12}$ | $0.31_{\pm 0.08}$ | $0.21_{\pm 0.02}$ | $0.26_{\pm 0.30}$ | $0.88_{\pm 0.12}$ | $0.36_{\pm 0.27}$ |
| CTGAN | $0.22_{\pm 0.01}$ | $0.97_{\pm 0.00}$ | $0.90_{\pm 0.05}$ | $0.15_{\pm 0.03}$ | $0.22_{\pm 0.03}$ | $0.03_{\pm 0.01}$ | $0.95_{\pm 0.11}$ | $0.29_{\pm 0.26}$ |
| NFlow | $0.23_{\pm 0.02}$ | $0.97_{\pm 0.00}$ | $0.87_{\pm 0.07}$ | $0.07_{\pm 0.02}$ | $0.20_{\pm 0.10}$ | $0.02_{\pm 0.01}$ | $0.87_{\pm 0.12}$ | $0.53_{\pm 0.21}$ |
| ARF | $0.24_{\pm 0.01}$ | $0.98_{\pm 0.00}$ | $0.96_{\pm 0.00}$ | $0.24_{\pm 0.01}$ | $0.26_{\pm 0.05}$ | $0.00_{\pm 0.00}$ | $0.94_{\pm 0.09}$ | $0.73_{\pm 0.17}$ |
| TabDDPM | $0.35_{\pm 0.02}$ | $0.93_{\pm 0.06}$ | $0.50_{\pm 0.41}$ | $0.21_{\pm 0.19}$ | $0.10_{\pm 0.09}$ | $0.22_{\pm 0.38}$ | $0.88_{\pm 0.12}$ | $0.74_{\pm 0.16}$ |
| TabSyn | $0.29_{\pm 0.07}$ | $0.91_{\pm 0.12}$ | $0.51_{\pm 0.43}$ | $0.19_{\pm 0.20}$ | $0.22_{\pm 0.11}$ | $0.32_{\pm 0.50}$ | $0.98_{\pm 0.07}$ | $0.73_{\pm 0.16}$ |
| TabDiff | $0.44_{\pm 0.09}$ | $0.96_{\pm 0.03}$ | $0.80_{\pm 0.15}$ | $0.19_{\pm 0.20}$ | $0.33_{\pm 0.16}$ | $0.21_{\pm 0.30}$ | $0.98_{\pm 0.07}$ | $0.70_{\pm 0.18}$ |
| TabEBM | $0.43_{\pm 0.08}$ | $0.98_{\pm 0.00}$ | $0.93_{\pm 0.04}$ | $0.27_{\pm 0.13}$ | $0.24_{\pm 0.07}$ | $0.01_{\pm 0.00}$ | $0.98_{\pm 0.07}$ | $0.36_{\pm 0.27}$ |
| NRGBoost | $0.30_{\pm 0.06}$ | $0.96_{\pm 0.03}$ | $0.45_{\pm 0.47}$ | $0.19_{\pm 0.20}$ | $0.29_{\pm 0.13}$ | $2.07_{\pm 2.80}$ | $0.98_{\pm 0.07}$ | $0.20_{\pm 0.21}$ |
| GReaT | $0.36_{\pm 0.02}$ | $0.97_{\pm 0.02}$ | $0.78_{\pm 0.12}$ | $0.31_{\pm 0.08}$ | $0.21_{\pm 0.04}$ | $0.26_{\pm 0.30}$ | $0.88_{\pm 0.12}$ | $0.36_{\pm 0.27}$ |

Table 45: **Raw benchmark results of 13 tabular generators on "Characters" dataset.** We report the normalised mean $\pm$ std metric values across datasets. We highlight the **First**, **Second** and **Third** best performances for each metric. For visualisation, we abbreviate "conditional independence" as "CI". SMOTE generally achieves the highest performance in capturing local structure (i.e., local utility or local CI), while diffusion models typically excel at capturing global structure (i.e., global CI or global utility).

| Generator | Density Estimation | | | | Privacy Preservation | | ML Efficacy | Structural Fidelity |
|---|---|---|---|---|---|---|---|---|
| | Shape $\uparrow$ | Trend $\uparrow$ | $\alpha$-precision $\uparrow$ | $\beta$-recall $\uparrow$ | DCR $\uparrow$ | $\delta$-Presence $\uparrow$ | Local utility $\uparrow$ | Global utility $\uparrow$ |
| $\mathcal{D}_{\text{ref}}$ | $1.00_{\pm 0.00}$ | $1.00_{\pm 0.00}$ | $1.00_{\pm 0.00}$ | $1.00_{\pm 0.00}$ | $0.00_{\pm 0.00}$ | $0.00_{\pm 0.00}$ | $1.00_{\pm 0.00}$ | $1.00_{\pm 0.00}$ |
| SMOTE | $0.83_{\pm 0.00}$ | $0.99_{\pm 0.00}$ | $0.99_{\pm 0.01}$ | $0.41_{\pm 0.03}$ | $0.06_{\pm 0.01}$ | $0.00_{\pm 0.00}$ | $0.97_{\pm 0.05}$ | $0.40_{\pm 0.42}$ |
| BN | $0.85_{\pm 0.02}$ | $0.93_{\pm 0.00}$ | $0.98_{\pm 0.00}$ | $0.01_{\pm 0.00}$ | $0.32_{\pm 0.02}$ | $0.01_{\pm 0.00}$ | $0.30_{\pm 0.13}$ | $0.05_{\pm 0.05}$ |
| TVAE | $0.82_{\pm 0.02}$ | $0.91_{\pm 0.01}$ | $0.95_{\pm 0.01}$ | $0.04_{\pm 0.00}$ | $0.31_{\pm 0.01}$ | $0.01_{\pm 0.00}$ | $0.77_{\pm 0.14}$ | $0.41_{\pm 0.35}$ |
| GOGGLE | $0.85_{\pm 0.01}$ | $0.93_{\pm 0.01}$ | $0.96_{\pm 0.01}$ | $0.19_{\pm 0.05}$ | $0.23_{\pm 0.02}$ | $0.01_{\pm 0.00}$ | $0.40_{\pm 0.23}$ | $0.19_{\pm 0.19}$ |
| CTGAN | $0.80_{\pm 0.02}$ | $0.93_{\pm 0.01}$ | $0.94_{\pm 0.03}$ | $0.02_{\pm 0.00}$ | $0.30_{\pm 0.03}$ | $0.02_{\pm 0.01}$ | $0.75_{\pm 0.26}$ | $0.07_{\pm 0.07}$ |
| NFlow | $0.82_{\pm 0.02}$ | $0.88_{\pm 0.01}$ | $0.94_{\pm 0.04}$ | $0.00_{\pm 0.00}$ | $0.41_{\pm 0.03}$ | $0.02_{\pm 0.01}$ | $0.20_{\pm 0.03}$ | $0.02_{\pm 0.02}$ |
| ARF | $0.85_{\pm 0.00}$ | $0.89_{\pm 0.01}$ | $0.99_{\pm 0.00}$ | $0.11_{\pm 0.00}$ | $0.11_{\pm 0.02}$ | $0.00_{\pm 0.00}$ | $0.82_{\pm 0.03}$ | $0.49_{\pm 0.04}$ |
| TabDDPM | $0.84_{\pm 0.01}$ | $0.95_{\pm 0.02}$ | $0.98_{\pm 0.01}$ | $0.17_{\pm 0.06}$ | $0.16_{\pm 0.08}$ | $0.01_{\pm 0.00}$ | $0.46_{\pm 0.30}$ | $0.76_{\pm 0.27}$ |
| TabSyn | $0.84_{\pm 0.02}$ | $0.92_{\pm 0.02}$ | $0.95_{\pm 0.03}$ | $0.14_{\pm 0.09}$ | $0.20_{\pm 0.02}$ | $0.01_{\pm 0.00}$ | $0.79_{\pm 0.22}$ | $0.69_{\pm 0.32}$ |
| TabDiff | $0.86_{\pm 0.01}$ | $0.90_{\pm 0.04}$ | $0.96_{\pm 0.01}$ | $0.12_{\pm 0.12}$ | $0.23_{\pm 0.07}$ | $0.01_{\pm 0.00}$ | $0.80_{\pm 0.21}$ | $0.66_{\pm 0.36}$ |
| TabEBM | $0.88_{\pm 0.03}$ | $0.95_{\pm 0.02}$ | $0.98_{\pm 0.01}$ | $0.16_{\pm 0.07}$ | $0.31_{\pm 0.10}$ | $0.01_{\pm 0.00}$ | $0.88_{\pm 0.15}$ | $0.25_{\pm 0.27}$ |
| NRGBoost | $0.84_{\pm 0.02}$ | $0.92_{\pm 0.01}$ | $0.97_{\pm 0.01}$ | $0.12_{\pm 0.12}$ | $0.28_{\pm 0.07}$ | $0.01_{\pm 0.00}$ | $0.77_{\pm 0.24}$ | $0.11_{\pm 0.16}$ |
| GReaT | $0.79_{\pm 0.07}$ | $0.89_{\pm 0.04}$ | $0.89_{\pm 0.09}$ | $0.12_{\pm 0.12}$ | $0.31_{\pm 0.11}$ | $0.04_{\pm 0.03}$ | $0.29_{\pm 0.21}$ | $0.10_{\pm 0.17}$ |

Table 46: **Raw benchmark results of 13 tabular generators on "Credit-g" dataset.** We report the normalised mean ± std metric values across datasets. We highlight the First, Second and Third best performances for each metric. For visualisation, we abbreviate "conditional independence" as "CI". SMOTE generally achieves the highest performance in capturing local structure (i.e., local utility or local CI), while diffusion models typically excel at capturing global structure (i.e., global CI or global utility).

| Generator | Density Estimation | | | | Privacy Preservation | | ML Efficacy | Structural Fidelity |
|---|---|---|---|---|---|---|---|---|
| | Shape ↑ | Trend ↑ | $\alpha$-precision ↑ | $\beta$-recall ↑ | DCR ↑ | $\delta$-Presence ↑ | Local utility ↑ | Global utility ↑ |
| $\mathcal{D}_{\text{ref}}$ | $1.00_{\pm 0.00}$ | $1.00_{\pm 0.00}$ | $1.00_{\pm 0.00}$ | $1.00_{\pm 0.00}$ | $0.00_{\pm 0.00}$ | $0.00_{\pm 0.00}$ | $1.00_{\pm 0.00}$ | $1.00_{\pm 0.00}$ |
| SMOTE | $0.95_{\pm 0.00}$ | $0.90_{\pm 0.01}$ | $0.87_{\pm 0.02}$ | $0.79_{\pm 0.02}$ | $0.19_{\pm 0.02}$ | $0.00_{\pm 0.00}$ | $1.17_{\pm 0.30}$ | $0.42_{\pm 0.30}$ |
| BN | $0.97_{\pm 0.00}$ | $0.95_{\pm 0.00}$ | $0.97_{\pm 0.01}$ | $0.68_{\pm 0.02}$ | $0.06_{\pm 0.00}$ | $0.00_{\pm 0.00}$ | $0.92_{\pm 0.09}$ | $0.39_{\pm 0.28}$ |
| TVAE | $0.93_{\pm 0.01}$ | $0.86_{\pm 0.02}$ | $0.80_{\pm 0.04}$ | $0.48_{\pm 0.02}$ | $0.55_{\pm 0.04}$ | $0.02_{\pm 0.01}$ | $1.11_{\pm 0.34}$ | $0.47_{\pm 0.14}$ |
| GOGGLE | $0.79_{\pm 0.17}$ | $0.67_{\pm 0.25}$ | $0.55_{\pm 0.42}$ | $0.35_{\pm 0.24}$ | $0.36_{\pm 0.06}$ | $0.37_{\pm 0.49}$ | $1.05_{\pm 0.36}$ | $0.33_{\pm 0.21}$ |
| CTGAN | $0.80_{\pm 0.06}$ | $0.72_{\pm 0.09}$ | $0.83_{\pm 0.12}$ | $0.27_{\pm 0.07}$ | $0.50_{\pm 0.05}$ | $0.21_{\pm 0.16}$ | $1.13_{\pm 0.32}$ | $0.21_{\pm 0.10}$ |
| NFlow | $0.90_{\pm 0.01}$ | $0.84_{\pm 0.01}$ | $0.84_{\pm 0.08}$ | $0.27_{\pm 0.04}$ | $0.50_{\pm 0.07}$ | $0.02_{\pm 0.01}$ | $0.85_{\pm 0.04}$ | $0.24_{\pm 0.11}$ |
| ARF | $0.97_{\pm 0.00}$ | $0.86_{\pm 0.01}$ | $0.98_{\pm 0.01}$ | $0.45_{\pm 0.03}$ | $0.53_{\pm 0.05}$ | $0.00_{\pm 0.00}$ | $0.87_{\pm 0.04}$ | $0.43_{\pm 0.06}$ |
| TabDDPM | $0.75_{\pm 0.19}$ | $0.63_{\pm 0.25}$ | $0.45_{\pm 0.47}$ | $0.28_{\pm 0.29}$ | $0.17_{\pm 0.17}$ | $0.15_{\pm 0.13}$ | $0.91_{\pm 0.09}$ | $0.55_{\pm 0.29}$ |
| TabSyn | $0.86_{\pm 0.08}$ | $0.76_{\pm 0.13}$ | $0.67_{\pm 0.24}$ | $0.41_{\pm 0.15}$ | $0.44_{\pm 0.12}$ | $0.03_{\pm 0.02}$ | $1.15_{\pm 0.31}$ | $0.64_{\pm 0.17}$ |
| TabDiff | $0.90_{\pm 0.03}$ | $0.74_{\pm 0.14}$ | $0.91_{\pm 0.04}$ | $0.40_{\pm 0.17}$ | $0.43_{\pm 0.11}$ | $0.02_{\pm 0.02}$ | $1.15_{\pm 0.31}$ | $0.62_{\pm 0.19}$ |
| TabEBM | $0.92_{\pm 0.01}$ | $0.84_{\pm 0.03}$ | $0.93_{\pm 0.04}$ | $0.50_{\pm 0.06}$ | $0.38_{\pm 0.05}$ | $0.02_{\pm 0.02}$ | $1.15_{\pm 0.31}$ | $0.33_{\pm 0.22}$ |
| NRGBoost | $0.87_{\pm 0.07}$ | $0.80_{\pm 0.08}$ | $0.75_{\pm 0.15}$ | $0.34_{\pm 0.23}$ | $0.38_{\pm 0.06}$ | $0.03_{\pm 0.02}$ | $1.15_{\pm 0.31}$ | $0.25_{\pm 0.16}$ |
| GReaT | $0.88_{\pm 0.06}$ | $0.80_{\pm 0.08}$ | $0.64_{\pm 0.27}$ | $0.43_{\pm 0.14}$ | $0.42_{\pm 0.09}$ | $0.10_{\pm 0.09}$ | $0.91_{\pm 0.09}$ | $0.26_{\pm 0.16}$ |

Table 47: **Raw benchmark results of 13 tabular generators on "Electricity" dataset.** We report the normalised mean ± std metric values across datasets. We highlight the First, Second and Third best performances for each metric. For visualisation, we abbreviate "conditional independence" as "CI". SMOTE generally achieves the highest performance in capturing local structure (i.e., local utility or local CI), while diffusion models typically excel at capturing global structure (i.e., global CI or global utility).

| Generator | Density Estimation | | | | Privacy Preservation | | ML Efficacy | Structural Fidelity |
|---|---|---|---|---|---|---|---|---|
| | Shape ↑ | Trend ↑ | $\alpha$-precision ↑ | $\beta$-recall ↑ | DCR ↑ | $\delta$-Presence ↑ | Local utility ↑ | Global utility ↑ |
| $\mathcal{D}_{\text{ref}}$ | $1.00_{\pm 0.00}$ | $1.00_{\pm 0.00}$ | $1.00_{\pm 0.00}$ | $1.00_{\pm 0.00}$ | $0.00_{\pm 0.00}$ | $0.00_{\pm 0.00}$ | $1.00_{\pm 0.00}$ | $1.00_{\pm 0.00}$ |
| SMOTE | $0.86_{\pm 0.00}$ | $0.99_{\pm 0.00}$ | $0.99_{\pm 0.00}$ | $0.78_{\pm 0.00}$ | $0.01_{\pm 0.00}$ | $0.00_{\pm 0.01}$ | $0.98_{\pm 0.02}$ | $0.41_{\pm 0.41}$ |
| BN | $0.93_{\pm 0.00}$ | $0.97_{\pm 0.00}$ | $0.98_{\pm 0.00}$ | $0.21_{\pm 0.00}$ | $0.05_{\pm 0.03}$ | $0.01_{\pm 0.01}$ | $0.75_{\pm 0.11}$ | $0.23_{\pm 0.25}$ |
| TVAE | $0.89_{\pm 0.01}$ | $0.92_{\pm 0.03}$ | $0.96_{\pm 0.02}$ | $0.20_{\pm 0.00}$ | $0.09_{\pm 0.03}$ | $0.21_{\pm 0.23}$ | $0.90_{\pm 0.06}$ | $0.56_{\pm 0.28}$ |
| GOGGLE | $0.86_{\pm 0.03}$ | $0.91_{\pm 0.02}$ | $0.93_{\pm 0.04}$ | $0.33_{\pm 0.07}$ | $0.07_{\pm 0.02}$ | $0.43_{\pm 0.76}$ | $0.76_{\pm 0.12}$ | $0.27_{\pm 0.26}$ |
| CTGAN | $0.86_{\pm 0.01}$ | $0.92_{\pm 0.01}$ | $0.96_{\pm 0.03}$ | $0.19_{\pm 0.01}$ | $0.02_{\pm 0.00}$ | $0.01_{\pm 0.01}$ | $0.92_{\pm 0.08}$ | $0.21_{\pm 0.24}$ |
| NFlow | $0.84_{\pm 0.02}$ | $0.85_{\pm 0.02}$ | $0.91_{\pm 0.05}$ | $0.09_{\pm 0.02}$ | $0.14_{\pm 0.06}$ | $0.03_{\pm 0.01}$ | $0.75_{\pm 0.03}$ | $0.25_{\pm 0.22}$ |
| ARF | $0.86_{\pm 0.00}$ | $0.81_{\pm 0.04}$ | $0.95_{\pm 0.00}$ | $0.26_{\pm 0.01}$ | $0.02_{\pm 0.01}$ | $0.00_{\pm 0.00}$ | $0.90_{\pm 0.01}$ | $0.62_{\pm 0.13}$ |
| TabDDPM | $0.87_{\pm 0.02}$ | $0.95_{\pm 0.03}$ | $0.98_{\pm 0.01}$ | $0.32_{\pm 0.07}$ | $0.03_{\pm 0.02}$ | $0.03_{\pm 0.03}$ | $0.77_{\pm 0.13}$ | $0.76_{\pm 0.23}$ |
| TabSyn | $0.54_{\pm 0.37}$ | $0.73_{\pm 0.20}$ | $0.48_{\pm 0.51}$ | $0.20_{\pm 0.21}$ | $0.12_{\pm 0.14}$ | $4.77_{\pm 9.27}$ | $0.86_{\pm 0.18}$ | $0.60_{\pm 0.42}$ |
| TabDiff | $0.88_{\pm 0.01}$ | $0.88_{\pm 0.05}$ | $0.96_{\pm 0.01}$ | $0.23_{\pm 0.17}$ | $0.06_{\pm 0.04}$ | $0.03_{\pm 0.03}$ | $0.93_{\pm 0.07}$ | $0.75_{\pm 0.24}$ |
| TabEBM | $0.90_{\pm 0.01}$ | $0.92_{\pm 0.03}$ | $0.97_{\pm 0.01}$ | $0.21_{\pm 0.19}$ | $0.21_{\pm 0.18}$ | $0.03_{\pm 0.03}$ | $0.93_{\pm 0.07}$ | $0.26_{\pm 0.26}$ |
| NRGBoost | $0.87_{\pm 0.02}$ | $0.92_{\pm 0.01}$ | $0.97_{\pm 0.00}$ | $0.22_{\pm 0.18}$ | $0.04_{\pm 0.01}$ | $0.03_{\pm 0.03}$ | $0.92_{\pm 0.08}$ | $0.22_{\pm 0.21}$ |
| GReaT | $0.86_{\pm 0.03}$ | $0.91_{\pm 0.02}$ | $0.93_{\pm 0.04}$ | $0.33_{\pm 0.07}$ | $0.07_{\pm 0.02}$ | $0.43_{\pm 0.76}$ | $0.76_{\pm 0.12}$ | $0.27_{\pm 0.26}$ |

Table 48: **Raw benchmark results of 13 tabular generators on "Higgs" dataset.** We report the normalised mean ± std metric values across datasets. We highlight the First, Second and Third best performances for each metric. For visualisation, we abbreviate "conditional independence" as "CI". SMOTE generally achieves the highest performance in capturing local structure (i.e., local utility or local CI), while diffusion models typically excel at capturing global structure (i.e., global CI or global utility).

| Generator | Density Estimation | | | | Privacy Preservation | | ML Efficacy | Structural Fidelity |
| --- | --- | --- | --- | --- | --- | --- | --- | --- |
| | Shape ↑ | Trend ↑ | $\alpha$-precision ↑ | $\beta$-recall ↑ | DCR ↑ | $\delta$-Presence ↑ | Local utility ↑ | Global utility ↑ |
| $\mathcal{D}_{\text{ref}}$ | $1.00_{\pm 0.00}$ | $1.00_{\pm 0.00}$ | $1.00_{\pm 0.00}$ | $1.00_{\pm 0.00}$ | $0.00_{\pm 0.00}$ | $0.00_{\pm 0.00}$ | $1.00_{\pm 0.00}$ | $1.00_{\pm 0.00}$ |
| SMOTE | $0.90_{\pm 0.00}$ | $0.99_{\pm 0.00}$ | $0.76_{\pm 0.00}$ | $0.82_{\pm 0.00}$ | $0.12_{\pm 0.03}$ | $0.00_{\pm 0.00}$ | $0.99_{\pm 0.01}$ | $0.45_{\pm 0.39}$ |
| BN | $0.92_{\pm 0.00}$ | $0.99_{\pm 0.00}$ | $0.98_{\pm 0.01}$ | $0.29_{\pm 0.00}$ | $0.06_{\pm 0.01}$ | $0.00_{\pm 0.00}$ | $0.85_{\pm 0.09}$ | $0.24_{\pm 0.17}$ |
| TVAE | $0.86_{\pm 0.00}$ | $0.97_{\pm 0.00}$ | $0.92_{\pm 0.01}$ | $0.37_{\pm 0.01}$ | $0.25_{\pm 0.04}$ | $0.12_{\pm 0.14}$ | $0.93_{\pm 0.04}$ | $0.63_{\pm 0.22}$ |
| GOGGLE | $0.90_{\pm 0.01}$ | $0.97_{\pm 0.00}$ | $0.90_{\pm 0.01}$ | $0.40_{\pm 0.07}$ | $0.13_{\pm 0.02}$ | $0.12_{\pm 0.19}$ | $0.84_{\pm 0.08}$ | $0.29_{\pm 0.22}$ |
| CTGAN | $0.85_{\pm 0.01}$ | $0.97_{\pm 0.00}$ | $0.95_{\pm 0.02}$ | $0.32_{\pm 0.02}$ | $0.11_{\pm 0.02}$ | $0.09_{\pm 0.08}$ | $0.95_{\pm 0.05}$ | $0.23_{\pm 0.17}$ |
| NFlow | $0.83_{\pm 0.01}$ | $0.95_{\pm 0.00}$ | $0.87_{\pm 0.08}$ | $0.23_{\pm 0.02}$ | $0.18_{\pm 0.05}$ | $0.69_{\pm 1.20}$ | $0.77_{\pm 0.04}$ | $0.16_{\pm 0.09}$ |
| ARF | $0.88_{\pm 0.00}$ | $0.95_{\pm 0.00}$ | $0.91_{\pm 0.00}$ | $0.26_{\pm 0.00}$ | $0.13_{\pm 0.02}$ | $0.00_{\pm 0.00}$ | $0.89_{\pm 0.01}$ | $0.50_{\pm 0.06}$ |
| TabDDPM | $0.92_{\pm 0.03}$ | $0.97_{\pm 0.00}$ | $0.93_{\pm 0.03}$ | $0.42_{\pm 0.05}$ | $0.06_{\pm 0.06}$ | $0.10_{\pm 0.19}$ | $0.85_{\pm 0.08}$ | $0.80_{\pm 0.22}$ |
| TabSyn | $0.91_{\pm 0.02}$ | $0.97_{\pm 0.00}$ | $0.94_{\pm 0.03}$ | $0.38_{\pm 0.10}$ | $0.14_{\pm 0.04}$ | $0.10_{\pm 0.19}$ | $0.95_{\pm 0.05}$ | $0.76_{\pm 0.24}$ |
| TabDiff | $0.89_{\pm 0.00}$ | $0.96_{\pm 0.02}$ | $0.85_{\pm 0.07}$ | $0.27_{\pm 0.22}$ | $0.20_{\pm 0.10}$ | $0.11_{\pm 0.19}$ | $0.95_{\pm 0.06}$ | $0.70_{\pm 0.31}$ |
| TabEBM | $0.91_{\pm 0.02}$ | $0.97_{\pm 0.00}$ | $0.93_{\pm 0.03}$ | $0.25_{\pm 0.24}$ | $0.26_{\pm 0.15}$ | $0.10_{\pm 0.19}$ | $0.95_{\pm 0.05}$ | $0.22_{\pm 0.17}$ |
| NRGBoost | $0.91_{\pm 0.01}$ | $0.97_{\pm 0.01}$ | $0.78_{\pm 0.13}$ | $0.25_{\pm 0.23}$ | $0.11_{\pm 0.02}$ | $0.12_{\pm 0.19}$ | $0.95_{\pm 0.06}$ | $0.18_{\pm 0.18}$ |
| GReaT | $0.90_{\pm 0.01}$ | $0.97_{\pm 0.00}$ | $0.90_{\pm 0.01}$ | $0.40_{\pm 0.07}$ | $0.13_{\pm 0.02}$ | $0.12_{\pm 0.19}$ | $0.84_{\pm 0.08}$ | $0.29_{\pm 0.22}$ |

Table 49: **Raw benchmark results of 13 tabular generators on "Jasmine" dataset.** We report the normalised mean ± std metric values across datasets. We highlight the First, Second and Third best performances for each metric. For visualisation, we abbreviate "conditional independence" as "CI". SMOTE generally achieves the highest performance in capturing local structure (i.e., local utility or local CI), while diffusion models typically excel at capturing global structure (i.e., global CI or global utility).

| Generator | Density Estimation | | | | Privacy Preservation | | ML Efficacy | Structural Fidelity |
| --- | --- | --- | --- | --- | --- | --- | --- | --- |
| | Shape ↑ | Trend ↑ | $\alpha$-precision ↑ | $\beta$-recall ↑ | DCR ↑ | $\delta$-Presence ↑ | Local utility ↑ | Global utility ↑ |
| $\mathcal{D}_{\text{ref}}$ | $1.00_{\pm 0.00}$ | $1.00_{\pm 0.00}$ | $1.00_{\pm 0.00}$ | $1.00_{\pm 0.00}$ | $0.00_{\pm 0.00}$ | $0.00_{\pm 0.00}$ | $1.00_{\pm 0.00}$ | $1.00_{\pm 0.00}$ |
| SMOTE | $0.98_{\pm 0.00}$ | $0.98_{\pm 0.00}$ | $0.86_{\pm 0.01}$ | $0.82_{\pm 0.01}$ | $0.06_{\pm 0.01}$ | $0.00_{\pm 0.00}$ | $0.98_{\pm 0.02}$ | $0.46_{\pm 0.18}$ |
| BN | $0.96_{\pm 0.03}$ | $0.94_{\pm 0.06}$ | $0.84_{\pm 0.13}$ | $0.36_{\pm 0.07}$ | $0.39_{\pm 0.11}$ | $0.13_{\pm 0.20}$ | $0.91_{\pm 0.06}$ | $0.42_{\pm 0.13}$ |
| TVAE | $0.96_{\pm 0.00}$ | $0.94_{\pm 0.01}$ | $0.83_{\pm 0.02}$ | $0.28_{\pm 0.02}$ | $0.49_{\pm 0.04}$ | $0.14_{\pm 0.05}$ | $0.97_{\pm 0.02}$ | $0.47_{\pm 0.10}$ |
| GOGGLE | $0.95_{\pm 0.03}$ | $0.91_{\pm 0.04}$ | $0.79_{\pm 0.10}$ | $0.34_{\pm 0.07}$ | $0.31_{\pm 0.04}$ | $0.18_{\pm 0.18}$ | $0.90_{\pm 0.06}$ | $0.40_{\pm 0.11}$ |
| CTGAN | $0.94_{\pm 0.03}$ | $0.92_{\pm 0.04}$ | $0.84_{\pm 0.15}$ | $0.18_{\pm 0.08}$ | $0.36_{\pm 0.07}$ | $0.04_{\pm 0.05}$ | $0.96_{\pm 0.04}$ | $0.36_{\pm 0.08}$ |
| NFlow | $0.95_{\pm 0.01}$ | $0.91_{\pm 0.01}$ | $0.77_{\pm 0.05}$ | $0.01_{\pm 0.00}$ | $0.31_{\pm 0.05}$ | $0.03_{\pm 0.01}$ | $0.85_{\pm 0.04}$ | $0.31_{\pm 0.04}$ |
| ARF | $0.99_{\pm 0.00}$ | $0.90_{\pm 0.00}$ | $0.93_{\pm 0.01}$ | $0.21_{\pm 0.02}$ | $0.37_{\pm 0.06}$ | $0.00_{\pm 0.00}$ | $0.94_{\pm 0.02}$ | $0.46_{\pm 0.05}$ |
| TabDDPM | $0.81_{\pm 0.17}$ | $0.72_{\pm 0.24}$ | $0.44_{\pm 0.46}$ | $0.21_{\pm 0.22}$ | $0.40_{\pm 0.13}$ | $1.24_{\pm 1.45}$ | $0.90_{\pm 0.06}$ | $0.59_{\pm 0.16}$ |
| TabSyn | $0.91_{\pm 0.07}$ | $0.83_{\pm 0.14}$ | $0.58_{\pm 0.37}$ | $0.21_{\pm 0.22}$ | $0.40_{\pm 0.13}$ | $0.07_{\pm 0.15}$ | $0.97_{\pm 0.03}$ | $0.61_{\pm 0.14}$ |
| TabDiff | $0.93_{\pm 0.06}$ | $0.87_{\pm 0.10}$ | $0.63_{\pm 0.40}$ | $0.24_{\pm 0.18}$ | $0.37_{\pm 0.10}$ | $0.39_{\pm 0.69}$ | $0.97_{\pm 0.03}$ | $0.61_{\pm 0.14}$ |
| TabEBM | $0.98_{\pm 0.01}$ | $0.96_{\pm 0.02}$ | $0.92_{\pm 0.06}$ | $0.41_{\pm 0.01}$ | $0.37_{\pm 0.09}$ | $0.03_{\pm 0.04}$ | $0.97_{\pm 0.03}$ | $0.42_{\pm 0.13}$ |
| NRGBoost | $0.96_{\pm 0.01}$ | $0.94_{\pm 0.01}$ | $0.87_{\pm 0.03}$ | $0.21_{\pm 0.22}$ | $0.23_{\pm 0.07}$ | $0.09_{\pm 0.08}$ | $0.97_{\pm 0.03}$ | $0.36_{\pm 0.12}$ |
| GReaT | $0.95_{\pm 0.03}$ | $0.91_{\pm 0.04}$ | $0.79_{\pm 0.10}$ | $0.34_{\pm 0.07}$ | $0.31_{\pm 0.04}$ | $0.18_{\pm 0.18}$ | $0.90_{\pm 0.06}$ | $0.40_{\pm 0.11}$ |

Table 50: **Raw benchmark results of 13 tabular generators on "Nomao" dataset.** We report the normalised mean ± std metric values across datasets. We highlight the First, Second and Third best performances for each metric. For visualisation, we abbreviate "conditional independence" as "CI". SMOTE generally achieves the highest performance in capturing local structure (i.e., local utility or local CI), while diffusion models typically excel at capturing global structure (i.e., global CI or global utility).

| Generator | Density Estimation | | | | Privacy Preservation | | ML Efficacy | Structural Fidelity |
|---|---|---|---|---|---|---|---|---|
| | Shape ↑ | Trend ↑ | $\alpha$-precision ↑ | $\beta$-recall ↑ | DCR ↑ | $\delta$-Presence ↑ | Local utility ↑ | Global utility ↑ |
| $\mathcal{D}_{\text{ref}}$ | $1.00_{\pm 0.00}$ | $1.00_{\pm 0.00}$ | $1.00_{\pm 0.00}$ | $1.00_{\pm 0.00}$ | $0.00_{\pm 0.00}$ | $0.00_{\pm 0.00}$ | $1.00_{\pm 0.00}$ | $1.00_{\pm 0.00}$ |
| SMOTE | $0.70_{\pm 0.00}$ | $0.99_{\pm 0.00}$ | $0.98_{\pm 0.00}$ | $0.77_{\pm 0.01}$ | $0.03_{\pm 0.01}$ | $0.00_{\pm 0.00}$ | $0.99_{\pm 0.01}$ | $0.40_{\pm 0.38}$ |
| BN | $0.77_{\pm 0.00}$ | $0.93_{\pm 0.00}$ | $0.95_{\pm 0.01}$ | $0.21_{\pm 0.01}$ | $0.15_{\pm 0.02}$ | $0.00_{\pm 0.00}$ | $0.72_{\pm 0.19}$ | $0.38_{\pm 0.36}$ |
| TVAE | $0.73_{\pm 0.01}$ | $0.88_{\pm 0.01}$ | $0.89_{\pm 0.01}$ | $0.13_{\pm 0.00}$ | $0.05_{\pm 0.01}$ | $0.06_{\pm 0.02}$ | $0.96_{\pm 0.00}$ | $0.61_{\pm 0.17}$ |
| GOGGLE | $0.73_{\pm 0.03}$ | $0.85_{\pm 0.05}$ | $0.84_{\pm 0.07}$ | $0.25_{\pm 0.07}$ | $0.11_{\pm 0.03}$ | $1.58_{\pm 1.11}$ | $0.72_{\pm 0.19}$ | $0.26_{\pm 0.23}$ |
| CTGAN | $0.68_{\pm 0.01}$ | $0.89_{\pm 0.01}$ | $0.92_{\pm 0.02}$ | $0.02_{\pm 0.00}$ | $0.06_{\pm 0.00}$ | $0.07_{\pm 0.06}$ | $0.96_{\pm 0.04}$ | $0.19_{\pm 0.16}$ |
| NFlow | $0.70_{\pm 0.01}$ | $0.81_{\pm 0.01}$ | $0.61_{\pm 0.06}$ | $0.00_{\pm 0.00}$ | $0.20_{\pm 0.16}$ | $5.97_{\pm 2.37}$ | $0.55_{\pm 0.03}$ | $0.05_{\pm 0.05}$ |
| ARF | $0.74_{\pm 0.01}$ | $0.76_{\pm 0.05}$ | $0.95_{\pm 0.03}$ | $0.07_{\pm 0.10}$ | $0.04_{\pm 0.01}$ | $0.15_{\pm 0.38}$ | $0.96_{\pm 0.01}$ | $0.53_{\pm 0.05}$ |
| TabDDPM | $0.64_{\pm 0.13}$ | $0.75_{\pm 0.16}$ | $0.45_{\pm 0.47}$ | $0.16_{\pm 0.17}$ | $0.18_{\pm 0.13}$ | $2.77_{\pm 2.45}$ | $0.72_{\pm 0.19}$ | $0.60_{\pm 0.34}$ |
| TabSyn | $0.58_{\pm 0.20}$ | $0.72_{\pm 0.18}$ | $0.75_{\pm 0.26}$ | $0.16_{\pm 0.17}$ | $0.12_{\pm 0.11}$ | $7.12_{\pm 12.67}$ | $0.95_{\pm 0.06}$ | $0.60_{\pm 0.34}$ |
| TabDiff | $0.74_{\pm 0.03}$ | $0.78_{\pm 0.12}$ | $0.80_{\pm 0.10}$ | $0.16_{\pm 0.17}$ | $0.10_{\pm 0.07}$ | $0.48_{\pm 0.54}$ | $0.95_{\pm 0.06}$ | $0.68_{\pm 0.22}$ |
| TabEBM | $0.74_{\pm 0.03}$ | $0.84_{\pm 0.05}$ | $0.94_{\pm 0.05}$ | $0.18_{\pm 0.15}$ | $0.26_{\pm 0.19}$ | $0.47_{\pm 0.54}$ | $0.95_{\pm 0.06}$ | $0.30_{\pm 0.27}$ |
| NRGBoost | $0.73_{\pm 0.03}$ | $0.86_{\pm 0.04}$ | $0.78_{\pm 0.12}$ | $0.16_{\pm 0.17}$ | $0.09_{\pm 0.01}$ | $1.92_{\pm 2.41}$ | $0.95_{\pm 0.06}$ | $0.17_{\pm 0.22}$ |
| GReaT | $0.73_{\pm 0.03}$ | $0.85_{\pm 0.05}$ | $0.84_{\pm 0.07}$ | $0.25_{\pm 0.07}$ | $0.11_{\pm 0.03}$ | $1.58_{\pm 1.11}$ | $0.72_{\pm 0.19}$ | $0.26_{\pm 0.23}$ |

Table 51: **Raw benchmark results of 13 tabular generators on "Phoneme" dataset.** We report the normalised mean ± std metric values across datasets. We highlight the First, Second and Third best performances for each metric. For visualisation, we abbreviate "conditional independence" as "CI". SMOTE generally achieves the highest performance in capturing local structure (i.e., local utility or local CI), while diffusion models typically excel at capturing global structure (i.e., global CI or global utility).

| Generator | Density Estimation | | | | Privacy Preservation | | ML Efficacy | Structural Fidelity |
|---|---|---|---|---|---|---|---|---|
| | Shape ↑ | Trend ↑ | $\alpha$-precision ↑ | $\beta$-recall ↑ | DCR ↑ | $\delta$-Presence ↑ | Local utility ↑ | Global utility ↑ |
| $\mathcal{D}_{\text{ref}}$ | $1.00_{\pm 0.00}$ | $1.00_{\pm 0.00}$ | $1.00_{\pm 0.00}$ | $1.00_{\pm 0.00}$ | $0.00_{\pm 0.00}$ | $0.00_{\pm 0.00}$ | $1.00_{\pm 0.00}$ | $1.00_{\pm 0.00}$ |
| SMOTE | $0.96_{\pm 0.00}$ | $0.95_{\pm 0.01}$ | $0.99_{\pm 0.00}$ | $0.74_{\pm 0.01}$ | $0.08_{\pm 0.01}$ | $0.00_{\pm 0.00}$ | $1.00_{\pm 0.04}$ | $0.44_{\pm 0.41}$ |
| BN | $0.97_{\pm 0.00}$ | $0.98_{\pm 0.00}$ | $0.98_{\pm 0.00}$ | $0.46_{\pm 0.01}$ | $0.12_{\pm 0.01}$ | $0.00_{\pm 0.00}$ | $0.82_{\pm 0.17}$ | $0.42_{\pm 0.38}$ |
| TVAE | $0.91_{\pm 0.00}$ | $0.86_{\pm 0.01}$ | $0.94_{\pm 0.01}$ | $0.13_{\pm 0.01}$ | $0.17_{\pm 0.01}$ | $0.01_{\pm 0.00}$ | $0.93_{\pm 0.07}$ | $0.52_{\pm 0.29}$ |
| GOGGLE | $0.88_{\pm 0.14}$ | $0.89_{\pm 0.08}$ | $0.93_{\pm 0.08}$ | $0.30_{\pm 0.12}$ | $0.14_{\pm 0.03}$ | $0.37_{\pm 0.94}$ | $0.79_{\pm 0.15}$ | $0.27_{\pm 0.24}$ |
| CTGAN | $0.80_{\pm 0.07}$ | $0.79_{\pm 0.04}$ | $0.89_{\pm 0.09}$ | $0.07_{\pm 0.01}$ | $0.19_{\pm 0.04}$ | $0.45_{\pm 0.74}$ | $0.90_{\pm 0.13}$ | $0.11_{\pm 0.12}$ |
| NFlow | $0.90_{\pm 0.02}$ | $0.90_{\pm 0.01}$ | $0.94_{\pm 0.04}$ | $0.09_{\pm 0.01}$ | $0.16_{\pm 0.02}$ | $0.02_{\pm 0.01}$ | $0.80_{\pm 0.04}$ | $0.22_{\pm 0.13}$ |
| ARF | $0.95_{\pm 0.00}$ | $0.91_{\pm 0.02}$ | $0.99_{\pm 0.00}$ | $0.22_{\pm 0.01}$ | $0.11_{\pm 0.02}$ | $0.00_{\pm 0.00}$ | $0.91_{\pm 0.01}$ | $0.67_{\pm 0.05}$ |
| TabDDPM | $0.94_{\pm 0.02}$ | $0.95_{\pm 0.03}$ | $0.97_{\pm 0.02}$ | $0.31_{\pm 0.08}$ | $0.10_{\pm 0.03}$ | $0.03_{\pm 0.05}$ | $0.79_{\pm 0.15}$ | $0.81_{\pm 0.20}$ |
| TabSyn | $0.90_{\pm 0.03}$ | $0.87_{\pm 0.04}$ | $0.95_{\pm 0.01}$ | $0.25_{\pm 0.14}$ | $0.16_{\pm 0.04}$ | $0.11_{\pm 0.13}$ | $0.88_{\pm 0.18}$ | $0.71_{\pm 0.30}$ |
| TabDiff | $0.92_{\pm 0.01}$ | $0.91_{\pm 0.01}$ | $0.96_{\pm 0.02}$ | $0.22_{\pm 0.18}$ | $0.18_{\pm 0.07}$ | $0.03_{\pm 0.05}$ | $0.93_{\pm 0.10}$ | $0.69_{\pm 0.33}$ |
| TabEBM | $0.94_{\pm 0.02}$ | $0.92_{\pm 0.02}$ | $0.97_{\pm 0.02}$ | $0.29_{\pm 0.11}$ | $0.24_{\pm 0.13}$ | $0.03_{\pm 0.05}$ | $0.97_{\pm 0.06}$ | $0.31_{\pm 0.27}$ |
| NRGBoost | $0.94_{\pm 0.01}$ | $0.93_{\pm 0.02}$ | $0.97_{\pm 0.01}$ | $0.23_{\pm 0.17}$ | $0.14_{\pm 0.03}$ | $0.03_{\pm 0.05}$ | $0.96_{\pm 0.07}$ | $0.21_{\pm 0.20}$ |
| GReaT | $0.90_{\pm 0.03}$ | $0.89_{\pm 0.03}$ | $0.86_{\pm 0.10}$ | $0.22_{\pm 0.18}$ | $0.19_{\pm 0.08}$ | $0.12_{\pm 0.21}$ | $0.71_{\pm 0.13}$ | $0.17_{\pm 0.21}$ |

Table 52: **Raw benchmark results of 13 tabular generators on "Plants" dataset.** We report the normalised mean ± std metric values across datasets. We highlight the First, Second and Third best performances for each metric. For visualisation, we abbreviate "conditional independence" as "CI". SMOTE generally achieves the highest performance in capturing local structure (i.e., local utility or local CI), while diffusion models typically excel at capturing global structure (i.e., global CI or global utility).

| Generator | Density Estimation | | | | Privacy Preservation | | ML Efficacy | Structural Fidelity |
|---|---|---|---|---|---|---|---|---|
| | Shape ↑ | Trend ↑ | $\alpha$-precision ↑ | $\beta$-recall ↑ | DCR ↑ | $\delta$-Presence ↑ | Local utility ↑ | Global utility ↑ |
| $\mathcal{D}_{\text{ref}}$ | $1.00_{\pm0.00}$ | $1.00_{\pm0.00}$ | $1.00_{\pm0.00}$ | $1.00_{\pm0.00}$ | $0.00_{\pm0.00}$ | $0.00_{\pm0.00}$ | $1.00_{\pm0.00}$ | $1.00_{\pm0.00}$ |
| SMOTE | $0.84_{\pm0.00}$ | $0.98_{\pm0.01}$ | $0.88_{\pm0.00}$ | $0.82_{\pm0.01}$ | $0.14_{\pm0.02}$ | $0.00_{\pm0.00}$ | $1.00_{\pm0.02}$ | $0.45_{\pm0.01}$ |
| BN | $0.87_{\pm0.00}$ | $0.97_{\pm0.00}$ | $0.97_{\pm0.01}$ | $0.26_{\pm0.00}$ | $0.15_{\pm0.02}$ | $0.00_{\pm0.00}$ | $0.89_{\pm0.08}$ | $0.29_{\pm0.11}$ |
| TVAE | $0.82_{\pm0.02}$ | $0.92_{\pm0.01}$ | $0.80_{\pm0.07}$ | $0.20_{\pm0.07}$ | $0.25_{\pm0.03}$ | $0.08_{\pm0.06}$ | $0.95_{\pm0.04}$ | $0.52_{\pm0.11}$ |
| GOGGLE | $0.84_{\pm0.06}$ | $0.94_{\pm0.03}$ | $0.88_{\pm0.05}$ | $0.37_{\pm0.06}$ | $0.20_{\pm0.03}$ | $0.18_{\pm0.16}$ | $0.88_{\pm0.08}$ | $0.28_{\pm0.01}$ |
| CTGAN | $0.79_{\pm0.05}$ | $0.91_{\pm0.02}$ | $0.89_{\pm0.06}$ | $0.24_{\pm0.08}$ | $0.20_{\pm0.04}$ | $0.20_{\pm0.22}$ | $0.95_{\pm0.06}$ | $0.19_{\pm0.07}$ |
| NFlow | $0.82_{\pm0.02}$ | $0.94_{\pm0.01}$ | $0.87_{\pm0.06}$ | $0.24_{\pm0.05}$ | $0.22_{\pm0.04}$ | $0.26_{\pm0.38}$ | $0.86_{\pm0.03}$ | $0.30_{\pm0.09}$ |
| ARF | $0.86_{\pm0.01}$ | $0.94_{\pm0.01}$ | $0.90_{\pm0.02}$ | $0.29_{\pm0.04}$ | $0.19_{\pm0.03}$ | $0.02_{\pm0.01}$ | $0.93_{\pm0.01}$ | $0.57_{\pm0.05}$ |
| TabDDPM | $0.82_{\pm0.08}$ | $0.95_{\pm0.02}$ | $0.78_{\pm0.16}$ | $0.33_{\pm0.09}$ | $0.19_{\pm0.08}$ | $0.10_{\pm0.08}$ | $0.88_{\pm0.08}$ | $0.74_{\pm0.11}$ |
| TabSyn | $0.85_{\pm0.03}$ | $0.94_{\pm0.02}$ | $0.91_{\pm0.03}$ | $0.31_{\pm0.06}$ | $0.18_{\pm0.05}$ | $0.08_{\pm0.05}$ | $0.94_{\pm0.08}$ | $0.74_{\pm0.03}$ |
| TabDiff | $0.85_{\pm0.02}$ | $0.93_{\pm0.02}$ | $0.86_{\pm0.05}$ | $0.25_{\pm0.03}$ | $0.27_{\pm0.11}$ | $0.09_{\pm0.06}$ | $0.96_{\pm0.05}$ | $0.68_{\pm0.03}$ |
| TabEBM | $0.87_{\pm0.02}$ | $0.96_{\pm0.01}$ | $0.94_{\pm0.03}$ | $0.35_{\pm0.14}$ | $0.30_{\pm0.05}$ | $0.05_{\pm0.05}$ | $0.97_{\pm0.04}$ | $0.28_{\pm0.06}$ |
| NRGBoost | $0.86_{\pm0.02}$ | $0.96_{\pm0.01}$ | $0.89_{\pm0.07}$ | $0.27_{\pm0.05}$ | $0.19_{\pm0.03}$ | $0.05_{\pm0.05}$ | $0.97_{\pm0.04}$ | $0.22_{\pm0.05}$ |
| GReaT | $0.85_{\pm0.02}$ | $0.94_{\pm0.01}$ | $0.86_{\pm0.05}$ | $0.34_{\pm0.11}$ | $0.21_{\pm0.05}$ | $0.10_{\pm0.04}$ | $0.85_{\pm0.07}$ | $0.24_{\pm0.06}$ |

Table 53: **Raw benchmark results of 13 tabular generators on "QSAR" dataset.** We report the normalised mean ± std metric values across datasets. We highlight the First, Second and Third best performances for each metric. For visualisation, we abbreviate "conditional independence" as "CI". SMOTE generally achieves the highest performance in capturing local structure (i.e., local utility or local CI), while diffusion models typically excel at capturing global structure (i.e., global CI or global utility).

| Generator | Density Estimation | | | | Privacy Preservation | | ML Efficacy | Structural Fidelity |
|---|---|---|---|---|---|---|---|---|
| | Shape ↑ | Trend ↑ | $\alpha$-precision ↑ | $\beta$-recall ↑ | DCR ↑ | $\delta$-Presence ↑ | Local utility ↑ | Global utility ↑ |
| $\mathcal{D}_{\text{ref}}$ | $1.00_{\pm0.00}$ | $1.00_{\pm0.00}$ | $1.00_{\pm0.00}$ | $1.00_{\pm0.00}$ | $0.00_{\pm0.00}$ | $0.00_{\pm0.00}$ | $1.00_{\pm0.00}$ | $1.00_{\pm0.00}$ |
| SMOTE | $0.76_{\pm0.00}$ | $0.96_{\pm0.01}$ | $0.95_{\pm0.01}$ | $0.76_{\pm0.01}$ | $0.08_{\pm0.01}$ | $0.01_{\pm0.01}$ | $0.93_{\pm0.11}$ | $0.41_{\pm0.03}$ |
| BN | $0.77_{\pm0.01}$ | $0.98_{\pm0.00}$ | $0.98_{\pm0.00}$ | $0.54_{\pm0.01}$ | $0.12_{\pm0.00}$ | $0.00_{\pm0.00}$ | $0.78_{\pm0.20}$ | $0.40_{\pm0.03}$ |
| TVAE | $0.71_{\pm0.01}$ | $0.89_{\pm0.00}$ | $0.83_{\pm0.03}$ | $0.09_{\pm0.01}$ | $0.15_{\pm0.02}$ | $0.10_{\pm0.04}$ | $0.85_{\pm0.17}$ | $0.47_{\pm0.08}$ |
| GOGGLE | $0.74_{\pm0.08}$ | $0.92_{\pm0.04}$ | $0.87_{\pm0.08}$ | $0.30_{\pm0.08}$ | $0.14_{\pm0.00}$ | $0.24_{\pm0.18}$ | $0.70_{\pm0.12}$ | $0.26_{\pm0.02}$ |
| CTGAN | $0.64_{\pm0.04}$ | $0.86_{\pm0.02}$ | $0.89_{\pm0.01}$ | $0.04_{\pm0.01}$ | $0.15_{\pm0.03}$ | $0.27_{\pm0.25}$ | $0.85_{\pm0.18}$ | $0.12_{\pm0.01}$ |
| NFlow | $0.72_{\pm0.01}$ | $0.91_{\pm0.01}$ | $0.85_{\pm0.07}$ | $0.05_{\pm0.01}$ | $0.16_{\pm0.00}$ | $0.10_{\pm0.07}$ | $0.66_{\pm0.03}$ | $0.19_{\pm0.03}$ |
| ARF | $0.77_{\pm0.00}$ | $0.93_{\pm0.01}$ | $0.96_{\pm0.01}$ | $0.15_{\pm0.01}$ | $0.13_{\pm0.03}$ | $0.01_{\pm0.01}$ | $0.75_{\pm0.01}$ | $0.55_{\pm0.07}$ |
| TabDDPM | $0.71_{\pm0.08}$ | $0.93_{\pm0.03}$ | $0.71_{\pm0.24}$ | $0.25_{\pm0.09}$ | $0.08_{\pm0.02}$ | $0.22_{\pm0.26}$ | $0.70_{\pm0.12}$ | $0.70_{\pm0.16}$ |
| TabSyn | $0.75_{\pm0.02}$ | $0.92_{\pm0.02}$ | $0.92_{\pm0.03}$ | $0.26_{\pm0.02}$ | $0.15_{\pm0.01}$ | $0.08_{\pm0.04}$ | $0.87_{\pm0.18}$ | $0.73_{\pm0.02}$ |
| TabDiff | $0.76_{\pm0.01}$ | $0.92_{\pm0.01}$ | $0.90_{\pm0.03}$ | $0.20_{\pm0.02}$ | $0.22_{\pm0.06}$ | $0.07_{\pm0.05}$ | $0.89_{\pm0.14}$ | $0.67_{\pm0.03}$ |
| TabEBM | $0.81_{\pm0.04}$ | $0.94_{\pm0.01}$ | $0.95_{\pm0.03}$ | $0.32_{\pm0.05}$ | $0.27_{\pm0.04}$ | $0.04_{\pm0.01}$ | $0.91_{\pm0.12}$ | $0.30_{\pm0.01}$ |
| NRGBoost | $0.76_{\pm0.03}$ | $0.93_{\pm0.00}$ | $0.77_{\pm0.16}$ | $0.21_{\pm0.03}$ | $0.17_{\pm0.03}$ | $0.04_{\pm0.02}$ | $0.90_{\pm0.12}$ | $0.19_{\pm0.03}$ |
| GReaT | $0.71_{\pm0.06}$ | $0.91_{\pm0.03}$ | $0.76_{\pm0.15}$ | $0.20_{\pm0.02}$ | $0.17_{\pm0.02}$ | $0.17_{\pm0.07}$ | $0.66_{\pm0.11}$ | $0.18_{\pm0.02}$ |

Table 54: **Raw benchmark results of 13 tabular generators on "SpeedDating" dataset.** We report the normalised mean $\pm$ std metric values across datasets. We highlight the First, Second and Third best performances for each metric. For visualisation, we abbreviate "conditional independence" as "CI". SMOTE generally achieves the highest performance in capturing local structure (i.e., local utility or local CI), while diffusion models typically excel at capturing global structure (i.e., global CI or global utility).

| Generator | Density Estimation | | | | Privacy Preservation | | ML Efficacy | Structural Fidelity |
|---|---|---|---|---|---|---|---|---|
| | Shape ↑ | Trend ↑ | $\alpha$-precision ↑ | $\beta$-recall ↑ | DCR ↑ | $\delta$-Presence ↑ | Local utility ↑ | Global utility ↑ |
| $\mathcal{D}_{\text{ref}}$ | $1.00_{\pm 0.00}$ | $1.00_{\pm 0.00}$ | $1.00_{\pm 0.00}$ | $1.00_{\pm 0.00}$ | $0.00_{\pm 0.00}$ | $0.00_{\pm 0.00}$ | $1.00_{\pm 0.00}$ | $1.00_{\pm 0.00}$ |
| SMOTE | $0.95_{\pm 0.00}$ | $0.94_{\pm 0.01}$ | $0.96_{\pm 0.01}$ | $0.78_{\pm 0.01}$ | $0.14_{\pm 0.03}$ | $0.00_{\pm 0.00}$ | $1.02_{\pm 0.03}$ | $0.44_{\pm 0.01}$ |
| BN | $0.95_{\pm 0.00}$ | $0.95_{\pm 0.00}$ | $0.97_{\pm 0.00}$ | $0.28_{\pm 0.01}$ | $0.34_{\pm 0.02}$ | $0.01_{\pm 0.00}$ | $0.83_{\pm 0.10}$ | $0.36_{\pm 0.09}$ |
| TVAE | $0.90_{\pm 0.00}$ | $0.86_{\pm 0.00}$ | $0.91_{\pm 0.02}$ | $0.07_{\pm 0.00}$ | $0.34_{\pm 0.03}$ | $0.01_{\pm 0.00}$ | $0.98_{\pm 0.05}$ | $0.52_{\pm 0.00}$ |
| GOGGLE | $0.88_{\pm 0.00}$ | $0.87_{\pm 0.02}$ | $0.87_{\pm 0.07}$ | $0.28_{\pm 0.03}$ | $0.25_{\pm 0.03}$ | $0.24_{\pm 0.18}$ | $0.83_{\pm 0.11}$ | $0.25_{\pm 0.03}$ |
| CTGAN | $0.85_{\pm 0.04}$ | $0.84_{\pm 0.02}$ | $0.92_{\pm 0.05}$ | $0.04_{\pm 0.01}$ | $0.26_{\pm 0.03}$ | $0.23_{\pm 0.31}$ | $0.93_{\pm 0.11}$ | $0.13_{\pm 0.02}$ |
| NFlow | $0.86_{\pm 0.01}$ | $0.84_{\pm 0.01}$ | $0.75_{\pm 0.04}$ | $0.05_{\pm 0.00}$ | $0.25_{\pm 0.03}$ | $0.15_{\pm 0.19}$ | $0.81_{\pm 0.03}$ | $0.13_{\pm 0.08}$ |
| ARF | $0.92_{\pm 0.02}$ | $0.89_{\pm 0.03}$ | $0.90_{\pm 0.03}$ | $0.24_{\pm 0.03}$ | $0.23_{\pm 0.03}$ | $0.06_{\pm 0.06}$ | $0.92_{\pm 0.01}$ | $0.56_{\pm 0.06}$ |
| TabDDPM | $0.86_{\pm 0.08}$ | $0.81_{\pm 0.13}$ | $0.71_{\pm 0.24}$ | $0.24_{\pm 0.11}$ | $0.23_{\pm 0.04}$ | $0.31_{\pm 0.40}$ | $0.84_{\pm 0.10}$ | $0.72_{\pm 0.14}$ |
| TabSyn | $0.86_{\pm 0.05}$ | $0.83_{\pm 0.07}$ | $0.86_{\pm 0.08}$ | $0.21_{\pm 0.06}$ | $0.27_{\pm 0.06}$ | $0.09_{\pm 0.03}$ | $0.93_{\pm 0.12}$ | $0.69_{\pm 0.04}$ |
| TabDiff | $0.89_{\pm 0.04}$ | $0.86_{\pm 0.05}$ | $0.88_{\pm 0.06}$ | $0.19_{\pm 0.04}$ | $0.30_{\pm 0.09}$ | $0.14_{\pm 0.15}$ | $0.96_{\pm 0.08}$ | $0.67_{\pm 0.03}$ |
| TabEBM | $0.93_{\pm 0.01}$ | $0.91_{\pm 0.01}$ | $0.95_{\pm 0.03}$ | $0.27_{\pm 0.02}$ | $0.33_{\pm 0.13}$ | $0.02_{\pm 0.01}$ | $0.98_{\pm 0.06}$ | $0.30_{\pm 0.01}$ |
| NRGBoost | $0.91_{\pm 0.03}$ | $0.90_{\pm 0.02}$ | $0.87_{\pm 0.06}$ | $0.19_{\pm 0.05}$ | $0.22_{\pm 0.03}$ | $0.04_{\pm 0.02}$ | $0.97_{\pm 0.07}$ | $0.18_{\pm 0.05}$ |
| GReaT | $0.89_{\pm 0.01}$ | $0.87_{\pm 0.02}$ | $0.84_{\pm 0.04}$ | $0.24_{\pm 0.03}$ | $0.27_{\pm 0.06}$ | $0.12_{\pm 0.00}$ | $0.80_{\pm 0.10}$ | $0.20_{\pm 0.04}$ |

Table 55: **Raw benchmark results of 13 tabular generators on "Splice" dataset.** We report the normalised mean $\pm$ std metric values across datasets. We highlight the First, Second and Third best performances for each metric. For visualisation, we abbreviate "conditional independence" as "CI". SMOTE generally achieves the highest performance in capturing local structure (i.e., local utility or local CI), while diffusion models typically excel at capturing global structure (i.e., global CI or global utility).

| Generator | Density Estimation | | | | Privacy Preservation | | ML Efficacy | Structural Fidelity |
|---|---|---|---|---|---|---|---|---|
| | Shape ↑ | Trend ↑ | $\alpha$-precision ↑ | $\beta$-recall ↑ | DCR ↑ | $\delta$-Presence ↑ | Local utility ↑ | Global utility ↑ |
| $\mathcal{D}_{\text{ref}}$ | $1.00_{\pm 0.00}$ | $1.00_{\pm 0.00}$ | $1.00_{\pm 0.00}$ | $1.00_{\pm 0.00}$ | $0.00_{\pm 0.00}$ | $0.00_{\pm 0.00}$ | $1.00_{\pm 0.00}$ | $1.00_{\pm 0.00}$ |
| SMOTE | $0.98_{\pm 0.00}$ | $0.97_{\pm 0.00}$ | $0.98_{\pm 0.00}$ | $0.88_{\pm 0.01}$ | $0.00_{\pm 0.00}$ | $0.00_{\pm 0.00}$ | $1.01_{\pm 0.04}$ | $0.47_{\pm 0.34}$ |
| BN | $0.99_{\pm 0.00}$ | $0.95_{\pm 0.00}$ | $0.93_{\pm 0.01}$ | $0.34_{\pm 0.01}$ | $0.64_{\pm 0.05}$ | $0.04_{\pm 0.04}$ | $0.69_{\pm 0.27}$ | $0.19_{\pm 0.06}$ |
| TVAE | $0.95_{\pm 0.00}$ | $0.92_{\pm 0.00}$ | $0.89_{\pm 0.01}$ | $0.56_{\pm 0.01}$ | $0.83_{\pm 0.04}$ | $0.03_{\pm 0.02}$ | $1.00_{\pm 0.04}$ | $0.48_{\pm 0.23}$ |
| GOGGLE | $0.94_{\pm 0.02}$ | $0.90_{\pm 0.03}$ | $0.83_{\pm 0.08}$ | $0.49_{\pm 0.07}$ | $0.50_{\pm 0.03}$ | $0.08_{\pm 0.08}$ | $0.64_{\pm 0.22}$ | $0.30_{\pm 0.16}$ |
| CTGAN | $0.94_{\pm 0.01}$ | $0.90_{\pm 0.01}$ | $0.94_{\pm 0.04}$ | $0.43_{\pm 0.01}$ | $0.67_{\pm 0.03}$ | $0.03_{\pm 0.05}$ | $0.99_{\pm 0.05}$ | $0.24_{\pm 0.10}$ |
| NFlow | $0.85_{\pm 0.01}$ | $0.77_{\pm 0.01}$ | $0.70_{\pm 0.15}$ | $0.21_{\pm 0.04}$ | $0.64_{\pm 0.07}$ | $0.04_{\pm 0.03}$ | $0.47_{\pm 0.07}$ | $0.31_{\pm 0.20}$ |
| ARF | $0.99_{\pm 0.00}$ | $0.95_{\pm 0.00}$ | $0.91_{\pm 0.01}$ | $0.45_{\pm 0.01}$ | $0.91_{\pm 0.02}$ | $0.01_{\pm 0.00}$ | $0.78_{\pm 0.07}$ | $0.21_{\pm 0.05}$ |
| TabDDPM | $0.95_{\pm 0.01}$ | $0.91_{\pm 0.02}$ | $0.70_{\pm 0.21}$ | $0.38_{\pm 0.18}$ | $0.61_{\pm 0.09}$ | $0.06_{\pm 0.06}$ | $0.54_{\pm 0.20}$ | $0.60_{\pm 0.28}$ |
| TabSyn | $0.85_{\pm 0.11}$ | $0.77_{\pm 0.17}$ | $0.54_{\pm 0.39}$ | $0.32_{\pm 0.25}$ | $0.57_{\pm 0.06}$ | $0.35_{\pm 0.63}$ | $0.93_{\pm 0.11}$ | $0.66_{\pm 0.24}$ |
| TabDiff | $0.87_{\pm 0.09}$ | $0.80_{\pm 0.13}$ | $0.56_{\pm 0.38}$ | $0.31_{\pm 0.25}$ | $0.58_{\pm 0.07}$ | $0.39_{\pm 0.47}$ | $0.88_{\pm 0.18}$ | $0.65_{\pm 0.25}$ |
| TabEBM | $0.96_{\pm 0.01}$ | $0.93_{\pm 0.01}$ | $0.94_{\pm 0.05}$ | $0.59_{\pm 0.04}$ | $0.26_{\pm 0.28}$ | $0.01_{\pm 0.01}$ | $0.98_{\pm 0.07}$ | $0.35_{\pm 0.23}$ |
| NRGBoost | $0.92_{\pm 0.04}$ | $0.88_{\pm 0.05}$ | $0.82_{\pm 0.10}$ | $0.38_{\pm 0.18}$ | $0.28_{\pm 0.26}$ | $0.04_{\pm 0.03}$ | $0.97_{\pm 0.07}$ | $0.31_{\pm 0.20}$ |
| GReaT | $0.94_{\pm 0.02}$ | $0.90_{\pm 0.03}$ | $0.83_{\pm 0.08}$ | $0.49_{\pm 0.07}$ | $0.50_{\pm 0.03}$ | $0.08_{\pm 0.08}$ | $0.64_{\pm 0.22}$ | $0.30_{\pm 0.16}$ |

Table 56: **Raw benchmark results of 13 tabular generators on "Vehicle" dataset.** We report the normalised mean $\pm$ std metric values across datasets. We highlight the **First**, **Second** and **Third** best performances for each metric. For visualisation, we abbreviate "conditional independence" as "CI". SMOTE generally achieves the highest performance in capturing local structure (i.e., local utility or local CI), while diffusion models typically excel at capturing global structure (i.e., global CI or global utility).

| Generator | Density Estimation | | | | Privacy Preservation | | ML Efficacy | Structural Fidelity |
|---|---|---|---|---|---|---|---|---|
| | Shape $\uparrow$ | Trend $\uparrow$ | $\alpha$-precision $\uparrow$ | $\beta$-recall $\uparrow$ | DCR $\uparrow$ | $\delta$-Presence $\uparrow$ | Local utility $\uparrow$ | Global utility $\uparrow$ |
| $\mathcal{D}_{ref}$ | $1.00_{\pm0.00}$ | $1.00_{\pm0.00}$ | $1.00_{\pm0.00}$ | $1.00_{\pm0.00}$ | $0.00_{\pm0.00}$ | $0.00_{\pm0.00}$ | $1.00_{\pm0.00}$ | $1.00_{\pm0.00}$ |
| SMOTE | $\mathbf{0.95}_{\pm0.00}$ | $\mathbf{0.97}_{\pm0.01}$ | $\mathbf{0.96}_{\pm0.01}$ | $\mathbf{0.86}_{\pm0.01}$ | $0.12_{\pm0.04}$ | $0.01_{\pm0.01}$ | $\mathbf{0.98}_{\pm0.04}$ | $0.41_{\pm0.37}$ |
| BN | $\mathbf{0.94}_{\pm0.00}$ | $\mathbf{0.96}_{\pm0.00}$ | $\mathbf{0.97}_{\pm0.01}$ | $\mathbf{0.33}_{\pm0.03}$ | $0.17_{\pm0.02}$ | $0.01_{\pm0.01}$ | $0.64_{\pm0.24}$ | $0.28_{\pm0.24}$ |
| TVAE | $0.83_{\pm0.01}$ | $0.86_{\pm0.00}$ | $0.77_{\pm0.01}$ | $0.08_{\pm0.01}$ | $\mathbf{0.39}_{\pm0.08}$ | $\mathbf{0.16}_{\pm0.09}$ | $0.84_{\pm0.10}$ | $0.39_{\pm0.35}$ |
| GOGGLE | $0.89_{\pm0.01}$ | $0.91_{\pm0.01}$ | $0.88_{\pm0.03}$ | $0.30_{\pm0.05}$ | $0.22_{\pm0.04}$ | $0.07_{\pm0.02}$ | $0.59_{\pm0.19}$ | $0.23_{\pm0.18}$ |
| CTGAN | $0.78_{\pm0.02}$ | $0.90_{\pm0.01}$ | $0.82_{\pm0.05}$ | $0.02_{\pm0.01}$ | $0.24_{\pm0.06}$ | $0.13_{\pm0.13}$ | $0.82_{\pm0.19}$ | $0.08_{\pm0.05}$ |
| NFlow | $0.88_{\pm0.01}$ | $0.85_{\pm0.01}$ | $0.89_{\pm0.02}$ | $0.00_{\pm0.00}$ | $0.24_{\pm0.03}$ | $\mathbf{0.13}_{\pm0.07}$ | $0.46_{\pm0.06}$ | $0.09_{\pm0.05}$ |
| ARF | $\mathbf{0.94}_{\pm0.00}$ | $0.93_{\pm0.00}$ | $\mathbf{0.96}_{\pm0.01}$ | $0.16_{\pm0.02}$ | $0.17_{\pm0.03}$ | $0.01_{\pm0.00}$ | $0.84_{\pm0.04}$ | $0.43_{\pm0.05}$ |
| TabDDPM | $0.85_{\pm0.06}$ | $0.90_{\pm0.02}$ | $0.77_{\pm0.15}$ | $0.28_{\pm0.07}$ | $0.14_{\pm0.05}$ | $0.04_{\pm0.03}$ | $0.62_{\pm0.23}$ | $\mathbf{0.72}_{\pm0.28}$ |
| TabSyn | $0.88_{\pm0.02}$ | $0.93_{\pm0.01}$ | $0.92_{\pm0.03}$ | $0.27_{\pm0.09}$ | $0.21_{\pm0.04}$ | $0.04_{\pm0.03}$ | $\mathbf{0.88}_{\pm0.12}$ | $\mathbf{0.74}_{\pm0.26}$ |
| TabDiff | $0.88_{\pm0.03}$ | $0.87_{\pm0.06}$ | $0.84_{\pm0.08}$ | $0.18_{\pm0.19}$ | $\mathbf{0.31}_{\pm0.13}$ | $0.07_{\pm0.05}$ | $0.84_{\pm0.19}$ | $\mathbf{0.62}_{\pm0.40}$ |
| TabEBM | $0.91_{\pm0.01}$ | $\mathbf{0.94}_{\pm0.02}$ | $0.93_{\pm0.03}$ | $\mathbf{0.40}_{\pm0.05}$ | $\mathbf{0.36}_{\pm0.18}$ | $0.04_{\pm0.03}$ | $\mathbf{0.93}_{\pm0.09}$ | $0.28_{\pm0.25}$ |
| NRGBoost | $0.91_{\pm0.00}$ | $0.88_{\pm0.05}$ | $0.88_{\pm0.03}$ | $0.18_{\pm0.18}$ | $0.26_{\pm0.08}$ | $0.12_{\pm0.10}$ | $0.88_{\pm0.13}$ | $0.15_{\pm0.16}$ |
| GReaT | $0.85_{\pm0.06}$ | $0.87_{\pm0.06}$ | $0.75_{\pm0.17}$ | $0.18_{\pm0.18}$ | $0.27_{\pm0.09}$ | $\mathbf{0.20}_{\pm0.17}$ | $0.49_{\pm0.19}$ | $0.15_{\pm0.16}$ |

Table 57: **Raw benchmark results of 13 tabular generators on "Zernike" dataset.** We report the normalised mean $\pm$ std metric values across datasets. We highlight the **First**, **Second** and **Third** best performances for each metric. For visualisation, we abbreviate "conditional independence" as "CI". SMOTE generally achieves the highest performance in capturing local structure (i.e., local utility or local CI), while diffusion models typically excel at capturing global structure (i.e., global CI or global utility).

| Generator | Density Estimation | | | | Privacy Preservation | | ML Efficacy | Structural Fidelity |
|---|---|---|---|---|---|---|---|---|
| | Shape $\uparrow$ | Trend $\uparrow$ | $\alpha$-precision $\uparrow$ | $\beta$-recall $\uparrow$ | DCR $\uparrow$ | $\delta$-Presence $\uparrow$ | Local utility $\uparrow$ | Global utility $\uparrow$ |
| $\mathcal{D}_{ref}$ | $1.00_{\pm0.00}$ | $1.00_{\pm0.00}$ | $1.00_{\pm0.00}$ | $1.00_{\pm0.00}$ | $0.00_{\pm0.00}$ | $0.00_{\pm0.00}$ | $1.00_{\pm0.00}$ | $1.00_{\pm0.00}$ |
| SMOTE | $\mathbf{0.97}_{\pm0.00}$ | $\mathbf{0.98}_{\pm0.00}$ | $\mathbf{0.90}_{\pm0.01}$ | $\mathbf{0.90}_{\pm0.01}$ | $0.20_{\pm0.03}$ | $0.00_{\pm0.00}$ | $\mathbf{0.98}_{\pm0.03}$ | $0.31_{\pm0.32}$ |
| BN | $\mathbf{0.97}_{\pm0.00}$ | $\mathbf{0.98}_{\pm0.00}$ | $\mathbf{0.96}_{\pm0.01}$ | $\mathbf{0.72}_{\pm0.01}$ | $0.18_{\pm0.02}$ | $0.00_{\pm0.00}$ | $0.54_{\pm0.42}$ | $0.31_{\pm0.31}$ |
| TVAE | $0.87_{\pm0.00}$ | $0.93_{\pm0.00}$ | $0.76_{\pm0.02}$ | $0.03_{\pm0.01}$ | $\mathbf{0.47}_{\pm0.03}$ | $\mathbf{0.31}_{\pm0.20}$ | $\mathbf{0.90}_{\pm0.06}$ | $0.38_{\pm0.37}$ |
| GOGGLE | $0.90_{\pm0.02}$ | $0.94_{\pm0.01}$ | $0.79_{\pm0.06}$ | $0.31_{\pm0.07}$ | $0.35_{\pm0.03}$ | $0.18_{\pm0.06}$ | $0.42_{\pm0.30}$ | $0.18_{\pm0.18}$ |
| CTGAN | $0.81_{\pm0.02}$ | $0.95_{\pm0.00}$ | $0.65_{\pm0.07}$ | $0.00_{\pm0.00}$ | $0.40_{\pm0.05}$ | $0.03_{\pm0.05}$ | $0.82_{\pm0.19}$ | $0.06_{\pm0.06}$ |
| NFlow | $0.90_{\pm0.01}$ | $0.87_{\pm0.00}$ | $0.77_{\pm0.02}$ | $0.00_{\pm0.00}$ | $\mathbf{0.41}_{\pm0.03}$ | $\mathbf{0.80}_{\pm0.20}$ | $0.14_{\pm0.03}$ | $0.01_{\pm0.01}$ |
| ARF | $\mathbf{0.96}_{\pm0.00}$ | $0.94_{\pm0.00}$ | $0.87_{\pm0.01}$ | $0.01_{\pm0.00}$ | $0.40_{\pm0.04}$ | $0.01_{\pm0.00}$ | $0.77_{\pm0.04}$ | $0.21_{\pm0.01}$ |
| TabDDPM | $0.68_{\pm0.26}$ | $0.92_{\pm0.03}$ | $0.44_{\pm0.43}$ | $0.19_{\pm0.20}$ | $\mathbf{0.51}_{\pm0.20}$ | $0.21_{\pm0.11}$ | $0.30_{\pm0.28}$ | $\mathbf{0.62}_{\pm0.41}$ |
| TabSyn | $0.92_{\pm0.01}$ | $0.96_{\pm0.01}$ | $0.83_{\pm0.03}$ | $0.24_{\pm0.14}$ | $0.36_{\pm0.05}$ | $0.09_{\pm0.09}$ | $0.84_{\pm0.17}$ | $\mathbf{0.70}_{\pm0.30}$ |
| TabDiff | $0.91_{\pm0.01}$ | $0.92_{\pm0.03}$ | $0.77_{\pm0.08}$ | $0.19_{\pm0.20}$ | $0.40_{\pm0.09}$ | $0.11_{\pm0.07}$ | $0.82_{\pm0.21}$ | $\mathbf{0.61}_{\pm0.42}$ |
| TabEBM | $0.94_{\pm0.02}$ | $\mathbf{0.96}_{\pm0.02}$ | $\mathbf{0.91}_{\pm0.07}$ | $\mathbf{0.40}_{\pm0.03}$ | $0.37_{\pm0.06}$ | $0.08_{\pm0.09}$ | $\mathbf{0.92}_{\pm0.10}$ | $0.23_{\pm0.23}$ |
| NRGBoost | $0.95_{\pm0.02}$ | $0.94_{\pm0.01}$ | $0.89_{\pm0.05}$ | $0.19_{\pm0.20}$ | $0.35_{\pm0.05}$ | $0.16_{\pm0.14}$ | $0.90_{\pm0.11}$ | $0.13_{\pm0.16}$ |
| GReaT | $0.84_{\pm0.09}$ | $0.91_{\pm0.04}$ | $0.55_{\pm0.31}$ | $0.19_{\pm0.20}$ | $0.29_{\pm0.05}$ | $\mathbf{0.56}_{\pm0.53}$ | $0.29_{\pm0.28}$ | $0.11_{\pm0.17}$ |

### F.2.2 REGRESSION DATASETS

Table 58: **Raw benchmark results of 13 tabular generators on "Ailerons" dataset.** We report the normalised mean $\pm$ std metric values across datasets. We highlight the **First**, **Second** and **Third** best performances for each metric. For visualisation, we abbreviate "conditional independence" as "CI". SMOTE generally achieves the highest performance in capturing local structure (i.e., local utility or local CI), while diffusion models typically excel at capturing global structure (i.e., global CI or global utility).

| Generator | Density Estimation | | | | Privacy Preservation | | ML Efficacy | Structural Fidelity |
|---|---|---|---|---|---|---|---|---|
| | Shape ↑ | Trend ↑ | $\alpha$-precision ↑ | $\beta$-recall ↑ | DCR ↑ | $\delta$-Presence ↑ | Local utility ↑ | Global utility ↑ |
| $\mathcal{D}_{\text{ref}}$ | $1.00_{\pm 0.00}$ | $1.00_{\pm 0.00}$ | $1.00_{\pm 0.00}$ | $1.00_{\pm 0.00}$ | $0.00_{\pm 0.00}$ | $0.00_{\pm 0.00}$ | $1.00_{\pm 0.00}$ | $1.00_{\pm 0.00}$ |
| SMOTE | $0.71_{\pm 0.03}$ | $0.99_{\pm 0.00}$ | $0.90_{\pm 0.01}$ | $0.85_{\pm 0.00}$ | $0.01_{\pm 0.00}$ | $0.00_{\pm 0.00}$ | $0.88_{\pm 0.20}$ | $0.35_{\pm 0.36}$ |
| BN | $0.75_{\pm 0.03}$ | $0.96_{\pm 0.00}$ | $0.94_{\pm 0.00}$ | $0.13_{\pm 0.00}$ | $0.05_{\pm 0.02}$ | $0.00_{\pm 0.00}$ | $0.56_{\pm 0.31}$ | $0.51_{\pm 0.18}$ |
| TVAE | $0.70_{\pm 0.03}$ | $0.96_{\pm 0.00}$ | $0.86_{\pm 0.02}$ | $0.28_{\pm 0.01}$ | $0.02_{\pm 0.00}$ | $0.18_{\pm 0.26}$ | $0.76_{\pm 0.23}$ | $0.46_{\pm 0.20}$ |
| GOGGLE | $0.57_{\pm 0.20}$ | $0.90_{\pm 0.06}$ | $0.53_{\pm 0.37}$ | $0.19_{\pm 0.19}$ | $0.05_{\pm 0.04}$ | $1.61_{\pm 2.36}$ | $0.80_{\pm 0.29}$ | $0.18_{\pm 0.20}$ |
| CTGAN | $0.68_{\pm 0.02}$ | $0.96_{\pm 0.00}$ | $0.91_{\pm 0.05}$ | $0.09_{\pm 0.02}$ | $0.02_{\pm 0.00}$ | $0.06_{\pm 0.06}$ | $0.75_{\pm 0.38}$ | $0.13_{\pm 0.12}$ |
| NFlow | $0.68_{\pm 0.03}$ | $0.89_{\pm 0.01}$ | $0.63_{\pm 0.06}$ | $0.00_{\pm 0.00}$ | $0.12_{\pm 0.10}$ | $0.94_{\pm 0.75}$ | $0.51_{\pm 0.33}$ | $0.05_{\pm 0.07}$ |
| ARF | $0.73_{\pm 0.02}$ | $0.98_{\pm 0.00}$ | $0.95_{\pm 0.01}$ | $0.22_{\pm 0.01}$ | $0.03_{\pm 0.00}$ | $0.00_{\pm 0.00}$ | $0.64_{\pm 0.26}$ | $0.58_{\pm 0.15}$ |
| TabDDPM | $0.72_{\pm 0.04}$ | $0.94_{\pm 0.02}$ | $0.84_{\pm 0.05}$ | $0.30_{\pm 0.07}$ | $0.02_{\pm 0.02}$ | $0.09_{\pm 0.10}$ | $0.52_{\pm 0.33}$ | $0.69_{\pm 0.24}$ |
| TabSyn | $0.52_{\pm 0.26}$ | $0.90_{\pm 0.07}$ | $0.60_{\pm 0.33}$ | $0.18_{\pm 0.19}$ | $0.17_{\pm 0.21}$ | $3.08_{\pm 4.43}$ | $0.81_{\pm 0.27}$ | $0.64_{\pm 0.32}$ |
| TabDiff | $0.76_{\pm 0.02}$ | $0.97_{\pm 0.02}$ | $0.87_{\pm 0.16}$ | $0.22_{\pm 0.16}$ | $0.07_{\pm 0.12}$ | $0.09_{\pm 0.10}$ | $0.87_{\pm 0.20}$ | $0.71_{\pm 0.23}$ |
| TabEBM | $0.76_{\pm 0.02}$ | $0.96_{\pm 0.00}$ | $0.93_{\pm 0.05}$ | $0.22_{\pm 0.15}$ | $0.07_{\pm 0.04}$ | $0.09_{\pm 0.10}$ | $0.57_{\pm 0.29}$ | $0.43_{\pm 0.03}$ |
| NRGBoost | $0.68_{\pm 0.08}$ | $0.92_{\pm 0.04}$ | $0.53_{\pm 0.37}$ | $0.18_{\pm 0.19}$ | $0.22_{\pm 0.22}$ | $0.39_{\pm 0.48}$ | $0.79_{\pm 0.29}$ | $0.18_{\pm 0.20}$ |
| GReaT | $0.67_{\pm 0.10}$ | $0.94_{\pm 0.03}$ | $0.67_{\pm 0.22}$ | $0.20_{\pm 0.18}$ | $0.07_{\pm 0.05}$ | $0.97_{\pm 1.33}$ | $0.49_{\pm 0.33}$ | $0.20_{\pm 0.20}$ |

Table 59: **Raw benchmark results of 13 tabular generators on "California" dataset.** We report the normalised mean $\pm$ std metric values across datasets. We highlight the **First**, **Second** and **Third** best performances for each metric. For visualisation, we abbreviate "conditional independence" as "CI". SMOTE generally achieves the highest performance in capturing local structure (i.e., local utility or local CI), while diffusion models typically excel at capturing global structure (i.e., global CI or global utility).

| Generator | Density Estimation | | | | Privacy Preservation | | ML Efficacy | Structural Fidelity |
|---|---|---|---|---|---|---|---|---|
| | Shape ↑ | Trend ↑ | $\alpha$-precision ↑ | $\beta$-recall ↑ | DCR ↑ | $\delta$-Presence ↑ | Local utility ↑ | Global utility ↑ |
| $\mathcal{D}_{\text{ref}}$ | $1.00_{\pm 0.00}$ | $1.00_{\pm 0.00}$ | $1.00_{\pm 0.00}$ | $1.00_{\pm 0.00}$ | $0.00_{\pm 0.00}$ | $0.00_{\pm 0.00}$ | $1.00_{\pm 0.00}$ | $1.00_{\pm 0.00}$ |
| SMOTE | $0.98_{\pm 0.00}$ | $0.99_{\pm 0.00}$ | $0.99_{\pm 0.00}$ | $0.78_{\pm 0.00}$ | $0.03_{\pm 0.02}$ | $0.00_{\pm 0.00}$ | $0.96_{\pm 0.05}$ | $0.44_{\pm 0.44}$ |
| BN | $0.98_{\pm 0.00}$ | $0.97_{\pm 0.01}$ | $0.98_{\pm 0.00}$ | $0.44_{\pm 0.00}$ | $0.04_{\pm 0.02}$ | $0.00_{\pm 0.00}$ | $0.73_{\pm 0.27}$ | $0.72_{\pm 0.14}$ |
| TVAE | $0.94_{\pm 0.01}$ | $0.91_{\pm 0.01}$ | $0.97_{\pm 0.01}$ | $0.23_{\pm 0.01}$ | $0.07_{\pm 0.02}$ | $0.10_{\pm 0.09}$ | $0.81_{\pm 0.12}$ | $0.53_{\pm 0.24}$ |
| GOGGLE | $0.71_{\pm 0.26}$ | $0.83_{\pm 0.10}$ | $0.72_{\pm 0.26}$ | $0.21_{\pm 0.22}$ | $0.08_{\pm 0.03}$ | $2.75_{\pm 3.91}$ | $0.80_{\pm 0.25}$ | $0.14_{\pm 0.22}$ |
| CTGAN | $0.91_{\pm 0.01}$ | $0.93_{\pm 0.00}$ | $0.96_{\pm 0.02}$ | $0.18_{\pm 0.02}$ | $0.03_{\pm 0.01}$ | $0.15_{\pm 0.12}$ | $0.84_{\pm 0.17}$ | $0.16_{\pm 0.16}$ |
| NFlow | $0.89_{\pm 0.02}$ | $0.86_{\pm 0.01}$ | $0.90_{\pm 0.04}$ | $0.08_{\pm 0.03}$ | $0.12_{\pm 0.05}$ | $0.33_{\pm 0.38}$ | $0.45_{\pm 0.10}$ | $0.06_{\pm 0.10}$ |
| ARF | $0.97_{\pm 0.00}$ | $0.87_{\pm 0.01}$ | $0.99_{\pm 0.00}$ | $0.26_{\pm 0.01}$ | $0.05_{\pm 0.01}$ | $0.00_{\pm 0.00}$ | $0.69_{\pm 0.24}$ | $0.68_{\pm 0.16}$ |
| TabDDPM | $0.93_{\pm 0.03}$ | $0.94_{\pm 0.01}$ | $0.94_{\pm 0.04}$ | $0.42_{\pm 0.00}$ | $0.04_{\pm 0.02}$ | $0.04_{\pm 0.04}$ | $0.60_{\pm 0.23}$ | $0.79_{\pm 0.19}$ |
| TabSyn | $0.95_{\pm 0.01}$ | $0.94_{\pm 0.01}$ | $0.92_{\pm 0.07}$ | $0.40_{\pm 0.03}$ | $0.06_{\pm 0.02}$ | $0.39_{\pm 0.54}$ | $0.88_{\pm 0.13}$ | $0.78_{\pm 0.20}$ |
| TabDiff | $0.94_{\pm 0.02}$ | $0.90_{\pm 0.04}$ | $0.96_{\pm 0.02}$ | $0.28_{\pm 0.16}$ | $0.09_{\pm 0.04}$ | $0.04_{\pm 0.04}$ | $0.88_{\pm 0.13}$ | $0.75_{\pm 0.22}$ |
| TabEBM | $0.93_{\pm 0.02}$ | $0.92_{\pm 0.01}$ | $0.97_{\pm 0.01}$ | $0.24_{\pm 0.19}$ | $0.11_{\pm 0.05}$ | $0.04_{\pm 0.04}$ | $0.62_{\pm 0.18}$ | $0.47_{\pm 0.05}$ |
| NRGBoost | $0.93_{\pm 0.02}$ | $0.89_{\pm 0.05}$ | $0.94_{\pm 0.07}$ | $0.23_{\pm 0.20}$ | $0.05_{\pm 0.01}$ | $0.05_{\pm 0.04}$ | $0.77_{\pm 0.30}$ | $0.15_{\pm 0.21}$ |
| GReaT | $0.88_{\pm 0.08}$ | $0.88_{\pm 0.05}$ | $0.87_{\pm 0.12}$ | $0.22_{\pm 0.21}$ | $0.12_{\pm 0.07}$ | $0.10_{\pm 0.06}$ | $0.49_{\pm 0.20}$ | $0.16_{\pm 0.21}$ |

Table 60: **Raw benchmark results of 13 tabular generators on "Elevators" dataset.** We report the normalised mean ± std metric values across datasets. We highlight the **First**, **Second** and **Third** best performances for each metric. For visualisation, we abbreviate "conditional independence" as "CI". SMOTE generally achieves the highest performance in capturing local structure (i.e., local utility or local CI), while diffusion models typically excel at capturing global structure (i.e., global CI or global utility).

| Generator | Density Estimation | | | | Privacy Preservation | | ML Efficacy | Structural Fidelity |
|---|---|---|---|---|---|---|---|---|
| | Shape ↑ | Trend ↑ | $\alpha$-precision ↑ | $\beta$-recall ↑ | DCR ↑ | $\delta$-Presence ↑ | Local utility ↑ | Global utility ↑ |
| $\mathcal{D}_{\text{ref}}$ | $1.00_{\pm 0.00}$ | $1.00_{\pm 0.00}$ | $1.00_{\pm 0.00}$ | $1.00_{\pm 0.00}$ | $0.00_{\pm 0.00}$ | $0.00_{\pm 0.00}$ | $1.00_{\pm 0.00}$ | $1.00_{\pm 0.00}$ |
| SMOTE | $0.85_{\pm 0.01}$ | $\mathbf{0.99}_{\pm 0.00}$ | $0.95_{\pm 0.00}$ | $\mathbf{0.81}_{\pm 0.00}$ | $0.02_{\pm 0.01}$ | $0.00_{\pm 0.00}$ | $\mathbf{0.92}_{\pm 0.06}$ | $0.39_{\pm 0.06}$ |
| BN | $\mathbf{0.87}_{\pm 0.01}$ | $\mathbf{0.96}_{\pm 0.00}$ | $\mathbf{0.96}_{\pm 0.00}$ | $0.28_{\pm 0.00}$ | $0.05_{\pm 0.00}$ | $0.00_{\pm 0.00}$ | $0.64_{\pm 0.12}$ | $0.61_{\pm 0.14}$ |
| TVAE | $0.82_{\pm 0.02}$ | $0.94_{\pm 0.00}$ | $0.91_{\pm 0.01}$ | $0.25_{\pm 0.01}$ | $0.05_{\pm 0.01}$ | $0.14_{\pm 0.06}$ | $0.78_{\pm 0.04}$ | $0.50_{\pm 0.05}$ |
| GOGGLE | $0.64_{\pm 0.09}$ | $0.87_{\pm 0.05}$ | $0.63_{\pm 0.13}$ | $0.20_{\pm 0.02}$ | $0.06_{\pm 0.02}$ | $2.18_{\pm 0.81}$ | $0.80_{\pm 0.00}$ | $0.16_{\pm 0.03}$ |
| CTGAN | $0.79_{\pm 0.02}$ | $0.94_{\pm 0.00}$ | $0.94_{\pm 0.04}$ | $0.14_{\pm 0.02}$ | $0.02_{\pm 0.00}$ | $0.11_{\pm 0.07}$ | $0.79_{\pm 0.06}$ | $0.14_{\pm 0.02}$ |
| NFlow | $0.78_{\pm 0.02}$ | $0.88_{\pm 0.01}$ | $0.77_{\pm 0.05}$ | $0.04_{\pm 0.01}$ | $0.12_{\pm 0.01}$ | $0.64_{\pm 0.43}$ | $0.48_{\pm 0.04}$ | $0.05_{\pm 0.01}$ |
| ARF | $0.85_{\pm 0.01}$ | $0.93_{\pm 0.01}$ | $0.97_{\pm 0.00}$ | $0.24_{\pm 0.01}$ | $0.04_{\pm 0.00}$ | $0.00_{\pm 0.00}$ | $0.67_{\pm 0.03}$ | $0.63_{\pm 0.07}$ |
| TabDDPM | $0.82_{\pm 0.04}$ | $0.94_{\pm 0.00}$ | $0.89_{\pm 0.04}$ | $\mathbf{0.36}_{\pm 0.04}$ | $0.03_{\pm 0.02}$ | $0.07_{\pm 0.03}$ | $0.56_{\pm 0.05}$ | $\mathbf{0.74}_{\pm 0.07}$ |
| TabSyn | $0.74_{\pm 0.14}$ | $0.92_{\pm 0.03}$ | $0.76_{\pm 0.20}$ | $0.29_{\pm 0.11}$ | $0.11_{\pm 0.07}$ | $1.73_{\pm 1.91}$ | $0.84_{\pm 0.05}$ | $0.71_{\pm 0.10}$ |
| TabDiff | $0.85_{\pm 0.02}$ | $0.93_{\pm 0.03}$ | $0.91_{\pm 0.07}$ | $0.25_{\pm 0.04}$ | $0.08_{\pm 0.01}$ | $0.06_{\pm 0.03}$ | $\mathbf{0.87}_{\pm 0.01}$ | $\mathbf{0.73}_{\pm 0.02}$ |
| TabEBM | $0.84_{\pm 0.02}$ | $0.94_{\pm 0.01}$ | $\mathbf{0.95}_{\pm 0.03}$ | $0.23_{\pm 0.01}$ | $0.09_{\pm 0.03}$ | $0.07_{\pm 0.03}$ | $0.59_{\pm 0.03}$ | $0.45_{\pm 0.03}$ |
| NRGBoost | $0.81_{\pm 0.05}$ | $0.91_{\pm 0.03}$ | $0.74_{\pm 0.22}$ | $0.21_{\pm 0.03}$ | $\mathbf{0.14}_{\pm 0.11}$ | $0.22_{\pm 0.25}$ | $0.78_{\pm 0.02}$ | $0.17_{\pm 0.02}$ |
| GReaT | $0.78_{\pm 0.09}$ | $0.91_{\pm 0.04}$ | $0.77_{\pm 0.14}$ | $0.21_{\pm 0.02}$ | $0.10_{\pm 0.04}$ | $0.54_{\pm 0.62}$ | $0.49_{\pm 0.00}$ | $0.18_{\pm 0.03}$ |

Table 61: **Raw benchmark results of 13 tabular generators on "H16" dataset.** We report the normalised mean ± std metric values across datasets. We highlight the **First**, **Second** and **Third** best performances for each metric. For visualisation, we abbreviate "conditional independence" as "CI". SMOTE generally achieves the highest performance in capturing local structure (i.e., local utility or local CI), while diffusion models typically excel at capturing global structure (i.e., global CI or global utility).

| Generator | Density Estimation | | | | Privacy Preservation | | ML Efficacy | Structural Fidelity |
|---|---|---|---|---|---|---|---|---|
| | Shape ↑ | Trend ↑ | $\alpha$-precision ↑ | $\beta$-recall ↑ | DCR ↑ | $\delta$-Presence ↑ | Local utility ↑ | Global utility ↑ |
| $\mathcal{D}_{\text{ref}}$ | $1.00_{\pm 0.00}$ | $1.00_{\pm 0.00}$ | $1.00_{\pm 0.00}$ | $1.00_{\pm 0.00}$ | $0.00_{\pm 0.00}$ | $0.00_{\pm 0.00}$ | $1.00_{\pm 0.00}$ | $1.00_{\pm 0.00}$ |
| SMOTE | $\mathbf{0.88}_{\pm 0.01}$ | $\mathbf{0.99}_{\pm 0.00}$ | $0.95_{\pm 0.00}$ | $\mathbf{0.83}_{\pm 0.00}$ | $0.05_{\pm 0.02}$ | $0.00_{\pm 0.01}$ | $\mathbf{0.98}_{\pm 0.02}$ | $0.45_{\pm 0.43}$ |
| BN | $\mathbf{0.89}_{\pm 0.01}$ | $\mathbf{0.99}_{\pm 0.00}$ | $\mathbf{0.99}_{\pm 0.00}$ | $\mathbf{0.61}_{\pm 0.01}$ | $0.03_{\pm 0.01}$ | $0.00_{\pm 0.00}$ | $0.80_{\pm 0.23}$ | $\mathbf{0.80}_{\pm 0.11}$ |
| TVAE | $0.85_{\pm 0.01}$ | $0.98_{\pm 0.00}$ | $0.94_{\pm 0.02}$ | $0.29_{\pm 0.01}$ | $0.10_{\pm 0.01}$ | $0.35_{\pm 0.50}$ | $0.86_{\pm 0.09}$ | $0.62_{\pm 0.22}$ |
| GOGGLE | $0.68_{\pm 0.21}$ | $0.95_{\pm 0.03}$ | $0.61_{\pm 0.37}$ | $0.23_{\pm 0.24}$ | $0.08_{\pm 0.03}$ | $6.50_{\pm 7.67}$ | $0.86_{\pm 0.17}$ | $0.20_{\pm 0.21}$ |
| CTGAN | $0.81_{\pm 0.02}$ | $0.97_{\pm 0.00}$ | $\mathbf{0.97}_{\pm 0.01}$ | $0.22_{\pm 0.03}$ | $0.05_{\pm 0.02}$ | $0.07_{\pm 0.09}$ | $\mathbf{0.87}_{\pm 0.13}$ | $0.20_{\pm 0.19}$ |
| NFlow | $0.83_{\pm 0.02}$ | $0.94_{\pm 0.00}$ | $0.86_{\pm 0.08}$ | $0.07_{\pm 0.01}$ | $0.11_{\pm 0.05}$ | $0.07_{\pm 0.05}$ | $0.57_{\pm 0.11}$ | $0.10_{\pm 0.14}$ |
| ARF | $\mathbf{0.90}_{\pm 0.00}$ | $\mathbf{0.98}_{\pm 0.00}$ | $0.94_{\pm 0.00}$ | $0.19_{\pm 0.01}$ | $0.06_{\pm 0.03}$ | $0.00_{\pm 0.00}$ | $0.74_{\pm 0.20}$ | $0.70_{\pm 0.16}$ |
| TabDDPM | $0.83_{\pm 0.06}$ | $0.96_{\pm 0.02}$ | $0.89_{\pm 0.07}$ | $\mathbf{0.40}_{\pm 0.06}$ | $0.04_{\pm 0.02}$ | $0.05_{\pm 0.07}$ | $0.65_{\pm 0.17}$ | $\mathbf{0.77}_{\pm 0.20}$ |
| TabSyn | $0.69_{\pm 0.20}$ | $0.95_{\pm 0.05}$ | $0.78_{\pm 0.23}$ | $0.24_{\pm 0.23}$ | $0.10_{\pm 0.06}$ | $3.88_{\pm 8.74}$ | $0.85_{\pm 0.18}$ | $0.67_{\pm 0.33}$ |
| TabDiff | $0.85_{\pm 0.04}$ | $0.96_{\pm 0.02}$ | $0.89_{\pm 0.07}$ | $0.24_{\pm 0.23}$ | $\mathbf{0.20}_{\pm 0.17}$ | $0.13_{\pm 0.12}$ | $\mathbf{0.87}_{\pm 0.16}$ | $\mathbf{0.71}_{\pm 0.28}$ |
| TabEBM | $0.87_{\pm 0.01}$ | $0.97_{\pm 0.01}$ | $\mathbf{0.96}_{\pm 0.02}$ | $0.26_{\pm 0.21}$ | $\mathbf{0.16}_{\pm 0.12}$ | $0.06_{\pm 0.08}$ | $0.64_{\pm 0.21}$ | $0.50_{\pm 0.05}$ |
| NRGBoost | $0.87_{\pm 0.01}$ | $0.95_{\pm 0.03}$ | $0.94_{\pm 0.01}$ | $0.24_{\pm 0.23}$ | $0.07_{\pm 0.02}$ | $0.23_{\pm 0.32}$ | $0.82_{\pm 0.23}$ | $0.17_{\pm 0.22}$ |
| GReaT | $0.77_{\pm 0.12}$ | $0.95_{\pm 0.03}$ | $0.88_{\pm 0.08}$ | $0.25_{\pm 0.22}$ | $\mathbf{0.12}_{\pm 0.07}$ | $\mathbf{0.72}_{\pm 0.83}$ | $0.58_{\pm 0.18}$ | $0.21_{\pm 0.21}$ |

Table 62: **Raw benchmark results of 13 tabular generators on "Liver" dataset.** We report the normalised mean $\pm$ std metric values across datasets. We highlight the First, Second and Third best performances for each metric. For visualisation, we abbreviate "conditional independence" as "CI". SMOTE generally achieves the highest performance in capturing local structure (i.e., local utility or local CI), while diffusion models typically excel at capturing global structure (i.e., global CI or global utility).

| Generator | Density Estimation | | | | Privacy Preservation | | ML Efficacy | Structural Fidelity |
| | Shape ↑ | Trend ↑ | $\alpha$-precision ↑ | $\beta$-recall ↑ | DCR ↑ | $\delta$-Presence ↑ | Local utility ↑ | Global utility ↑ |
|---|---|---|---|---|---|---|---|---|
| $\mathcal{D}_{ref}$ | $1.00_{\pm 0.00}$ | $1.00_{\pm 0.00}$ | $1.00_{\pm 0.00}$ | $1.00_{\pm 0.00}$ | $0.00_{\pm 0.00}$ | $0.00_{\pm 0.00}$ | $1.00_{\pm 0.00}$ | $1.00_{\pm 0.00}$ |
| SMOTE | $0.90_{\pm 0.02}$ | $0.97_{\pm 0.01}$ | $0.90_{\pm 0.03}$ | $0.81_{\pm 0.01}$ | $0.11_{\pm 0.01}$ | $0.01_{\pm 0.01}$ | $1.00_{\pm 0.05}$ | $0.43_{\pm 0.29}$ |
| BN | $0.91_{\pm 0.01}$ | $0.96_{\pm 0.01}$ | $0.94_{\pm 0.02}$ | $0.54_{\pm 0.03}$ | $0.23_{\pm 0.05}$ | $0.01_{\pm 0.01}$ | $0.91_{\pm 0.11}$ | $0.76_{\pm 0.14}$ |
| TVAE | $0.77_{\pm 0.01}$ | $0.91_{\pm 0.01}$ | $0.50_{\pm 0.05}$ | $0.47_{\pm 0.03}$ | $0.18_{\pm 0.04}$ | $0.12_{\pm 0.05}$ | $0.97_{\pm 0.07}$ | $0.72_{\pm 0.13}$ |
| GOGGLE | $0.65_{\pm 0.20}$ | $0.90_{\pm 0.04}$ | $0.77_{\pm 0.09}$ | $0.37_{\pm 0.19}$ | $0.18_{\pm 0.03}$ | $0.17_{\pm 0.12}$ | $0.94_{\pm 0.13}$ | $0.28_{\pm 0.21}$ |
| CTGAN | $0.49_{\pm 0.06}$ | $0.87_{\pm 0.03}$ | $0.61_{\pm 0.17}$ | $0.16_{\pm 0.06}$ | $0.29_{\pm 0.08}$ | $0.37_{\pm 0.27}$ | $0.95_{\pm 0.11}$ | $0.19_{\pm 0.13}$ |
| NFlow | $0.88_{\pm 0.01}$ | $0.92_{\pm 0.02}$ | $0.93_{\pm 0.04}$ | $0.47_{\pm 0.05}$ | $0.14_{\pm 0.02}$ | $0.02_{\pm 0.01}$ | $0.86_{\pm 0.10}$ | $0.37_{\pm 0.25}$ |
| ARF | $0.90_{\pm 0.01}$ | $0.96_{\pm 0.01}$ | $0.88_{\pm 0.05}$ | $0.48_{\pm 0.04}$ | $0.18_{\pm 0.01}$ | $0.01_{\pm 0.01}$ | $0.93_{\pm 0.13}$ | $0.81_{\pm 0.08}$ |
| TabDDPM | $0.84_{\pm 0.01}$ | $0.93_{\pm 0.02}$ | $0.88_{\pm 0.06}$ | $0.54_{\pm 0.02}$ | $0.13_{\pm 0.04}$ | $0.06_{\pm 0.05}$ | $0.87_{\pm 0.10}$ | $0.77_{\pm 0.14}$ |
| TabSyn | $0.86_{\pm 0.03}$ | $0.95_{\pm 0.01}$ | $0.89_{\pm 0.07}$ | $0.55_{\pm 0.02}$ | $0.17_{\pm 0.03}$ | $0.05_{\pm 0.05}$ | $1.00_{\pm 0.05}$ | $0.80_{\pm 0.13}$ |
| TabDiff | $0.86_{\pm 0.03}$ | $0.96_{\pm 0.02}$ | $0.87_{\pm 0.06}$ | $0.49_{\pm 0.09}$ | $0.20_{\pm 0.05}$ | $0.05_{\pm 0.05}$ | $1.00_{\pm 0.05}$ | $0.81_{\pm 0.13}$ |
| TabEBM | $0.86_{\pm 0.03}$ | $0.94_{\pm 0.01}$ | $0.89_{\pm 0.07}$ | $0.65_{\pm 0.10}$ | $0.13_{\pm 0.04}$ | $0.06_{\pm 0.05}$ | $0.93_{\pm 0.05}$ | $0.61_{\pm 0.05}$ |
| NRGBoost | $0.85_{\pm 0.02}$ | $0.91_{\pm 0.03}$ | $0.88_{\pm 0.07}$ | $0.52_{\pm 0.05}$ | $0.15_{\pm 0.03}$ | $0.06_{\pm 0.05}$ | $0.98_{\pm 0.06}$ | $0.37_{\pm 0.23}$ |
| GReaT | $0.78_{\pm 0.05}$ | $0.93_{\pm 0.02}$ | $0.81_{\pm 0.05}$ | $0.46_{\pm 0.11}$ | $0.17_{\pm 0.02}$ | $0.08_{\pm 0.04}$ | $0.87_{\pm 0.11}$ | $0.36_{\pm 0.23}$ |

Table 63: **Raw benchmark results of 13 tabular generators on "Sales" dataset.** We report the normalised mean $\pm$ std metric values across datasets. We highlight the First, Second and Third best performances for each metric. For visualisation, we abbreviate "conditional independence" as "CI". SMOTE generally achieves the highest performance in capturing local structure (i.e., local utility or local CI), while diffusion models typically excel at capturing global structure (i.e., global CI or global utility).

| Generator | Density Estimation | | | | Privacy Preservation | | ML Efficacy | Structural Fidelity |
| | Shape ↑ | Trend ↑ | $\alpha$-precision ↑ | $\beta$-recall ↑ | DCR ↑ | $\delta$-Presence ↑ | Local utility ↑ | Global utility ↑ |
|---|---|---|---|---|---|---|---|---|
| $\mathcal{D}_{ref}$ | $1.00_{\pm 0.00}$ | $1.00_{\pm 0.00}$ | $1.00_{\pm 0.00}$ | $1.00_{\pm 0.00}$ | $0.00_{\pm 0.00}$ | $0.00_{\pm 0.00}$ | $1.00_{\pm 0.00}$ | $1.00_{\pm 0.00}$ |
| SMOTE | $0.80_{\pm 0.00}$ | $0.96_{\pm 0.01}$ | $0.97_{\pm 0.01}$ | $0.79_{\pm 0.00}$ | $0.03_{\pm 0.01}$ | $0.00_{\pm 0.00}$ | $0.97_{\pm 0.04}$ | $0.39_{\pm 0.38}$ |
| BN | $0.79_{\pm 0.01}$ | $0.89_{\pm 0.00}$ | $0.94_{\pm 0.00}$ | $0.29_{\pm 0.00}$ | $0.22_{\pm 0.02}$ | $0.01_{\pm 0.00}$ | $0.58_{\pm 0.28}$ | $0.59_{\pm 0.24}$ |
| TVAE | $0.73_{\pm 0.01}$ | $0.87_{\pm 0.00}$ | $0.91_{\pm 0.02}$ | $0.27_{\pm 0.01}$ | $0.25_{\pm 0.03}$ | $0.03_{\pm 0.01}$ | $0.81_{\pm 0.12}$ | $0.62_{\pm 0.22}$ |
| GOGGLE | $0.57_{\pm 0.24}$ | $0.81_{\pm 0.11}$ | $0.68_{\pm 0.29}$ | $0.24_{\pm 0.25}$ | $0.18_{\pm 0.08}$ | $12.17_{\pm 17.42}$ | $0.80_{\pm 0.25}$ | $0.22_{\pm 0.20}$ |
| CTGAN | $0.71_{\pm 0.01}$ | $0.89_{\pm 0.01}$ | $0.95_{\pm 0.04}$ | $0.26_{\pm 0.02}$ | $0.11_{\pm 0.01}$ | $0.04_{\pm 0.08}$ | $0.83_{\pm 0.17}$ | $0.25_{\pm 0.25}$ |
| NFlow | $0.73_{\pm 0.01}$ | $0.84_{\pm 0.01}$ | $0.87_{\pm 0.11}$ | $0.24_{\pm 0.02}$ | $0.17_{\pm 0.05}$ | $0.07_{\pm 0.06}$ | $0.43_{\pm 0.19}$ | $0.14_{\pm 0.20}$ |
| ARF | $0.75_{\pm 0.04}$ | $0.89_{\pm 0.02}$ | $0.85_{\pm 0.09}$ | $0.38_{\pm 0.10}$ | $0.16_{\pm 0.03}$ | $1.38_{\pm 1.66}$ | $0.57_{\pm 0.28}$ | $0.62_{\pm 0.20}$ |
| TabDDPM | $0.66_{\pm 0.14}$ | $0.84_{\pm 0.08}$ | $0.47_{\pm 0.49}$ | $0.24_{\pm 0.25}$ | $0.24_{\pm 0.12}$ | $3.40_{\pm 6.24}$ | $0.42_{\pm 0.28}$ | $0.60_{\pm 0.40}$ |
| TabSyn | $0.78_{\pm 0.02}$ | $0.91_{\pm 0.00}$ | $0.96_{\pm 0.03}$ | $0.34_{\pm 0.14}$ | $0.16_{\pm 0.04}$ | $0.02_{\pm 0.02}$ | $0.90_{\pm 0.10}$ | $0.78_{\pm 0.20}$ |
| TabDiff | $0.78_{\pm 0.02}$ | $0.91_{\pm 0.00}$ | $0.96_{\pm 0.03}$ | $0.33_{\pm 0.15}$ | $0.15_{\pm 0.03}$ | $0.02_{\pm 0.02}$ | $0.90_{\pm 0.10}$ | $0.79_{\pm 0.20}$ |
| TabEBM | $0.77_{\pm 0.03}$ | $0.89_{\pm 0.02}$ | $0.91_{\pm 0.09}$ | $0.31_{\pm 0.19}$ | $0.13_{\pm 0.03}$ | $0.64_{\pm 1.37}$ | $0.66_{\pm 0.12}$ | $0.50_{\pm 0.03}$ |
| NRGBoost | $0.70_{\pm 0.09}$ | $0.84_{\pm 0.07}$ | $0.56_{\pm 0.40}$ | $0.27_{\pm 0.21}$ | $0.23_{\pm 0.10}$ | $0.15_{\pm 0.24}$ | $0.72_{\pm 0.37}$ | $0.18_{\pm 0.19}$ |
| GReaT | $0.75_{\pm 0.04}$ | $0.89_{\pm 0.02}$ | $0.85_{\pm 0.09}$ | $0.38_{\pm 0.10}$ | $0.16_{\pm 0.03}$ | $1.38_{\pm 1.66}$ | $0.51_{\pm 0.25}$ | $0.27_{\pm 0.25}$ |

Table 64: **Raw benchmark results of 13 tabular generators on "Space" dataset.** We report the normalised mean ± std metric values across datasets. We highlight the **First**, **Second** and **Third** best performances for each metric. For visualisation, we abbreviate "conditional independence" as "CI". SMOTE generally achieves the highest performance in capturing local structure (i.e., local utility or local CI), while diffusion models typically excel at capturing global structure (i.e., global CI or global utility).

| Generator | Density Estimation | | | | Privacy Preservation | | ML Efficacy | Structural Fidelity |
|---|---|---|---|---|---|---|---|---|
| | Shape ↑ | Trend ↑ | $\alpha$-precision ↑ | $\beta$-recall ↑ | DCR ↑ | $\delta$-Presence ↑ | Local utility ↑ | Global utility ↑ |
| $\mathcal{D}_{\text{ref}}$ | $1.00_{\pm 0.00}$ | $1.00_{\pm 0.00}$ | $1.00_{\pm 0.00}$ | $1.00_{\pm 0.00}$ | $0.00_{\pm 0.00}$ | $0.00_{\pm 0.00}$ | $1.00_{\pm 0.00}$ | $1.00_{\pm 0.00}$ |
| SMOTE | $0.98_{\pm 0.00}$ | $0.99_{\pm 0.00}$ | $0.98_{\pm 0.01}$ | $0.78_{\pm 0.01}$ | $0.08_{\pm 0.04}$ | $0.00_{\pm 0.00}$ | $0.96_{\pm 0.04}$ | $0.42_{\pm 0.41}$ |
| BN | $0.98_{\pm 0.00}$ | $0.99_{\pm 0.01}$ | $0.97_{\pm 0.01}$ | $0.57_{\pm 0.01}$ | $0.14_{\pm 0.03}$ | $0.00_{\pm 0.00}$ | $0.79_{\pm 0.19}$ | $0.92_{\pm 0.05}$ |
| TVAE | $0.87_{\pm 0.01}$ | $0.90_{\pm 0.01}$ | $0.85_{\pm 0.02}$ | $0.11_{\pm 0.01}$ | $0.20_{\pm 0.02}$ | $0.23_{\pm 0.18}$ | $0.75_{\pm 0.15}$ | $0.40_{\pm 0.36}$ |
| GOGGLE | $0.72_{\pm 0.21}$ | $0.88_{\pm 0.08}$ | $0.72_{\pm 0.24}$ | $0.21_{\pm 0.22}$ | $0.14_{\pm 0.04}$ | $1.91_{\pm 2.37}$ | $0.82_{\pm 0.21}$ | $0.15_{\pm 0.22}$ |
| CTGAN | $0.77_{\pm 0.05}$ | $0.93_{\pm 0.02}$ | $0.77_{\pm 0.10}$ | $0.05_{\pm 0.02}$ | $0.20_{\pm 0.06}$ | $0.21_{\pm 0.23}$ | $0.80_{\pm 0.22}$ | $0.08_{\pm 0.09}$ |
| NFlow | $0.89_{\pm 0.03}$ | $0.89_{\pm 0.02}$ | $0.91_{\pm 0.05}$ | $0.09_{\pm 0.02}$ | $0.15_{\pm 0.03}$ | $0.04_{\pm 0.04}$ | $0.57_{\pm 0.09}$ | $0.11_{\pm 0.12}$ |
| ARF | $0.97_{\pm 0.00}$ | $0.98_{\pm 0.00}$ | $0.98_{\pm 0.01}$ | $0.32_{\pm 0.01}$ | $0.10_{\pm 0.01}$ | $0.00_{\pm 0.00}$ | $0.73_{\pm 0.15}$ | $0.73_{\pm 0.17}$ |
| TabDDPM | $0.91_{\pm 0.01}$ | $0.96_{\pm 0.01}$ | $0.94_{\pm 0.03}$ | $0.38_{\pm 0.04}$ | $0.09_{\pm 0.04}$ | $0.05_{\pm 0.05}$ | $0.65_{\pm 0.15}$ | $0.80_{\pm 0.22}$ |
| TabSyn | $0.93_{\pm 0.02}$ | $0.97_{\pm 0.02}$ | $0.94_{\pm 0.03}$ | $0.35_{\pm 0.07}$ | $0.15_{\pm 0.03}$ | $0.04_{\pm 0.05}$ | $0.89_{\pm 0.11}$ | $0.79_{\pm 0.22}$ |
| TabDiff | $0.94_{\pm 0.03}$ | $0.97_{\pm 0.02}$ | $0.94_{\pm 0.03}$ | $0.34_{\pm 0.08}$ | $0.13_{\pm 0.02}$ | $0.04_{\pm 0.05}$ | $0.89_{\pm 0.12}$ | $0.78_{\pm 0.22}$ |
| TabEBM | $0.94_{\pm 0.02}$ | $0.95_{\pm 0.01}$ | $0.94_{\pm 0.03}$ | $0.32_{\pm 0.03}$ | $0.14_{\pm 0.02}$ | $0.04_{\pm 0.05}$ | $0.67_{\pm 0.12}$ | $0.47_{\pm 0.05}$ |
| NRGBoost | $0.93_{\pm 0.01}$ | $0.87_{\pm 0.09}$ | $0.90_{\pm 0.02}$ | $0.21_{\pm 0.22}$ | $0.20_{\pm 0.09}$ | $0.40_{\pm 0.40}$ | $0.84_{\pm 0.19}$ | $0.16_{\pm 0.21}$ |
| GReaT | $0.89_{\pm 0.03}$ | $0.91_{\pm 0.04}$ | $0.82_{\pm 0.11}$ | $0.27_{\pm 0.16}$ | $0.15_{\pm 0.03}$ | $0.12_{\pm 0.06}$ | $0.57_{\pm 0.14}$ | $0.17_{\pm 0.21}$ |

Table 65: **Raw benchmark results of 13 tabular generators on "Superconductivity" dataset.** We report the normalised mean ± std metric values across datasets. We highlight the **First**, **Second** and **Third** best performances for each metric. For visualisation, we abbreviate "conditional independence" as "CI". SMOTE generally achieves the highest performance in capturing local structure (i.e., local utility or local CI), while diffusion models typically excel at capturing global structure (i.e., global CI or global utility).

| Generator | Density Estimation | | | | Privacy Preservation | | ML Efficacy | Structural Fidelity |
|---|---|---|---|---|---|---|---|---|
| | Shape ↑ | Trend ↑ | $\alpha$-precision ↑ | $\beta$-recall ↑ | DCR ↑ | $\delta$-Presence ↑ | Local utility ↑ | Global utility ↑ |
| $\mathcal{D}_{\text{ref}}$ | $1.00_{\pm 0.00}$ | $1.00_{\pm 0.00}$ | $1.00_{\pm 0.00}$ | $1.00_{\pm 0.00}$ | $0.00_{\pm 0.00}$ | $0.00_{\pm 0.00}$ | $1.00_{\pm 0.00}$ | $1.00_{\pm 0.00}$ |
| SMOTE | $0.95_{\pm 0.00}$ | $1.00_{\pm 0.00}$ | $0.99_{\pm 0.00}$ | $0.45_{\pm 0.00}$ | $0.01_{\pm 0.00}$ | $0.00_{\pm 0.00}$ | $0.97_{\pm 0.03}$ | $0.41_{\pm 0.42}$ |
| BN | $0.96_{\pm 0.00}$ | $0.99_{\pm 0.00}$ | $0.99_{\pm 0.00}$ | $0.16_{\pm 0.00}$ | $0.07_{\pm 0.02}$ | $0.00_{\pm 0.00}$ | $0.72_{\pm 0.31}$ | $0.85_{\pm 0.09}$ |
| TVAE | $0.89_{\pm 0.00}$ | $0.94_{\pm 0.00}$ | $0.91_{\pm 0.01}$ | $0.00_{\pm 0.00}$ | $0.35_{\pm 0.01}$ | $0.04_{\pm 0.01}$ | $0.73_{\pm 0.16}$ | $0.42_{\pm 0.34}$ |
| GOGGLE | $0.86_{\pm 0.14}$ | $0.94_{\pm 0.04}$ | $0.82_{\pm 0.15}$ | $0.17_{\pm 0.08}$ | $0.26_{\pm 0.08}$ | $2.10_{\pm 3.52}$ | $0.82_{\pm 0.22}$ | $0.20_{\pm 0.22}$ |
| CTGAN | $0.86_{\pm 0.02}$ | $0.95_{\pm 0.00}$ | $0.85_{\pm 0.04}$ | $0.00_{\pm 0.00}$ | $0.38_{\pm 0.02}$ | $0.17_{\pm 0.20}$ | $0.76_{\pm 0.24}$ | $0.05_{\pm 0.05}$ |
| NFlow | $0.87_{\pm 0.01}$ | $0.84_{\pm 0.01}$ | $0.63_{\pm 0.02}$ | $0.00_{\pm 0.00}$ | $0.50_{\pm 0.03}$ | $4.94_{\pm 3.48}$ | $0.32_{\pm 0.08}$ | $0.01_{\pm 0.01}$ |
| ARF | $0.95_{\pm 0.00}$ | $0.99_{\pm 0.00}$ | $0.96_{\pm 0.00}$ | $0.02_{\pm 0.00}$ | $0.18_{\pm 0.01}$ | $0.00_{\pm 0.00}$ | $0.64_{\pm 0.26}$ | $0.54_{\pm 0.27}$ |
| TabDDPM | $0.66_{\pm 0.28}$ | $0.90_{\pm 0.06}$ | $0.45_{\pm 0.48}$ | $0.12_{\pm 0.12}$ | $0.14_{\pm 0.10}$ | $2.46_{\pm 3.84}$ | $0.40_{\pm 0.21}$ | $0.62_{\pm 0.42}$ |
| TabSyn | $0.91_{\pm 0.03}$ | $0.97_{\pm 0.01}$ | $0.91_{\pm 0.04}$ | $0.12_{\pm 0.12}$ | $0.23_{\pm 0.04}$ | $0.33_{\pm 0.51}$ | $0.85_{\pm 0.16}$ | $0.73_{\pm 0.28}$ |
| TabDiff | $0.93_{\pm 0.01}$ | $0.97_{\pm 0.02}$ | $0.93_{\pm 0.03}$ | $0.13_{\pm 0.11}$ | $0.24_{\pm 0.01}$ | $0.33_{\pm 0.51}$ | $0.85_{\pm 0.16}$ | $0.75_{\pm 0.25}$ |
| TabEBM | $0.92_{\pm 0.00}$ | $0.97_{\pm 0.01}$ | $0.93_{\pm 0.02}$ | $0.12_{\pm 0.12}$ | $0.18_{\pm 0.06}$ | $0.33_{\pm 0.51}$ | $0.47_{\pm 0.27}$ | $0.37_{\pm 0.10}$ |
| NRGBoost | $0.93_{\pm 0.01}$ | $0.89_{\pm 0.07}$ | $0.75_{\pm 0.17}$ | $0.12_{\pm 0.12}$ | $0.31_{\pm 0.08}$ | $4.29_{\pm 4.34}$ | $0.74_{\pm 0.32}$ | $0.12_{\pm 0.21}$ |
| GReaT | $0.90_{\pm 0.03}$ | $0.95_{\pm 0.01}$ | $0.86_{\pm 0.06}$ | $0.19_{\pm 0.05}$ | $0.23_{\pm 0.01}$ | $1.15_{\pm 0.74}$ | $0.48_{\pm 0.21}$ | $0.22_{\pm 0.22}$ |

Table 66: **Raw benchmark results of 13 tabular generators on "Wine" dataset.** We report the normalised mean $\pm$ std metric values across datasets. We highlight the **First**, **Second** and **Third** best performances for each metric. For visualisation, we abbreviate "conditional independence" as "CI". SMOTE generally achieves the highest performance in capturing local structure (i.e., local utility or local CI), while diffusion models typically excel at capturing global structure (i.e., global CI or global utility).

| Generator | Density Estimation | | | | Privacy Preservation | | ML Efficacy | Structural Fidelity |
|---|---|---|---|---|---|---|---|---|
| | Shape ↑ | Trend ↑ | $\alpha$-precision ↑ | $\beta$-recall ↑ | DCR ↑ | $\delta$-Presence ↑ | Local utility ↑ | Global utility ↑ |
| $\mathcal{D}_{\text{ref}}$ | $1.00_{\pm 0.00}$ | $1.00_{\pm 0.00}$ | $1.00_{\pm 0.00}$ | $1.00_{\pm 0.00}$ | $0.00_{\pm 0.00}$ | $0.00_{\pm 0.00}$ | $1.00_{\pm 0.00}$ | $1.00_{\pm 0.00}$ |
| SMOTE | $\mathbf{0.97}_{\pm 0.00}$ | $\mathbf{0.99}_{\pm 0.00}$ | $0.94_{\pm 0.01}$ | $\mathbf{0.67}_{\pm 0.01}$ | $0.08_{\pm 0.03}$ | $0.00_{\pm 0.00}$ | $\mathbf{0.98}_{\pm 0.02}$ | $0.44_{\pm 0.44}$ |
| BN | $\mathbf{0.97}_{\pm 0.00}$ | $0.93_{\pm 0.00}$ | $\mathbf{0.96}_{\pm 0.01}$ | $0.18_{\pm 0.01}$ | $\mathbf{0.22}_{\pm 0.02}$ | $0.01_{\pm 0.00}$ | $0.78_{\pm 0.11}$ | $0.49_{\pm 0.30}$ |
| TVAE | $0.89_{\pm 0.01}$ | $0.95_{\pm 0.01}$ | $0.78_{\pm 0.05}$ | $0.18_{\pm 0.02}$ | $\mathbf{0.23}_{\pm 0.04}$ | $\mathbf{0.07}_{\pm 0.10}$ | $0.88_{\pm 0.07}$ | $0.48_{\pm 0.31}$ |
| GOGGLE | $0.72_{\pm 0.23}$ | $0.92_{\pm 0.04}$ | $0.63_{\pm 0.32}$ | $0.18_{\pm 0.18}$ | $\mathbf{0.26}_{\pm 0.14}$ | $\mathbf{1.49}_{\pm 1.82}$ | $0.87_{\pm 0.17}$ | $0.12_{\pm 0.19}$ |
| CTGAN | $0.88_{\pm 0.01}$ | $0.97_{\pm 0.00}$ | $0.95_{\pm 0.01}$ | $0.16_{\pm 0.02}$ | $0.13_{\pm 0.04}$ | $0.03_{\pm 0.05}$ | $0.92_{\pm 0.08}$ | $0.17_{\pm 0.17}$ |
| NFlow | $0.89_{\pm 0.01}$ | $0.91_{\pm 0.00}$ | $0.92_{\pm 0.04}$ | $0.10_{\pm 0.02}$ | $0.16_{\pm 0.05}$ | $0.04_{\pm 0.01}$ | $0.70_{\pm 0.08}$ | $0.08_{\pm 0.12}$ |
| ARF | $\mathbf{0.96}_{\pm 0.00}$ | $\mathbf{0.98}_{\pm 0.00}$ | $\mathbf{0.97}_{\pm 0.01}$ | $0.22_{\pm 0.02}$ | $0.17_{\pm 0.03}$ | $0.00_{\pm 0.00}$ | $0.81_{\pm 0.13}$ | $0.66_{\pm 0.21}$ |
| TabDDPM | $0.93_{\pm 0.01}$ | $0.97_{\pm 0.01}$ | $0.93_{\pm 0.01}$ | $\mathbf{0.29}_{\pm 0.07}$ | $0.09_{\pm 0.05}$ | $0.02_{\pm 0.01}$ | $0.76_{\pm 0.12}$ | $\mathbf{0.75}_{\pm 0.25}$ |
| TabSyn | $0.93_{\pm 0.01}$ | $0.97_{\pm 0.01}$ | $0.95_{\pm 0.02}$ | $\mathbf{0.28}_{\pm 0.09}$ | $0.16_{\pm 0.03}$ | $0.01_{\pm 0.02}$ | $\mathbf{0.93}_{\pm 0.07}$ | $\mathbf{0.76}_{\pm 0.24}$ |
| TabDiff | $0.94_{\pm 0.01}$ | $\mathbf{0.98}_{\pm 0.01}$ | $\mathbf{0.96}_{\pm 0.03}$ | $0.27_{\pm 0.09}$ | $0.16_{\pm 0.04}$ | $0.01_{\pm 0.02}$ | $\mathbf{0.93}_{\pm 0.08}$ | $\mathbf{0.76}_{\pm 0.24}$ |
| TabEBM | $0.94_{\pm 0.00}$ | $0.97_{\pm 0.00}$ | $0.95_{\pm 0.02}$ | $0.26_{\pm 0.10}$ | $0.17_{\pm 0.05}$ | $0.01_{\pm 0.02}$ | $0.80_{\pm 0.07}$ | $0.45_{\pm 0.03}$ |
| NRGBoost | $0.94_{\pm 0.00}$ | $0.93_{\pm 0.04}$ | $0.91_{\pm 0.02}$ | $0.20_{\pm 0.17}$ | $0.13_{\pm 0.02}$ | $0.02_{\pm 0.01}$ | $0.90_{\pm 0.11}$ | $0.14_{\pm 0.18}$ |
| GReaT | $0.86_{\pm 0.08}$ | $0.92_{\pm 0.04}$ | $0.71_{\pm 0.23}$ | $0.21_{\pm 0.15}$ | $0.19_{\pm 0.07}$ | $\mathbf{0.36}_{\pm 0.45}$ | $0.71_{\pm 0.12}$ | $0.13_{\pm 0.18}$ |

