# OpenReview forum: "TabStruct: Measuring Structural Fidelity of Tabular Data"
_ICLR.cc/2026/Conference — ICLR 2026 Oral_

### Official Review · Reviewer_AASQ · 2025-10-31

**Soundness:** 4
**Presentation:** 4
**Contribution:** 4
**Rating:** 10
**Confidence:** 4

**Summary:**

The paper proposes a benchmarking framework including a novel metric for tabular data generators. The benchmarking framework is applied to 29 tabular SCM-based or real-world datasets and 13 generators.

The work has several contributions:
* A good motivation for introducing yet another benchmark for tabular data generators.
* A novel way of assessing the performance of data generators, based on structural fidelity, inspired by a causal perspective.
* A benchmarking framework with datasets, generators, and the evaluation pipeline.
* Many insights into existing failure modes and strengths of tabular data generators.
* An extreme commitment to reproducibility and scientific rigor in code and all parts of the paper.
* A detailed and extended overview of related work for all parts of the work, from motivation to benchmark to insights.

**Strengths:**

The authors have executed all the above-mentioned contributions with an extremely high quality. The motivation and reasons for the benchmark are novel and original, and well related to prior work. The work seems highly significant, as it improves upon the benchmarking landscape for tabular data generators and may finally deliver actionable insights that allow us to identify good, usable generators for practical applications. The paper is written very clearly, and all information is easily accessible to the reader.

**Weaknesses:**

There are no notable weaknesses as far as I can tell. Some of my questions might allude to potential weaknesses, but nothing concrete enough to mention here.

**Questions:**

* Using SCM-based datasets to benchmark tabular data generators is well motivated. Yet, it is hard to tell if datasets, for example, about physical laws, are representative of real-world tabular data. Real tabular data is often noisier than datasets we have / can get from data with a ground-truth SCM. Thus, it is hard to predict how useful and representative the subset of SCM-based data will be for TabStruct in the future to guide generator development. Have the authors considered ways to get more such data or more realistic data? For example, priors of tabular foundation models such as TabPFN often use SCM-based data for pretraining that might be more realistic. Likewise, they usually add noise to the SCMs' data.
* A core use of SCM-based data could be, as mentioned, to check for extrapolation instead of interpolation. This would also point to potentially benchmarking an entirely new kind of generators, that is, generators that can generate non-IID data. This would be akin to model benchmarks for non-IID data, e.g., https://arxiv.org/abs/2406.19380, rather than purely IID data, e.g., https://arxiv.org/abs/2506.16791. Given this comparison, TabStruct and the SCM-based data could enable benchmarks for non-IID data generators, and the Global CI would be appropriate for this. At the same time, Global utility might need to use non-IID models/validation splits. Was this intended by the distinction of interpolation vs extrapolation in Figure 1? And how does this framing position the current benchmark?
* The current formulation of the Global utility metric utilizes a binary output of the CI test. While theoretically well motivated, I was wondering if a more continuous output of the CI test might not be better suited. For example, one could return the alpha value at which the test fails, rather than 0/1. Then, the "decision boundary" of the Global utility would be smoother. Moreover, it would be more similar to the smoothness of global utility (which averages over continuous values) and maybe provide a higher correlation. Or, one could use a similar significance test in global utility, such as a Wilcoxon test at alpha 0.01 to test if Perf(D_ref) is equal to Perf(D) over samples or bootstraps.
* Why was balanced accuracy chosen as a metric for classification tasks? It is well established in model benchmarks that metrics such as ROC AUC for binary and log loss for multiclass classification are more appropriate for comparing models (e.g., see https://arxiv.org/abs/2506.16791 and its cited/related benchmarks and https://pages.cs.wisc.edu/~dpage/cs760/roc.pdf). I think here, a non-threshold-based metric would also be more appropriate. The impact of this choice might be less severe because AutoGluon was used, which, if optimized for balanced accuracy, should have employed threshold tuning by default (as required for evaluation for balanced accuracy). Do you know if threshold tuning was used in your experiments? On that note, the current formulation could also create some numerical problems for outliers where RMSE approaches 0. It might be worth improving the implementation with safeguards against numerical issues when computing relative metrics.
* The description of nested validation in Line 314 is very hard to parse, in my opinion. The appendix and text make it clear enough later. But the work could benefit from shortly introducing the validation method in the first sentence of the paragraph. I think in this case, a nested repeated shuffle split would be one way to describe it (?).  Furthermore, is stratification used for splits for classification datasets? If not, this might be important for imbalanced classification datasets in the future.
* How does the framework control randomness and seeds of the methods, data splits, and tuning?
* The correlation analysis in Section 4.1 is nice and insightful, but it remains unclear if the correlation is high enough to be representative enough for a good benchmark. There might be ways to test or verify that the correlation is sufficiently high by checking if model rankings are also correlated well enough -- especially for generators that have a very high, similar performance. Have you looked into such experiments next to the results in the appendix (Figure 6 etc)? The answer might depend on how valuable the small model difference might be, or if preserving the ranking of models is required. The danger is that developers might start to optimize for a potentially noisy metric when comparing only the best models or marginal improvements over the best models.
* The preprocessing as described in Appendix D.2 might be very suboptimal for some of the ML models from AutoGluon (as they expect to do their own model-specific preprocessing). But this make-everything-continuous preprocessing is needed for many of the generators, correct?
* As a final question / side note, a public, easy-to-share leaderboard of the methods might be cool, such as a Hugging Face leaderboard.

---

> ### Author Response · Authors · 2025-11-19
> **Response-AASQ (part 1/4)**
>
> Thank you very much for the positive and very encouraging feedback! Below we address all of your comments and have updated the manuscript accordingly to incorporate the additional clarifications and results (highlighted in blue). Please note that all references to lines, figures, tables, and sections refer to the revised manuscript by default.
>
> > **Q1: Real tabular data is often noisier … ways to get more such data or more realistic data? … they usually add noise to the SCMs' data.**
> >
>
> **Firstly**, we fully agree that evaluations of SCM data are more informative when the ground-truth SCMs are realistic. Consequently, we have made a concerted effort to ensure that the considered SCM datasets are as realistic as possible. This is exactly why, to assess structural fidelity, we employ expert-validated SCMs rather than toy SCMs used in prior work `[1]`.
>
> **Secondly**, we note that the physical law dataset shown in Figure 1 serves primarily as an illustrative example. For the SCM datasets utilised in TabStruct (Appendix D.1.1), the data-generation process explicitly incorporates exogenous noise (Lines 207-211 and Lines 1590-1592). Therefore, the SCM datasets used in our main evaluations can better reflect realistic causal structures with noise, in line with the prior datasets used by TabPFN.
>
> **Thirdly**, we did consider using the prior datasets of TabPFN. By default, we refer to TabPFN-v2 as TabPFN. However, to the best of our knowledge, TabPFN does not open-source its prior datasets `[8]`. Moreover, TabPFN is pretrained on around 100 million datasets `[2]`, and it remains unclear which of these datasets accurately reflects real-world causal structures. Some very recent work `[3]` has begun investigating this issue by ablating the impact of different priors, but there is still no robust criterion for determining what constitutes a realistic SCM. Ideally, one might evaluate the tabular generators across the entire pretraining corpus of 100 million datasets, but such an approach would likely be computationally prohibitive. As a practical and principled solution, we use expert-validated SCMs with exogenous noise. It is also worth noting that TabPFN converges well on these selected SCM datasets, suggesting that the chosen expert-validated SCMs likely lie within the broader distribution of causal structures represented in TabPFN’s pretraining corpus.
>
> > **Q2: Was this intended by the distinction of interpolation vs extrapolation in Figure 1? And how does this framing position the current benchmark?**
> >
>
> **Firstly**, yes, one of the core insights we aim to highlight through Figure 1 is that existing evaluation approaches tend to favour interpolation-based methods, such as SMOTE, while paying insufficient attention to the underlying data structures. These fundamental structures are crucial, as they enable generative models to extrapolate beyond the observed reference data towards more generalised data distributions.
>
> **Secondly**, we do not only position TabStruct as a comprehensive benchmark for evaluating the structural fidelity of tabular generative models. Our broader goal is to investigate the capabilities of current tabular generators to accurately capture the underlying structures of tabular data. We believe this to be a fundamental ability of a well-performing tabular generator (and a foundation model). In particular, generative models should be able to extrapolate beyond the training data distribution and generalise to unseen distributions encountered during inference.
>
> **Thirdly**, we agree that global CI naturally extends to benchmarking non-IID generators. When selecting the datasets for TabStruct, we also considered incorporating non-IID data (Lines 2534-2539), such as the temporal datasets from TabReD `[4]`. However, most existing generators assume purely IID data. Evaluating them on non-IID datasets could violate their key modelling assumption, potentially leading to unfair or uninformative comparisons. That said, we fully agree with the reviewer that extending the analysis of structural fidelity to non-IID data is a valuable direction. As TabStruct is fully open source, it offers a viable and extensible platform for the community. We are committed to incorporating non-IID datasets and supporting generators designed with non-IID assumptions, in line with the evolving needs and contributions of the research community.

---

> ### Author Response · Authors · 2025-11-19
> **Response-AASQ (part 2/4)**
>
> > **Q3: a more continuous output of the CI test … or use a similar significance test in global utility**
> >
>
> Below, we report the correlation between the suggested variants of global CI and global utility (Table R4). In the revised manuscript, this is addressed in Appendix E.4.
>
> **Experimental setup**
>
> - We consider two variants of global CI:
>     - **Global CI (discrete):** The current global CI reported in Section 4.
>     - **Global CI (continuous):** Instead of using binary CI test outcomes as in Equation 3, we compute the average $\alpha$ level at which CI tests fail across all features to obtain a continuous global CI score.
> - We consider two variants of global utility:
>     - **Global utility (continuous):** The current global utility reported in Section 4.
>     - **Global utility (discrete):** Instead of normalised downstream performance in Equation 4, we perform a Wilcoxon signed-rank test ($\alpha=0.01$) between $\text{Perf}(D_{\text{ref}})$ and $\text{Perf}(D)$ for each feature, then average the resulting binary outcomes.
> - All other settings (generators, data splitting, etc.) are the same as Section 4. The reported results are averaged over 10 runs.
>
> **Results**
>
> - **Important:** Table R4 shows that all variants of global CI and global utility, both continuous and discrete, exhibit strong mutual correlations. Notably, their correlation strengths are very similar, ranging between $[0.80, 0.86]$, indicating consistent alignment across formulations.
>
> We would like to thank the reviewer for this insightful suggestion. Our choice of the discrete version of global CI follows prior studies `[1, 5]` that have shown it to be a stable and interpretable measure. Given the consistently strong correlation, we plan to add a configurable parameter into the TabStruct functions, allowing users to select either the continuous or discrete variant for global CI and global utility.
>
> *Table R4.* Spearman’s rank correlation across the variants of global CI and global utility (p<0.001).
>
> |  | Global CI (discrete) | Global CI (continuous) |
> | --- | --- | --- |
> | Global utility (continuous) | 0.84 | 0.86 |
> | Global utility (discrete) | 0.80 | 0.83 |
>
> > **Q4: Why was balanced accuracy chosen as a metric for classification tasks? .. if threshold tuning was used in your experiments? … improving the implementation with safeguards against numerical issues**
> >
>
> We agree that ROC AUC and log loss are widely used and well-justified metrics for model comparison. In TabStruct, we choose balanced accuracy because it is robust to skewed class distributions `[6, 7]`, and it simplifies aggregation by using a single metric across all classification tasks.
>
> As the reviewer correctly noted, we explicitly called `TabularPredictor.calibrate_decision_threshold()` for the AutoGluon predictor to optimise the decision threshold for balanced accuracy. Given this setup, we believe that implementing global utility with balanced accuracy represents a reasonable design choice.
>
> Finally, across the 150K runs we have conducted, we have not encountered numerical issues in computing global utility. That said, we agree with the reviewer that incorporating safeguards against potential numerical instabilities is crucial for the robustness and practical usability of global utility. To this end, we plan to create a channel in the open-source repository to identify and address potential numerical issues arising from the use of the TabStruct benchmark suite.
>
> > **Q5: The description of nested validation in Line 314 is very hard to parse … is stratification used for splits for classification datasets?**
> >
>
> Thanks for the feedback. We have revised the descriptions of the nested cross-validation procedure in Lines 348-353. Specifically, we provide more detailed explanations of the nested cross-validation setup and the repeated shuffle of the data splitting strategy.
>
> In addition, as stated in Appendix D.2, we perform stratified splitting for all classification datasets to preserve the class distribution. We have also revised the manuscript to emphasise this point more clearly in the main text (Lines 351-352).

---

> ### Author Response · Authors · 2025-11-19
> **Response-AASQ (part 3/4)**
>
> > **Q6: How does the framework control randomness and seeds of the methods, data splits, and tuning?**
> >
>
> TabStruct supports reproducible and deterministic experiments through a global function, `seed_everything()` in `src/common/runtime/config/env.py` of the associated codebase, which fixes the randomness of all relevant third-party libraries, such as `PyTorch`, `NumPy`, and `Lightning`.
>
> As illustrated in Figure 2 and the README of the associated codebase, the benchmark suite consists of three modular components: data, model, and evaluation. These components are pluggable and independently configurable. In addition to the global seeding function, each component can have its own local seed. This design allows, for instance, a change in the random seed used for data splitting does not impact the initialisation of model weights. We implement this mechanism by explicitly decoupling the random seeds for each component. In practice, this means that, on the same machine and within the same virtual environment, running the same experiment command will yield identical results.
>
> The randomness introduced by hyperparameter tuning necessitates additional handling, especially since we utilise multithreading for efficiency. To address this, we explicitly initialise each `Optuna` trial with `optuna.samplers.TPESampler(seed)` and consistently start the tuning process using the default hyperparameter configuration of each model. As a result, TabStruct guarantees that the tuning trajectory remains deterministic and reproducible once the seed is set.
>
> > **Q7: if model rankings are also correlated well enough … developers might start to optimize for a potentially noisy metric**
> >
>
> Below, we report the Spearman’s rank correlation based on generator rankings across different metrics (Tables R5 and R6). In the revised manuscript, this is addressed in Appendix E.2.
>
> **Experimental setup**
>
> - We select the six SCM datasets (Table 5 and Table 6).
> - In addition to the correlation analysis of metric values, we further compute Spearman’s rank correlation between the generator rankings induced by three metrics: local utility, global CI, and global utility.
> - We further analyse the rank correlations among the top-5 generators according to global CI: TabDDPM, TabSyn, TabDiff, TVAE, and ARF.
> - All other settings (generators, data splitting, etc.) are the same as in Section 4. The reported results are averaged over 10 runs.
>
> **Results**
>
> - As shown in Table R5, across all 13 generators evaluated on six SCM datasets, generator rankings induced by global CI and global utility exhibit a strong correlation ($r_s=0.95$, $\text{p}<0.001$), whereas local utility shows substantially weaker alignment with the other two metrics.
> - When restricting the analysis to the top-5 generators based on global CI, Table R6 shows the correlation between global CI and global utility remains high ($r_s=0.92$, $\text{p}<0.001$). This suggests that global utility maintains a consistent ranking across high-performing generators, which supports our findings of robustness in distinguishing between top-performing models.
>
> *Table R5.* Spearman’s rank correlation based on generator rankings of 13 generators on six SCM datasets (p<0.001).
>
> |  | Local utility | Global CI |
> | --- | --- | --- |
> | Local utility | - | - |
> | Global CI | 0.29 | - |
> | Global utility | 0.35 | 0.95 |
>
> *Table R6.* Spearman’s rank correlation based on generator rankings of the top-5 generators (by global CI) on six SCM datasets (p<0.001).
>
> |  | Local utility | Global CI |
> | --- | --- | --- |
> | Local utility | - | - |
> | Global CI | 0.38 | - |
> | Global utility | 0.32 | 0.92 |
>
> > **Q8: The preprocessing … might be very suboptimal for some of the ML models … this make-everything-continuous preprocessing is needed for many of the generators?**
> >
>
> The preprocessing steps described in Appendix D.2 are applied only to the generative models. The synthetic data produced by generators is then transformed back to the original feature space before being passed to the downstream predictors. In other words, the AutoGluon models receive input data in the original, unprocessed feature space, allowing them to apply their own model-specific preprocessing strategies. We have updated the manuscript to highlight this point regarding AutoGluon internal preprocessing procedures in Lines 1668-1672.
>
> In addition, we apply the *“make-everything-continuous preprocessing”* as we empirically find that most generators benefit from such a preprocessing strategy, and it is consistent with observations reported in prior work `[7]`. An exception is TabDiff, which tends to perform better with ordinal encoding for categorical features. We have revised the manuscript to explicitly discuss these model-specific preprocessing choices in Lines 1663-1665.

---

> ### Author Response · Authors · 2025-11-19
> **Response-AASQ (part 4/4)**
>
> > **Q9: a public, easy-to-share leaderboard of the methods might be cool**
> >
>
> We fully agree that high-quality open-source initiatives are essential to the tabular generation community (and in general!). As TabStruct is fully open-source, we indeed plan to release an interactive version of the raw results through an online leaderboard (e.g., on Hugging Face, as suggested by the reviewer). This will enable researchers and practitioners to explore and analyse the benchmark results using customised workflows.
>
> **References**
>
> - `[1]` Tu, Ruibo, et al. "Causality for tabular data synthesis: A high-order structure causal benchmark framework." *arXiv preprint arXiv:2406.08311* (2024).
> - `[2]` Hollmann, Noah, et al. "Accurate predictions on small data with a tabular foundation model." *Nature* 637.8045 (2025): 319-326.
> - `[3]` Zhang, Xiyuan, et al. "Mitra: Mixed synthetic priors for enhancing tabular foundation models." *arXiv preprint arXiv:2510.21204* (2025).
> - `[4]` Rubachev, Ivan, et al. "Tabred: Analyzing pitfalls and filling the gaps in tabular deep learning benchmarks." *arXiv preprint arXiv:2406.19380* (2024).
> - `[5]` Spirtes, Peter, Clark N. Glymour, and Richard Scheines. *Causation, prediction, and search*. MIT press, 2000.
> - `[6]` Jiang, Xiangjian, et al. "ProtoGate: prototype-based neural networks with global-to-local feature selection for tabular biomedical data." *Proceedings of the 41st International Conference on Machine Learning*. 2024.
> - `[7]` Margeloiu, Andrei, et al. "Tabebm: A tabular data augmentation method with distinct class-specific energy-based models." *Advances in Neural Information Processing Systems* 37 (2024): 72094-72144.
> - `[8]` TabPFN official explanations on prior datasets: https://github.com/PriorLabs/TabPFN/issues/447

---

### Official Review · Reviewer_mx31 · 2025-10-31

**Soundness:** 3
**Presentation:** 3
**Contribution:** 3
**Rating:** 4
**Confidence:** 3

**Summary:**

The paper addresses the problem of evaluating tabular data generators. The paper argues that the existing benchmarks are insufficient because they either neglect the structural properties of the tabular data, or are biased towards downstream task performances, and are often limited to toy examples where the causal graph is known.

The paper proposes to incorporate structural fidelity as a core evaluation dimension, and introduces a metric called, "global utility", which is an SCM-free metric for assessing structural fidelity. A comprehensive benchamark " TabStruct" is proposed to evaluate 13 generators across 29 datasets including real-world and SCM-based data.

**Strengths:**

+ the proposed problem is highly relevant. The paper identified a significant gap in the evaluation of tabular generative models and moves beyond simple ML efficacy (performance on downstream tasks) and density estimation
+ the proposed metric is novel (global utility) and interesting.
+ the empirical validation is rigorous and comprehensive with 13 generators from 9 categories and 29 datasets. It is also commendable that the authors chose some challenging datasets where the models don't easily achieve perfect scores.
+ Tabstruct library is a valuable open-source contribution

**Weaknesses:**

- The global utility metric is a heuristic proxy, not a formal measure. Although the paper provides strong empirical evidence it is important to acknowledge that it is a proxy. Although it is acknowledged in the paper, it should probably be emphasized more.
- The metric measures "predictability" rather than the "causal structure". A generative model could learn powerful but spurious correlations that allow for excellent cross-prediction of variables. This would make it score high on global utility, but be structurally unfaithful.
- The "full-tuned" configuration of the global utility metric requires training an ensemble of predcitors for each of the columns in the dataset. That is computationally expensive, especially for high-dimension data. Although the "tiny-default" is much faster and seems to be stable ranking wise, the fundamental approach is very expensive and this could lead to scalablity issues in practice.
- The finding that the BAyesian networks and GOGGLE perform poorly on structural fidelity metrics is counter-intuitive. Although there is some explanation in the paper, this should probably be probed more deeply. Where is the failure happening?

**Questions:**

1. Could you elaborate on the theoretical gap between high global utility and the preservation of the Markov equivalence class? Are you aware of any hypothetical data-generating processes where a model could achieve high global utility while  violating key conditional independencies? How likely is this to happen in practice?
2. The high correlation between the global utility and the global_CI is taken as the validation of the global utility; but, the global_CI score is an imperfect ground truth because  the CI test have limitations themselves. So how does this inherent noise and potential inaccuracies of the CI-based ground truth affect your confidence in the correlation values and the conclusions you draw from it?
3. global utility is normalized by the performance on the reference data. How does this metric behave in low-data regimes? What is the relationship between the reliability of the global utility and the size and quality of the reference dataset?
. How does the computational cost of global utility scale with the number of features?

---

> ### Author Response · Authors · 2025-11-19
> **Response-mx31 (part 1/5)**
>
> Thank you for the detailed review and constructive feedback! We address all of the comments below and have updated the manuscript accordingly to incorporate the additional clarifications and results (highlighted in blue text). Please note that all references to lines, figures, tables, and sections refer to the revised manuscript by default.
>
> > **W1: It should probably be emphasized more that global utility metric is a heuristic proxy, not a formal measure.**
> >
>
> We are encouraged to hear that the reviewer acknowledges our transparency in presenting the proposed global utility as *“not a formal measure”*, and also finds that global utility is supported with “*strong empirical evidence”*. In line with the reviewer, we also strongly believe that honesty in presentation is a crucial aspect of scientific communication. In the revised manuscript, we have made this more explicit in several places:
>
> - **Section 3.3** (Lines 281-284): Add clarifications on the “SCM-inspired and real-data-validated” rationale for designing global utility.
> - **Section 4.2** (Lines 414-418): Add explicit emphases on the empirical correlation between global CI and global utility and the heuristic nature of global utility.
> - **Section 4.4** (Lines 516-524): Add discussion on the heuristic nature of global utility and its potential extension via future advances in causal modelling.

---

> ### Author Response · Authors · 2025-11-19
> **Response-mx31 (part 2/5)**
>
> > **W2: A generative model score high on global utility, but be structurally unfaithful.
> Q1-1: the theoretical gap between high global utility and the preservation of the Markov equivalence class?
> Q1-2: Any hypothetical data-generating processes with high global utility while violating key conditional independencies? How likely is this to happen in practice?**
> >
>
> **Firstly**, global utility is an SCM-free score *by design*, rather than a formal guarantee of a topology over causal structures, and we do acknowledge this as a limitation throughout the manuscript (Section 3, Section 4, and Section 5). It is an empirically validated measure for structural fidelity, motivated by the lack of ground-truth causal structures and the limitations of current causal discovery methods on real-world data `[1, 2, 3]`.
>
> Specifically, developing a theoretically provable structural fidelity metric for real-world tabular data is highly challenging, as ground-truth causal structures are rarely available. This limits our ability to validate structural fidelity directly on real-world data. In practice, the theoretical guarantee would require access to a reliable causal discovery method that can recover the underlying DAG. This, however, is hardly the case. Despite substantial research efforts, recent work `[1]` shows that even state-of-the-art causal discovery methods often perform poorly on real-world data and may lead to misleading results. These findings do *not* challenge the validity of existing causality theorems, but rather highlight the gap between theoretical assumptions and the complexities of real-world tabular data.
>
> **Secondly**, that said, TabStruct adopts an empirical but “SCM-inspired and real-data-validated” paradigm to address the absence of reliable causal structures. This is in line with prior work on tabular foundation models `[7, 8, 9]`. Within this framework, we present a three-step chain comprising theoretical motivations and quantitative validations to support the alignment between global utility and global structural fidelity.
>
> - **Step-1 Decomposition** (Lines 243-255): We split structural fidelity metrics into local CI (an individual variable) and global CI (across all variables), where global CI reflects preservation of the Markov equivalence class by definition (Lines 219-223).
> - **Step-2 Local CI vs. Local utility** (Lines 301-306): Achieving high local CI requires the generator to correctly identify (un-)informative features under various d-separation/connection sets. This aligns closely with Markov blanket based feature selection `[10]`. Importantly, correctly identifying (un-)informative features generally improves downstream predictive performance `[11]`. The strong correlation between local CI and local utility further validates their connections.
> - **Step-3 Aggregation to global level** (Lines 306-315): Building on the heuristics from Step-2, we examine the relationship between global CI and global utility. Our evaluation spans 13 generators across six SCM datasets, one of the most comprehensive to date (Section 2 and Appendix A). The results consistently show a strong correlation between global CI and global utility, in line with the insights derived in the above steps. Thus, global utility serves as an empirically effective measure for preservation of the Markov equivalence class when the ground-truth SCM is unavailable.
>
> **Thirdly**, as global utility provides empirically grounded insights, it does not rule out the theoretical possibility of constructing such *“hypothetical data-generating processes”* that achieve high global utility while violating some conditional independencies. However, to the best of our knowledge, we are not aware of prior studies that explicitly construct or analyse such a *“hypothetical generator”* in the context of structural fidelity for tabular data. Furthermore, our empirical results suggest the practical risk is substantially lower for global utility compared to existing metrics.
>
> Specifically, within the extensive scope of our experiments, and in particular across the SCM datasets, we did not find evidence of such behaviour that compromises the strong alignment between global CI and global utility. In particular, Figure 3 (left) shows that global utility achieves a correlation of 0.84 with global CI, notably higher than the next best metric, $\beta$-recall, which reaches only 0.49. This substantial gap indicates that a generator performing well on existing metrics, such as SMOTE, is considerably more prone to learning inaccurate structures. In contrast, a generator achieving a high global utility score is far less likely to learn unfaithful structures.

---

> ### Author Response · Authors · 2025-11-19
> **Response-mx31 (part 3/5)**
>
> > **W3: Global utility is computationally expensive.
> Q3-2: How does the computational cost of global utility scale with the number of features?**
> >
>
> The computational cost of global utility scales approximately linearly with the number of features, as global utility aggregates per-feature predictive performance. Thus, the “fully tuned” configuration can be computationally demanding on very high-dimensional datasets. This is precisely why we also introduce the “tiny-default” configuration, which is substantially cheaper while preserving the generator ranking. Even with our basic multithreading implementation, global utility is computationally efficient and practical. For instance, when applied to the largest benchmark dataset, *Link* (100,000 samples with 724 features), the “tiny-default” configuration completed in just 894 seconds (approximately 15 minutes). And the generator ranking by “tiny-default” is identical to that produced by “full tuned” for global utility. In contrast, local utility, which is highly sensitive to predictor configurations, required up to 4,542 seconds (approximately 76 minutes) to produce reliable evaluation results (e.g., generator rankings), even when using the same multithreading approach as global utility.
>
> We believe that a 15-minute computation budget is generally feasible in many research and practical settings, particularly given the unique role of global utility as an SCM-free metric for assessing global structural fidelity. The relative cost becomes even clearer when compared with the training time of modern generative models: For instance, training a diffusion-based model such as TabDiff takes 1,734 seconds on the *Link* dataset (around 29 minutes).
>
> Note that our current implementation of global utility is simple *by design.* As TabStruct is fully open source, it offers opportunities for further refinements and extensions. For instance, future research could explore feature selection strategies to enable global utility computation on a smaller subset of features (or feature groups) without sacrificing reliability. On the engineering side, more sophisticated parallelisation (e.g., finer-grained multithreading or GPU-based implementations) could further reduce wall-clock time.
>
> > **W4: Where is the failure of Bayesian Network and GOGGLE happening?**
> >
>
> **For Bayesian Network (BN)**, we discuss its potential limitations in Lines 430-431 and Lines 1462-1469. Below, we summarise the key points for clarity:
>
> - BN relies on a causal discovery algorithm to construct a directed acyclic graph (DAG). To ensure a fair evaluation, we compared eight causal discovery methods (Lines 1695-1705) and selected AutoCD `[4]` for its consistently competitive performance in global CI across the six SCM datasets.
> - However, prior work `[1]` has shown that even state-of-the-art causal discovery methods often perform poorly on challenging tabular data. This is highly relevant in our case, as we focus primarily on challenging datasets (Lines 341-342). As a result, the learned DAGs can be inaccurate, and the BN inherits this weakness, leading to poor performance in structural fidelity.
> - This suggests that, in our experiments, the main failure point for BN is the structure-learning step: when the discovered DAG is inaccurate, the downstream BN generator cannot achieve high structural fidelity.
>
> **For GOGGLE**, we discuss its limitations in Lines 450-451 and Lines 1730-1737. Here, we highlight the key points:
>
> - GOGGLE incorporates a trainable module to learn structure in a data-driven manner. However, even in Appendix C of its original paper `[6]`, the authors acknowledge that GOGGLE does not outperform BN in structure learning tasks. Our empirical results align with this: on our SCM benchmarks, BN achieves a global CI of 0.35, whereas GOGGLE reaches only 0.14 (Table 2).
> - This indicates that, in our setting, GOGGLE’s learned graphs can be even less faithful than those obtained by AutoCD for BN. In other words, the failure of GOGGLE is also likely to be a suboptimal structure-learning module, leading to poor structural fidelity.
>
> **Overall,** the suboptimal performance of BN and GOGGLE underscores the difficulty of obtaining reliable causal structures for real-world data. This further motivates our focus on an SCM-free, empirically validated measure for global structural fidelity when ground-truth causal structures are unavailable.
>
> > **Q1-1: the theoretical gap between high global utility and the preservation of the Markov equivalence class?**
> >
>
> Please refer to our response to W2.
>
> > **Q1-2: Any hypothetical data-generating processes with high global utility while violating key conditional independencies? How likely is this to happen in practice?**
> >
>
> Please refer to our response to W2.

---

> ### Author Response · Authors · 2025-11-19
> **Response-mx31 (part 4/5)**
>
> > **Q2: global CI score is an imperfect ground truth because the CI test have limitations themselves**
> >
>
> **Firstly**, to clarify the role of CI tests in our setting: for SCM datasets, the ground-truth CI statements are derived directly from the known causal graphs (Appendix B.1). In this sense, the *“ground truth”* is the set of CI statements encoded in the SCM, not the output of CI test. Thus, the potential *“inaccuracies”* in CI tests arise when these ground-truth CI statements are empirically validated on the evaluation datasets (i.e., $D_{\text{ref}}$ or $D_{\text{syn}}$).
>
> **Secondly**, we carefully assess existing causal metrics before conducting the structural fidelity evaluation, and we agree with the reviewer that some of these metrics, including global CI, are imperfect. However, global CI offers a reasonable trade-off between semantic expressiveness and computational feasibility. In particular, it operates at the CPDAG level `[3, 12]`, which makes it more informative than skeleton-level metrics and more reliable than DAG-level metrics (Lines 255-275). Moreover, CI tests yield consistently high scores on reference datasets, indicating their ability to distinguish between high- and low-quality datasets and thus produce meaningful fidelity assessments (Lines 1496-1501).
>
> **Thirdly**, we acknowledge that global CI has its imperfections, and the proposed global utility metric, due to its strong empirical correlation with global CI, may partially inherit these limitations. Nonetheless, achieving high global CI still implies that the generator captures a Markov-equivalent SCM to the ground truth, which is a meaningful achievement in the tabular data generation, as Figure 1 shows. Therefore, we position global utility as an SCM-inspired, empirically validated measure for global structural fidelity at the CPDAG level.
>
> > **Q3-1: How does this metric behave in low-data regimes? What is the relationship between the reliability of the global utility and the size and quality of the reference dataset?**
> >
>
> Below, we report the correlation between global CI and other metrics in low-data regimes (Table R3). In the revised manuscript, this is addressed in Appendix E.4 (Lines 2381-2387).
>
> **Experimental setup**
>
> - We select six SCM datasets (Table 5 and Table 6) and simulate varying levels of data availability by subsampling to smaller values of $N_{\text{full}}$. The corresponding reference sample size is $N_{\text{ref}} = N_{\text{full}} \times 0.8 \times 0.9$, as illustrated in Figure 5.
> - All other settings (generators, data splitting, etc.) are the same as Section 4. The reported results are averaged over 10 runs.
>
> **Results**
>
> - **Important:** As shown in Table R3, the proposed global utility metric consistently achieves the highest correlation with global CI across all evaluated sample sizes, clearly outperforming existing evaluation metrics. Notably, this holds even in very low-data scenarios, such as $N_{\text{full}} \le 500$. These results suggest that global utility serves as a robust and reliable measure for global structural fidelity across a wide range of data availability.
>
> *Table R3.* Spearman’s rank correlation with global CI across different degrees of data availability (p<0.001).
>
> | $N_{\text{full}}$ | Global utility (Ours) | Shape | Trend | $\alpha$-precision | $\beta$-recall | DCR | $\delta$-presence | Local utility | Local CI |
> | --- | --- | --- | --- | --- | --- | --- | --- | --- | --- |
> | 100 | **0.82** | 0.40 | 0.51 | 0.36 | 0.46 | -0.41 | -0.37 | 0.20 | 0.26 |
> | 500 | **0.83** | 0.43 | 0.51 | 0.32 | 0.52 | -0.49 | -0.39 | 0.24 | 0.22 |
> | 1,000 | **0.87** | 0.43 | 0.46 | 0.35 | 0.53 | -0.49 | -0.42 | 0.22 | 0.19 |
> | 5,000 | **0.83** | 0.47 | 0.41 | 0.42 | 0.50 | -0.48 | -0.45 | 0.18 | 0.13 |
> | 10,000 | **0.89** | 0.43 | 0.46 | 0.33 | 0.55 | -0.47 | -0.42 | 0.14 | 0.15 |
> | 100,000 | **0.84** | 0.47 | 0.47 | 0.37 | 0.49 | -0.43 | -0.40 | 0.14 | 0.22 |
>
> > **Q3-2: How does the computational cost of global utility scale with the number of features?**
> >
>
> Please refer to our response to W3.

---

> ### Author Response · Authors · 2025-11-19
> **Response-mx31 (part 5/5)**
>
> **References**
>
> - `[1]` Nastl, Vivian, and Moritz Hardt. "Do causal predictors generalize better to new domains?." *Advances in Neural Information Processing Systems* 37 (2024): 31202-31315.
> - `[2]` Zanga, Alessio, Elif Ozkirimli, and Fabio Stella. "A survey on causal discovery: theory and practice." *International Journal of Approximate Reasoning* 151 (2022): 101-129.
> - `[3]` Tu, Ruibo, et al. "Causality for tabular data synthesis: A high-order structure causal benchmark framework." *arXiv preprint arXiv:2406.08311* (2024).
> - `[4]` Chan, Gerlise, et al. "AutoCD: Automated Machine Learning for Causal Discovery Algorithms." *AutoML 2024 Methods Track*. 2024.
> - `[5]` McElfresh, Duncan, et al. "When do neural nets outperform boosted trees on tabular data?." *Advances in Neural Information Processing Systems* 36 (2023): 76336-76369.
> - `[6]` Liu, Tennison, et al. "Goggle: Generative modelling for tabular data by learning relational structure." *The Eleventh International Conference on Learning Representations*. 2023.
> - `[7]` Hollmann, Noah, et al. "Accurate predictions on small data with a tabular foundation model." *Nature* 637.8045 (2025): 319-326.
> - `[8]` Qu, Jingang, et al. "TabICL: A Tabular Foundation Model for In-Context Learning on Large Data." *ICML 2025-Forty-Second International Conference on Machine Learning*. 2025.
> - `[9]` Zhang, Xiyuan, et al. "Mitra: Mixed synthetic priors for enhancing tabular foundation models." *arXiv preprint arXiv:2510.21204* (2025).
> - `[10]` Fu, Shunkai, and Michel C. Desmarais. "Markov blanket based feature selection: a review of past decade." *Proceedings of the world congress on engineering*. Vol. 1. Hong Kong, China: Newswood Ltd., 2010.
> - `[11]` Cherepanova, Valeriia, et al. "A performance-driven benchmark for feature selection in tabular deep learning." *Advances in Neural Information Processing Systems* 36 (2023): 41956-41979.
> - `[12]` Spirtes, Peter, Clark N. Glymour, and Richard Scheines. *Causation, prediction, and search*. MIT press, 2000.

---

> ### Author Response · Authors · 2025-11-27
>
> Dear Reviewer mx31,
>
> Thank you again for taking the time to provide thoughtful feedback on our work. In light of our response and the new experiments, and with the discussion window closing soon, please let us know if you have any further questions or if there is anything else we can clarify. If not, we would appreciate it if you would consider updating your review based on our response.
>
> We are looking forward to your response. Thank you for your time.
>
> With regards,
>
> Authors

---

### Official Review · Reviewer_7rse · 2025-11-01

**Soundness:** 4
**Presentation:** 4
**Contribution:** 4
**Rating:** 8
**Confidence:** 4

**Summary:**

The paper proposes a comprehensive benchmark for evaluating tabular generative models. It identifies a key gap in existing evaluation measures and proposes new metrics to address this issue. While most evaluation metrics focus on dimensions such as density estimation, privacy preservation, or machine learning efficiency, these metrics are not tailored to assess whether synthetic data preserves the underlying structural (causal) relationships among variables. To address this, the paper proposes a new evaluation dimension called structural fidelity, and a new metric, global CI, to measure this new dimension.  Because, this metric requires knowledge about the underlying structural causal model (SCM), which is almost always unknown for real datasets, the paper also introduces a SCM free heuristic metric, global utility, which correlates strongly with global CI and can be used as a proxy to measure structural fidelity.

The paper performs extensive evaluations using 13 tabular generators across 29 datasets (including both expert-validated causal and real-world datasets) and provides many interesting insights about the performance of popular tabular generators and of the evaluation metrics.

**Strengths:**

The paper addresses the very important (and difficult) problem of evaluating the performance of tabular data generators.

It identifies a blind spot (structural fidelity) in the current evaluation metrics and proposes new metrics to measure it.

The experimental setup is as rigorous as it gets.

The very careful and detailed analyses of both the different generators and the different metrics provide corroboration to many observations in previous studies, as well as several new and valuable insights.

The paper provides a very thorough discussion of the limitations of the proposed metrics.

The framework’s modular design allows researchers to easily plug in new datasets, generators, and metrics, making it a great resource to the synthetic tabular data generation community.

**Weaknesses:**

The one point I think is missing from the paper is a comparison against the detection test metric (C2ST), which measures data fidelity by evaluating the discriminating ability of classifiers trained to discriminate between synthetic and real data. C2ST is a very popular metric, widely used in the field, which also aims to measure fidelity at a full dataset scale.

Given the large scale of the experiments presented on the paper, I understand that it would likely be difficult to include extensive comparisons during the short time frame of the discussion window. However, I think the paper would benefit from providing at least some preliminary comparisons of C2ST against the proposed global utility metric.

**Questions:**

Some of the benchmark datasets are very large (e.g., Higgs, and SCM datasets). How did the paper handle the TabPFN model in situations where the evaluation dataset had more than 10,000 rows? Was TabPFN removed from the predictor’s ensemble in these cases? (My understanding is that the current version of TabPFN can only handle datasets with up to 10,000 rows.)

This is only a suggestion, that the paper might find useful for future work. In Table 2, in addition to the 13 generators, the paper includes the reference (training) data, $D_{ref}$, for direct evaluation. This helps better ground the interpretation of the scores achieved by the generators, showing that the metrics can distinguish between high- and low-quality data.  But, perhaps, another interesting comparison would be to include direct comparisons not only to the reference set but also to a separate independent set (of the same size as the reference set) from each dataset (which is not touched during training). Since the independent set, $D_{indep}$, is by construction independent and identically distributed to the reference set, it could be used to estimate the performance of an ideal generator, truly capable of generating independent samples from the same distribution as the reference data.

Having these comparisons would potentially provide complementary information for the $D_{ref}$ direct evaluations. For instance, in the case of privacy metric such as the DCR, while $D_{ref}$ will produce a score equal to zero, $D_{indep}$ would estimate the median DCR score we would expect to see in an ideal generator. This would represent a baseline value and sort of “ground truth” value the generators should be aiming to. (While low DCR values indicate low privacy, high DCR scores are not necessarily a good thing either, since they usually indicate low data fidelity.) Additionally, the direct evaluation of $D_{indep}$ could also be potentially used to check when models are generating data that is too close to the training set (e.g., when a model is generating data $D_{syn}$ with better fidelity scores than $D_{indep}$).

To generate this independent set the paper would need to change its current data split and re-split the full dataset of each benchmark into a test set, a validation set, a training (reference) set, and a independent set of the same size as the training set. While this would require re-running all the analyses in that paper (what is certainly not feasible during the short discussion phase) perhaps the paper might want to consider implementing these comparisons in the future, as they might provide additional insights (and the paper aims for TabStruct to be an ongoing effort aiming to continue to evolve).

Minor suggestions:

Line 1515: “TarStruct” -> “TabStruct”

---

> ### Author Response · Authors · 2025-11-19
> **Response-7rse (part 1/2)**
>
> We thank the reviewer for the positive feedback! We address all of their comments below and have updated the manuscript accordingly to incorporate the additional clarifications and results (highlighted in blue text). Please note that all references to lines, figures, tables, and sections refer to the revised manuscript by default.
>
> > **W1: preliminary comparisons of C2ST against the proposed global utility metric**
> >
>
> We thank the reviewer for suggesting comparisons with C2ST. We have added preliminary experiments and discussion in the revised manuscript (Appendix E.1, Lines 1966-1975) to compare C2ST with the proposed global utility. Due to the non-trivial computational cost of running C2ST across all datasets and generators, we focus here on a representative subset and leave a full-scope evaluation to future work. Below, we summarise the key findings for clarity:
>
> **Experimental setup**
>
> - Following prior work `[1]`, we compute the detection score (C2ST) using logistic regression classifier. Higher C2ST scores indicate better performance, i.e., synthetic data that is harder to distinguish from real data.
> - We select three SCM classification datasets (Table 5): Hailfinder, Insurance, and Sangiovese.
> - All other settings (generators, data splitting, etc.) are the same as Section 4. The reported results are averaged over 10 runs.
>
> **Results**
>
> - **Important:** Table R2 shows that, on these three datasets, the proposed global utility metric exhibits a substantially stronger Spearman correlation with global CI (0.83) than C2ST (0.50). This suggests that global utility is more closely aligned with global structural fidelity.
> - C2ST, $\alpha$-precision, and $\beta$-recall all show relatively lower correlations with global CI than global utility. Although these sample-level metrics are designed to capture high-order interactions across features, they fail to explicitly attend to inter-feature causal interactions.
>
> *Table R2.* Spearman’s rank correlation with global CI on three selected SCM datasets (p<0.001).
>
> |  | Global CI $\uparrow$ |
> | --- | --- |
> | $\alpha$-precision | 0.38 |
> | $\beta$-recall | 0.47 |
> | C2ST | 0.50 |
> | **Global utility (Ours)** | **0.83** |

---

> ### Author Response · Authors · 2025-11-19
> **Response-7rse (part 2/2)**
>
> > **Q1: How did the paper handle the TabPFN model in situations where the evaluation dataset had more than 10,000 rows?**
> >
>
> The 10,000-sample limit in TabPFN is not a strict constraint. The official implementation of TabPFN `[2]` can process more than 10,000 samples if sufficient hardware resources are available (e.g., GPU memory). We enable this by setting `ignore_pretraining_limits=True` for `TabPFNRegressor`/`TabPFNClassifier` classes. With our computational setup (an NVIDIA A100 GPU with 80 GB memory), we find that TabPFN performs well across all the benchmark datasets. Therefore, we retain TabPFN in all our experiments.
>
> > **Q2: include direct comparisons to a separate independent set**
> >
>
> We agree with the reviewer that incorporating an independent set can provide valuable insights into the behaviour of tabular data generators, especially regarding overfitting and underfitting. Within the available timeframe, we have conducted a proof-of-concept investigation under this new experimental setting with an independent set. The corresponding experiments and results are presented in Appendix E.6 (Lines 2483-2495). Below, we highlight the main setup and findings.
>
> **Experimental setup**
>
> - As the reviewer suggested, we modify the data splitting strategy into $D_{\text{train}} : D_{\text{test}} : D_{\text{indep}} : D_{\text{valid}} = 3:3:3:1$, where $D_{\text{train}}$ serves as the reference data $D_{\text{ref}}$.
> - We select three SCM classification datasets (Table 5): Hailfinder, Insurance, and Sangiovese.
> - We focus on three representative metrics: Shape, Trend, and DCR. For each metric, we report scores for both $D_{\text{indep}}$ and the synthetic datasets, enabling direct comparison as suggested.
> - We select six representative generators: SMOTE, TVAE, CTGAN, TabSyn, TabDiff, and TabEBM.
> - All other settings (metric aggregation, etc.) are the same as Section 4. The reported results are averaged over 10 runs.
>
> **Results**
>
> - **Important:** Including $D_{\text{indep}}$ provides a complementary perspective on generator performance: it reveals whether a generator is closer to the training data $D_{\text{train}}$ than independent data $D_{\text{indep}}$, thereby helping to diagnose overfitting and underfitting.
> - For density estimation, achieving Shape = 1 or Trend = 1 with respect to $D_{\text{train}}$ does not necessarily indicate optimal generator performance, because even $D_{\text{indep}}$ does not reach such scores due to sampling variation. In our experiments, we observe that scores substantially higher than those of $D_{\text{indep}}$ can signal overfitting to $D_{\text{train}}$. A more appropriate benchmark is therefore to match or closely approximate the Shape and Trend scores of $D_{\text{indep}}$, rather than pushing them to 1.
> - As the reviewer correctly pointed out, simply maximising DCR (i.e., maximising distance to the closest real record) can severely degrade performance on other metrics. Figure 9 in Appendix E.6 shows that some generators achieve very high DCR but perform poorly in Shape and Trend. Our results suggest that a more balanced and practical criterion is to aim for a DCR higher than that of **$D_{\text{indep}}$**, while maintaining reasonable scores for other metrics, rather than maximising DCR.
> - An interesting observation from Figure 9 in Appendix E.6 is that SMOTE outperforms $D_{\text{indep}}$ in the Trend metric but performs much worse in DCR. This indicates that SMOTE tends to overfit to $D_{\text{train}}$, producing synthetic samples that are very close to the training data rather than learning a truly generalisable distribution.
>
> > **Q3: Line 1515: “TarStruct” -> “TabStruct”**
> >
>
> We thank the reviewer for kindly pointing this out. We have updated the manuscript accordingly (Line 1569).
>
> **References**
>
> - `[1]` Zhang, Hengrui, et al. "Mixed-Type Tabular Data Synthesis with Score-based Diffusion in Latent Space." *The Twelfth International Conference on Learning Representations*.
> - `[2]` TabPFN official implementation: https://github.com/PriorLabs/TabPFN/blob/1992222f5eba2ebc873abf165d6aa7b438f7253e/src/tabpfn/classifier.py#L298

---

> > ### Comment · Reviewer_7rse · 2025-11-26
> >
> > Thank you for your responses. They addressed all my comments. I am keeping my initial recommendation of acceptance of the paper.

---

### Official Review · Reviewer_njCP · 2025-11-01

**Soundness:** 3
**Presentation:** 2
**Contribution:** 3
**Rating:** 6
**Confidence:** 3

**Summary:**

The paper proposes a benchmark framework to assess methods for generating synthetic data based on existing datasets. The main novelty is that it introduces a metric to measure structural fidelity based on a global utility score without the need for ground-truth causal graphs. This enables the study of structural fidelity for real-world data where such graphs are typically unavailable. In empirical experiments, they first demonstrate a high correlation between the new metric and existing metrics on synthetic tasks (where the ground truth is available) and then assess the performance of different generators on real-world tasks.

**Strengths:**

**Novelty** The paper introduces a new metric that complements the assessment of data generators on real-world tasks

**Clarity** I enjoyed reading the paper as it is straightforward, with a good structure

**Impact** Studying tabular generators is a very relevant task for a broader audience at ICLR and a much-needed research direction directly related to developing better foundation models for tabular data

**Weaknesses:**

**Aggregation of Metrics** I understand the importance of aggregating metrics to provide concise and interpretable results. However, a considerable amount of information may be lost in the aggregation process. It would be valuable to include some indication of the variability or distribution of utility scores across features. For instance, are there notable outliers where the generator fails to capture feature usefulness, or are all features consistently represented? Such an analysis could strengthen the empirical insights.

**Readability** (This did not influence my rating, as it is straightforward to address.)
The paper employs multiple text styles (boldface, italics, color) without a clear or consistent rationale. For example, in lines 117-131, it is unclear why certain words are boldfaced. Similarly, the colored text and structure of (the huge) Table 2 do sufficiently aid visual comprehension or highlight the key messages. For example, consistent colors for the Top-3 and Bottom-3 results would help visualize similarities across columns. Moreover, showing real-world and SCM results for each metric side-by-side would support following the content of Section 4.2.

**Ethics Statement** This is not an ethics statement but a summary/conclusion.

**Questions:**

[clarification] Would reaching a utility score >1 be possible?

[clarification] What is the size of the generated dataset D_synth with respect to D_ref? I assume they are the same size, but it might also be interesting to analyze how the proposed metrics behave for larger or smaller synthetic datasets relative to the reference data.

---

> ### Author Response · Authors · 2025-11-19
> **Response-njCP (part 1/2)**
>
> Thank you for the positive feedback! We address all of the comments below and have updated the manuscript accordingly to incorporate the additional clarifications and results (highlighted in blue text). Please note that all references to lines, figures, tables, and sections refer to the *revised manuscript* by default.
>
> > **W1: indication of the variability or distribution of utility scores across features**
> >
>
> In the revised manuscript, this is addressed in Appendix E.3 (Lines 2218-2232) with fine-grained analysis on per-feature utility score distributions. Here, we highlight the key findings for clarity:
>
> - **TL;DR:** Global utility tends to be a reliable and stable lens for evaluating tabular data generators, as a high global utility score typically reflects consistently high utility across individual features.
> - **Finding 1 (high global utility):** Generators with relatively high global utility scores, such as TabSyn and TabDiff, tend to achieve balanced performance across features. This is mainly because the global utility assigns equal weight to each feature, thereby mitigating bias towards any particular feature. Consequently, if a generator achieves high global utility, it is less likely to be skewed towards a limited subset of features.
> - **Finding 2 (low global utility):** Generators with relatively low global utility scores generally fall into two categories.
>     - The first category tends to exhibit high utility for the target feature but low utility for the remaining ones, such as SMOTE and CTGAN. We hypothesise that this pattern primarily stems from their target-specific model design, which inherently biases them towards the target feature. The results are consistent with Figure 3 (Right), which shows that such models prioritise capturing the local structure of the target feature at the expense of the global structure.
>     - The second category generally performs poorly across all features. We attribute this to their misaligned architectures for the unique characteristics of tabular data. A representative example is GReaT. While it attempts to incorporate domain knowledge from LLMs into tabular data generation, the fundamental mismatch between the homogeneous textual modality of LLMs and the heterogeneous nature of tabular data hinders their ability to model tabular data structures effectively.
>
> > **W2: multiple text styles without a clear or consistent rationale**
> >
>
> Thanks for the suggestions. In the revised manuscript, we refine the (over)use of text styles to enhance clarity. Below, we outline our reasoning:
>
> - **Boldface**: Used for keywords (e.g., Line 154), summary sentences (e.g., Line 408), and figure captions (e.g., Line 79). Specifically, we have removed 10 boldface instances in the revised manuscript.
> - **Italics**: Applied to the first occurrence of new terminology (e.g., Lines 143-144), key points within long paragraphs (e.g., Lines 48-49), and important conclusions (e.g., Lines 136-138).
> - **Colours**: We note that the colours for the Top-3 methods remain consistent across different metrics in Table 2, as well as in all performance tables in Appendix F.
> - **Arrangement of Table 2 columns:** We agree with the reviewer that presenting results for each metric on SCM and real-world datasets side-by-side would improve the readability of the large table. However, such a side-by-side layout would nearly double the number of columns, making Table 2 too wide to fit into a single page under the official formatting template. As TabStruct is fully open-source, we will provide an interactive version of the raw results through an online leaderboard (e.g., Hugging Face). This will enable researchers and practitioners to explore and analyse the evaluation results using customisable visualisation layouts.

---

> > ### Comment · Reviewer_njCP · 2025-11-24
> >
> > Thanks a lot for your response.
> >
> > Minor correction to my feedback regarding Table 2: I meant using the same single color for the best, second-best, and third-best methods, and a different color for the three lowest-performing methods. As far as I understand, the statistical difference between methods was not quantified, and we mostly care whether the same set of methods performs well, not whether a single method achieves the same rank in both settings.

---

> ### Author Response · Authors · 2025-11-19
> **Response-njCP (part 2/2)**
>
> > **Q1: Would reaching a utility score >1 be possible?**
> >
>
> Yes. By definition, utility is normalised against the utility of the reference data, so a score of 1 corresponds to the predictive performance obtained when training on $D_\text{ref}$. According to Equation 4, a generator can therefore achieve a utility score exceeding 1, if a model trained on $D_\text{syn}$ attains better performance than the model trained on $D_\text{ref}$.
>
> > **Q2: What is the size of the generated dataset $D_\text{syn}$ with respect to $D_\text{ref}$? … how the proposed metrics behave for larger or smaller synthetic datasets?**
> >
>
> As mentioned in Appendix D.2, in all main experiments, we set the size of the generated data $D_\text{syn}$ equal to that of the reference data, i.e., $N_\text{syn} = N_\text{ref}$. This choice follows prior work `[1, 2]` and is intended to control for sample size, so that the evaluation metrics reflect differences in data quality, rather than being confounded by the quantity of synthetic data. In other words, setting $N_\text{syn} = N_\text{ref}$ provides a fair comparison and meaningful correlation analysis across evaluation metrics.
>
> That said, we agree with the reviewer that it is valuable to explore how $N_\text{syn}$ affects utility scores, particularly for evaluating the robustness of the proposed global utility in downstream tasks. To address this, we have added new experiments and corresponding discussion in the revised manuscript (Lines 2166-2174). Below, we summarise the key points and results (Table R1) for clarity:
>
> **Experimental setup**
>
> - Across six SCM datasets (Table 5 and Table 6), we fix $N_\text{ref}$ while varying the ratio $N_\text{ref}:N_\text{syn}$.
> - We evaluate three representative tabular generators: SMOTE, TabSyn, and TabDDPM, which achieve the best results in local CI, global CI, and global utility, respectively.
> - All other settings (data splitting, etc.) are the same as Section 4. The reported results are averaged over 10 runs.
>
> **Results**
>
> - When $N_\text{syn} < N_\text{ref}$ (ratios 5:1 and 3:1), global utility generally increases with the sample size of $D_\text{syn}$.
> - **Important:** Once $N_\text{syn} \geq N_\text{ref}$ (ratios 1:1, 1:3, 1:5), global utility tends to saturate. This observation further validates the robustness of the proposed global utility score and justifies our design rationale for using equal-sized $D_\text{syn}$ and $D_\text{ref}$ in the evaluation framework.
>
> *Table R1.* Global utility scores under different synthetic sample sizes.
>
> | $N_\text{ref}:N_\text{syn}$ | SMOTE | TabSyn | TabDDPM |
> | --- | --- | --- | --- |
> | `5:1` | 0.13 | 0.62 | 0.64 |
> | `3:1` | 0.25 | 0.72 | 0.74 |
> | `1:1` | **0.39** | **0.76** | 0.80 |
> | `1:3` | **0.39** | 0.75 | 0.79 |
> | `1:5` | 0.38 | **0.76** | **0.81** |
>
> **References**
>
> - `[1]` Zhang, Hengrui, et al. "Mixed-Type Tabular Data Synthesis with Score-based Diffusion in Latent Space." *The Twelfth International Conference on Learning Representations*.
> - `[2]` Shi, Juntong, et al. "TabDiff: a Mixed-type Diffusion Model for Tabular Data Generation." *The Thirteenth International Conference on Learning Representations*.

---

> > ### Comment · Reviewer_njCP · 2025-11-24
> >
> > Thanks a lot for the additional details (and also the effort you put into the rebuttal). I think all of them make the paper even stronger, and I am happy to raise my score.

---

### Author Response · Authors · 2025-11-19
**General response to all reviewers**

We thank the reviewers for the very constructive feedback!

### **(1) Summary of positive things**

- **Strong motivation and potential for high impact**
    - `njCP`: *“a very relevant task for a broader audience at ICLR”; “much-needed research direction”*
    - `7rse`: *“identifies a key gap/blind spot”; “addresses the very important (and difficult) problem”;*
    - `mx31`: *“problem is highly relevant”; “identified a significant gap”*
    - `AASQ`: *"novel and original, and well related to prior work”; “highly significant”; “deliver actionable insights“*
- **Novel SCM-free evaluation of structural fidelity**
    - `njCP`: *“a new metric”;“enables the study of structural fidelity for real-world data”*
    - `7rse`: *“a new evaluation dimension … and a new metric”;*
    - `mx31`: *“the proposed metric is novel (global utility) and interesting”*
    - `AASQ`: *“A novel way of assessing the performance of data generators”*
- **Comprehensive and rigorous empirical study**
    - `7rse`: *“a comprehensive benchmark”; “extensive evaluations”; "experimental setup is as rigorous as it gets”*
    - `mx31`: *“A comprehensive benchmark”; “rigorous and comprehensive”; “commendable … chose some challenging datasets”*
- **Insightful analyses supported by strong evidence**
    - `7rse`: *“corroboration to many observations in previous studies”; “new and valuable insights”*
    - `mx31`: *“the paper provides strong empirical evidence”*
    - `AASQ`: *“Many insights into existing failure modes and strengths”*
- **Clear and well-organised presentation**
    - `njCP`: *“enjoyed reading the paper”*
    - `7rse`: *“very careful and detailed analyses”; “very thorough discussion”*
    - `AASQ`: *“A detailed and extended overview of related work”; “written very clearly”; “all information is easily accessible”*
- **High standards of reproducibility**
    - `7rse`: *“easily plug in new datasets, generators, and metrics”; ”great resource to the community.”*
    - `mx31`: *“a valuable open-source contribution”*
    - `AASQ`: *“An extreme commitment to reproducibility and scientific rigor in code and all parts of the paper.”*

### **(2) Summary of our responses and new experiments**

We replied to **all** questions and concerns raised by the reviewers. Specifically, we provide **7 new experiments**. The new results are consistent with the main text and further support the validity of the proposed TabStruct and global utility.

*Table R0*. Number of our responses to the reviewers’ comments (`#Raised`/`#Replied`).

| Reviewer | # Weaknesses  | # Questions | # New experiments |
| --- | --- | --- | --- |
| `njCP` | 2/2 | 2/2 | 2 |
| `7rse` | 1/1 | 3/3 | 2 |
| `mx31` | 4/4 | 3/3 | 1 |
| `AASQ` | 0/0 | 9/9 | 2 |

### **(3) Summary of updates to the manuscript**

We have revised the manuscript according to the feedback. In the revised manuscript, all changes are highlighted in blue. Below, we summarise the updates:

**Main text**

- `[njCP]` (throughout the main text): Minor edits to ensure consistency in text styles.
- `[mx31]` (Section 3.3, Section 4.2, and Section 4.4): Additional emphases on the empirical relationship between global CI and the proposed global utility.
- `[mx31]` (Section 3.2): Expanded explanation of the rationales behind CPDAG-level evaluation.
- `[AASQ]` (Section 4): Clarifications on the data-splitting strategy employed in TabStruct.

**Appendix**

- `[njCP]` (Appendix E.3): A fine-grained analysis of utility score distributions across different features.
- `[njCP]` (Appendix E.2): An ablation study examining the stability of the proposed global utility under varying synthetic sample sizes.
- `[7rse]` (Appendix E.1): An extended discussion on detection score C2ST and structural fidelity.
- `[7rse]` (Appendix E.6): Additional discussion and preliminary experiments exploring the concept of separate independent sets.
- `[mx31]` (Appendix E.4): An ablation study investigating the robustness of the proposed global utility across varying levels of data availability, with a particular focus on low-data scenarios.
- `[AASQ]` (Appendix E.4): An extended discussion on the variants of global CI and the proposed global utility.
- `[AASQ]` (Appendix D.2): Clarifications on the data preprocessing protocols used by TabStruct.
- `[AASQ]` (Appendix E.2): An extended analysis of the correlation between generator rankings induced by global CI and the proposed global utility.

---

### Meta-Review · Area_Chair_xwRY · 2026-01-05

**Summary:**

This paper presents a new evaluation protocol for tabular data ganertative models. They first defined a much more complicated structural causal model-based evaluation method than existing work, e.g., CauTabBench. The key point is the conditional independence in the structural causal relationship. Existing generative models, which are known to work well, are not able to actually capture those local/global conditional independence cases, according to this paper and moreover, these results are well reproduced in their benchmark code. At the end, they deisnged a benchmark framework consisting of 13 methods and 29 datasets, which is still small. More methods and mode datasets are needed.

**Reviewer Concerns:**

Overall, reviwers are satisfied with the paper while one reviewer raises several questions regarding i) theoretical ground, ii) robustness to noise, and iii) robustness under the low-data regime. I think this reviewer's concerns make sense since this paper's evaluation metric is based on a largely heuristical approach.

If the authors are able to enhance the designed metric or prove mathematically those theoretical factors, I think that this paper well deserves an oral presentation. However, it seems premature at this moment.

**Reviewer Scores:**

Three reviewers, njCP, 7rse, and AASQ, are much satisfied and their final scores are expected to be 8, 8, and 10, respectively (if they are engaged with the discussion with the authors).

However, mx31 seems not completely satisfied and may stay at his/her initial score.

---

### Decision · Program_Chairs · 2026-01-26

Accept (Oral)